# Thor: a platform for cell-level investigation of spatial transcriptomics and histology

Pengzhi Zhang [1,2,3,4,14], Weiqing Chen [1,5,14], Tu N. Tran[1,2,3,4], Minghao Zhou[6], Kaylee N. Carter [4], Ibrahem Kandel[1,2,3,4], Shengyu Li [1,2,3,4], Xen Ping Hoi [7,8,9], Yuxing Sun[10], Li Lai [4], Keith Youker [4], Qianqian Song[6], Yu Yang[11], Fotis Nikolos [7,8], Zejuan Li [12,13], Keith Syson Chan[7,8], John P. Cooke [2,3,4] & Guangyu Wang [1,2,3,4,5] ✉

Spatial transcriptomics links gene expression with tissue morphology, however, current tools often prioritize genomic analysis, lacking integrated image interpretation. To address this, we present Thor, a comprehensive platform for cell-level analysis of spatial transcriptomics and histological images. Thor employs an anti-shrinking Markov diffusion method to infer single-cell spatial transcriptome from spot-level data, effectively combining gene expression and cell morphology. The platform includes 10 modular tools for genomic and image-based analysis, and is paired with Mjolnir, a web-based interface for interactive exploration of gigapixel images. Thor is validated on simulated data and multiple spatial platforms (ISH, MERFISH, Xenium, Stereo-seq). Thor characterizes regenerative signatures in heart failure, screens breast cancer hallmarks, resolves fine layers in mouse olfactory bulb, and annotates fibrotic heart tissue. In high-resolution Visium HD data, it enhances spatial gene patterns aligned with histology. By bridging transcriptomic and histological analysis, Thor enables holistic tissue interpretation in spatial biology.

The complex organization of cells within tissues is profoundly connected to their biological function. This underpins the widespread utility of histological images in health and disease. The development of computational methods empowered by deep learning on histological images has drastically enhanced efficiency and accuracy in tissue analysis in diverse applications[1], including automated cancer diagnosis[2], survival prediction[3], histopathology image classification and retrieval[4], tissue segmentation[5,6], nucleus and cell segmentation[7–9],

and in silico staining[10]. Furthermore, rapid advancements in high-throughput technologies such as RNA sequencing (RNA-seq) and whole genome sequencing (WGS) are transforming the landscape of conventional histological analysis, offering unprecedented insights beyond tissue images. For example, recent research has demonstrated that the integration of histological images with genomic biomarker mutations and biological pathways leads to accurate predictions of survival across diverse conditions[3,11]. In the evolving landscape of

[1]Center for Bioinformatics and Computational Biology, Houston Methodist Research Institute, Houston, TX, USA. [2]Center for RNA Therapeutics, Houston Methodist Research Institute, Houston, TX, USA. [3]Department of Cardiothoracic Surgery, Weill Cornell Medicine, Cornell University, New York, NY, USA. [4]Department of Cardiovascular Sciences, Houston Methodist Research Institute, Houston, TX, USA. [5]Department of Physiology, Biophysics & Systems Biology, Weill Cornell Graduate School of Medical Science, Weill Cornell Medicine, Cornell University, New York, NY, USA. [6]Department of Health Outcomes and Biomedical Informatics, University of Florida, Gainesville, FL, USA. [7]Department of Urology, Houston Methodist Research Institute, Houston, TX, USA. [8]Spatial Omics Core, Neal Cancer Center, Houston Methodist Research Institute, Houston, TX, USA. [9]Graduate Program in Biomedical Sciences, Cedars-Sinai Medical Center, Los Angeles, CA, USA. [10]School of Computer Science, Georgia Institute of Technology, Atlanta, GA, USA. [11]Department of Pathology, Immunology and Laboratory Medicine, College of Medicine, University of Florida, Gainesville, FL, USA. [12]Department of Pathology and Genomic Medicine, Houston Methodist Hospital, Houston Methodist Research Institute, Houston, TX, USA. [13]Weill Cornell Medical College, New York, NY, USA. [14]These authors contributed equally: Pengzhi Zhang, Weiqing Chen. ✉e-mail: gwang2@houstonmethodist.org

biological investigation, spatially resolved molecular technologies have become a pivotal focus for unraveling cellular diversity, tissue organization, and functions. Spatial omics data have been incorporated and routinely acquired by programs such as the Human Cell Atlas (HCA) and the Human Biomolecular Atlas Program (HuBMAP), advancing the construction of comprehensive spatial maps featuring various biomolecules, including RNA, proteins, and metabolites[12,13]. A widely adopted molecular technology is spatial transcriptomics (ST), which involves slicing tissues into thin layers for hematoxylin and eosin (H&E) staining and spatial sequencing, enabling simultaneous investigation of tissue/cellular phenotype and molecular mechanism on the same slide.

Recent efforts to advance ST analysis have focused on incorporating spatial neighborhood information[14], or integrating histology images[15–17]. However, these tools typically operate at subspot or superpixel spatial scales, which do not correspond to individual cells, hindering biologically relevant insights, particularly in contexts requiring cell-level data, such as analyzing ligand-receptor interactions. Another branch of ST analysis frameworks addresses cellular heterogeneity by resolving cell-type compositions within spatial spots[18–20]. However, these approaches do not infer cell-level gene expression and are further restricted by the quality and availability of single-cell RNA-seq (scRNA-seq) reference data, especially for formalin-fixed paraffin-embedded (FFPE) tissues where transcriptomic data quality is often compromised. While emerging methods enable cellular-level histological structure analysis[21,22], similarly they do not generate single-cell resolution gene expression matrices, thereby hindering downstream functional or molecular analyses. Moreover, those platforms are mostly tailored to specific tasks (e.g., deconvolution), whereas comprehensive analysis platforms (e.g., Seurat) prioritize -omics analysis without deeply analyzing histopathological images[23–25].

To meet the urgent need for jointly analyzing genomics and histology, we present a multi-modal platform, Thor, for bridging and exploring cellular phenotypes and molecular insights. Thor enhances the incorporation of morphology and transcriptome data of individual cells by inferring cell-resolution transcriptome from spot-level ST data using an anti-shrinking Markov graph diffusion method. Moreover, Thor features extensible modules for comprehensive genomic analyses, such as immune response, functional pathway enrichment, transcription factor (TF) activity, and copy number variation (CNV), alongside tissue analyses such as semi-supervised tissue annotation and nucleus detection. Additionally, we develop Mjolnir, a user-friendly web-based platform for interactive exploration of cellular organization and pathogenesis in tissues, on a laptop, with no coding required.

We elucidated the principles of Thor and rigorously assessed its effectiveness and accuracy through simulations and various datasets, obtained from high-resolution experimental methods, including in situ hybridization (ISH), multiplexed error-robust fluorescence in situ hybridization (MERFISH)[26], spatio-temporal enhanced resolution omics-sequencing (Stereo-seq)[27], and Xenium[28]. Thor outperformed state-of-the-art (SOTA) methods in predicting cell-level ST on a breast cancer dataset using Xenium data as the ground truth. We analyzed a mouse olfactory bulb (MOB) tissue, human breast cancer tissues, and multi-sample heart failure patient tissues. Thor revealed a refined layered structure in MOB and identified distinct gene modules. In heart failure, Thor quantified fibrotic regions across different heart zones. Furthermore, we collected in-house heart failure samples from patients who received left ventricular assist device (LVAD) implantation to study the signature genes in vascular regeneration. We characterized regenerative signatures in heart failure and validated them through immunofluorescence (IF) staining. In breast cancer, Thor conducted an unbiased screening of breast cancer hallmarks, uncovering the intricate heterogeneity of immune responses in tumor regions. In summary, Thor enables comprehensive interpretation of ST data at the single-cell and whole-transcriptome levels, delivering advanced functional insights and providing an interactive interface for in-depth analyses.

## Results

### Thor infers cell-resolution spatial transcriptome for multi-modal analysis

Histological images and high-throughput sequencing data are widely adopted for various applications[2,29–31]. Despite their significance, these two sources of information are often examined independently with separate tools. Sequencing-based ST and the paired histological whole slide image (WSI) capture inherent cellular structures in the tissue at different resolutions, providing complementary information. For example, in human heart tissues with myocardial infarction (MI), we observed that the projection of histological features onto principal components segregated tissues at cellular resolution (Fig. 1a and Supplementary Note 1). Similarly, spatial patterns can be discerned through marker gene expression at a coarser resolution (spot level). Clustering results of spots by using either source of features were consistent and complementary, as demonstrated in the human MI samples, a human ductal carcinoma in situ (DCIS) sample, and a MOB sample (See details in Supplementary Note 1 and Supplementary Fig. 29). Previous studies also indicated that spatial gene expression can be predicted or refined based on histological images[15–17]. Therefore, we hypothesize that it is feasible to recover cell-level resolution transcriptomics data by learning shared patterns from both the histology and the transcriptome.

Multi-modal analysis in Thor involves two key steps. First, elevating spot-resolution ST data to single-cell resolution (Fig. 1a). Second, in-depth genomics and tissue image analyses (Fig. 1b, c). In the first step, Thor (i) applies deep learning methods to segment cells/nuclei from the WSI, termed in silico cells; (ii) extracts morphological and spot-level transcriptomic features into a combinatory feature space to construct a cell–cell network; (iii) creates a Markov transition matrix, representing the probabilities of transitioning from a cell to every other cell in the system in one step; (iv) infers gene expression of the in silico cells by data diffusion with the transition matrix (Fig. 1a). Thor represents the cellular patterns using a nearest neighbors graph, where cells are connected according to their distances in the combinatory feature space reflecting the physical separation, and the histological and genomic complexity. The Markov transition matrix is constructed such that information from "homogeneous" spots asymmetrically corrects information from "heterogeneous" spots, where heterogeneity of a spot is determined by the enclosed cells in the combinatory feature space (Fig. 1a). In the second step, we establish a standardized genomics analysis framework for in-depth research and clinical practice. The genomics analysis encompasses a wide array of insights, including cell type annotation, immune response analysis, biological functional pathway analysis, differential gene expression analysis, spatially expressed module detection, TF activity analysis, and CNV analysis (Fig. 1b). Thor also includes tissue image analysis tools including nucleus segmentation, region of interest (ROI) selection, and semi-supervised annotation (SSA). To enhance accessibility and usability, we introduce a web-based platform, Mjolnir, that seamlessly visualizes both histological images and genomic analyses (Fig. 1c). Altogether, Thor elevates tissue analysis by integrating image analysis and genomic insights.

### Thor demonstrates accuracy and robustness in simulation data

We systematically evaluated Thor's accuracy and robustness in simulations under realistic experimental conditions. We simulated expression profiles for 1000 genes in 6579 cells, whose spatial positions were extracted from a mouse cerebellum tissue as the ground truth[32]; and based on those cells, we created "spots" by aggregating gene

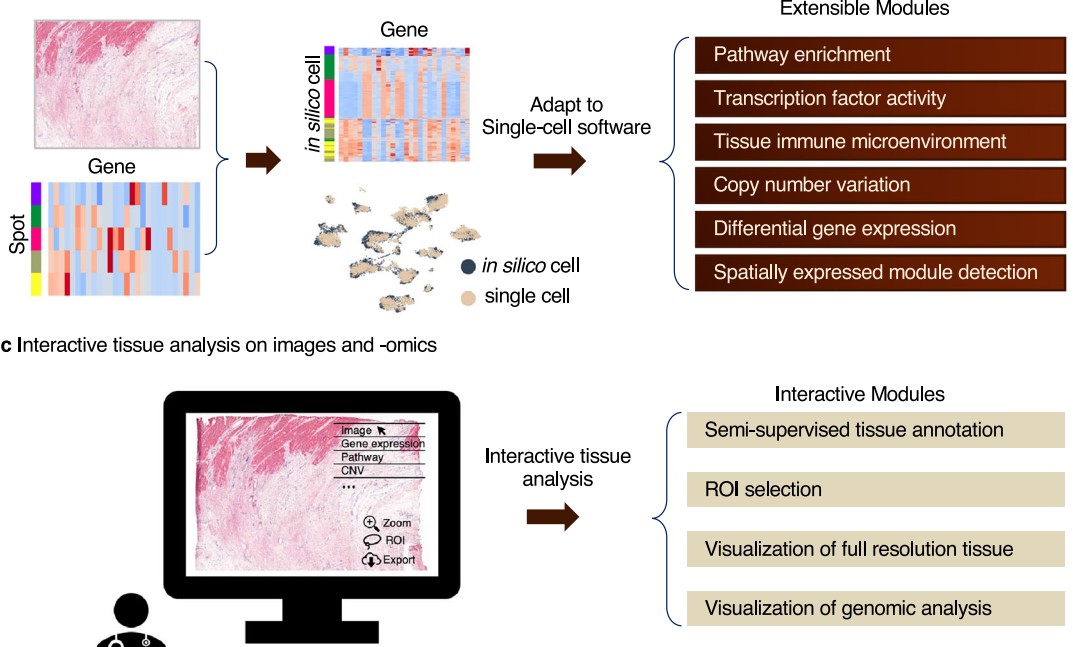

**a** Inference of *in silico* cell transcriptome

**Thor platform**

*Input*

Histology image

Visium spots

**Histology & Genomics**
PC1 min ■■■ max
Histology
Genomics
■ *MYH11*   ■ *RARRES2*

**Cell Graph** based on histology and genomics

Correct heterogenous spots with homogenous spots

**Transition matrix**

Step 1a. Cell segmentation & feature extraction

Step 1b. Spot to cell mapping

Step 2. Construct cell-cell graph in joint feature space

Step 3. Markov diffusion using transition matrix

*Output:*
Cell level spatial transcriptome

Cell-cell graph on joint features
Do not require external scRNA-seq data

**Anti-shrinking Markov Diffusion Model**

MYH11 (*Input*)    MYH11 (*Output*)    *in silico* cells

**b** Advanced analysis at single-cell resolution

Gene

*in silico* cell

Spot    Gene

Adapt to Single-cell software

● *in silico* cell
● single cell

**Extensible Modules**

Pathway enrichment

Transcription factor activity

Tissue immune microenvironment

Copy number variation

Differential gene expression

Spatially expressed module detection

**c** Interactive tissue analysis on images and -omics

Image
Gene expression
Pathway
CNV
...
⊕ Zoom
○ ROI
⊕ Export

Interactive tissue analysis

**Interactive Modules**

Semi-supervised tissue annotation

ROI selection

Visualization of full resolution tissue

Visualization of genomic analysis

**Fig. 1 | Integrated analyses of histology and transcriptomics data at the in silico cell level. a** Histological images and high-throughput sequencing data capture inherent cellular structures at different resolutions and share complementary information. The projection of the histological features to the first principal component highlights the tissue sections at cell resolution; meanwhile, expression patterns of marker genes of the cardiac smooth muscle cells (*MYH11*) and the fibroblast cells (*RARRES2*) demonstrate consistent patterns at spot resolution. The cell–cell network is constructed according to the distances in the combinatory feature space of histology (including location) and transcriptomics. In the illustration of cell–cell network, the nodes represent cells, edges represent connections, and the colors indicate cell types. Thor infers single-cell spatial transcriptome by utilizing an anti-shrinking Markov graph diffusion model. The expression profile of the marker gene *MYH11* in smooth muscle aligns with the texture of the H&E staining image, as visualized by the Mjolnir web platform. **b** Thor adapts and implements a diversity of modules for advanced single-cell analyses around the inferred spatially resolved whole transcriptome of the in silico cells. **c** The Mjolnir platform supports interactive multi-modal tissue analysis.

expression levels in cells covered by a spot (Supplementary Fig. 1a, see details in "Methods: Simulation details"). We assessed Thor's prediction accuracy by computing the normalized root mean squared error (NRMSE) between the predicted and the ground-truth gene expression values (see "Methods" for definition).

We first evaluated Thor's performance under suboptimal histology imaging conditions. Here we consider two primary issues that may impact its accuracy: (i) missed detection of cell nuclei, which commonly occurs in out-of-focus or high-density regions, and (ii) erroneous cell–cell connections resulting from poor histological features. We did not modify the spot-level gene expression profiles, thereby isolating the specific challenge posed by cell segmentation or erroneous cell–cell connections from gene expression dropouts, which would be investigated separately. The performance was evaluated on the detected cells between inferred and ground-truth gene expression. Under ideal conditions with neither cell dropouts nor randomized connections, Thor's predicted gene expression closely matched the ground truth, yielding a median NRMSE of 0.07 (Supplementary Figs. 1a and 2). Introducing random "missouts" of cells (0–40%) led to a slight increase in median NRMSE from 0.07 to 0.075 (Supplementary Fig. 1b), while introducing randomized connections in 30–40% of cells modestly increased the median NRMSE to 0.08 (Supplementary Fig. 1b). To further highlight the advantages of our algorithm, we compared Thor against two baseline methods: (a) nearest spot method, assigning gene expression based on the nearest spot; (b) BayesSpace, assigning gene expression based on local spatial neighborhoods of sub-spots[14]. These findings suggest that Thor maintains robust prediction accuracy in the presence of substantial missing cells and disrupted cell connections, outperforming those baseline methods.

Next, we examined the spatial resolution, a critical factor in spatial technologies ranging from subcellular scales to ~100 μm. Larger spots lead to greater cell heterogeneity within each spot (Supplementary Fig. 3). When we varied the spot diameter from 25 to 150 μm, Thor accurately predicted single-cell gene expression for spots up to ~100 μm in diameter, although the median NRMSE increased to 0.08 at 150 μm. We further compared Thor with three baseline methods: (a) nearest spot method; (b) k-nearest neighbors (KNN) smoothing method, assigning gene expression by averaging over the nearest twenty cells; and (c) BayesSpace. At 25 μm, both the nearest spot method and Thor exhibited high accuracy (median NRMSE 0.06). The nearest spot method's performance declined sharply as spot size increased beyond 25 μm, while Thor remained accurate with the spot size up to 100 μm. This suggests that Thor's superior performance is not solely due to incorporating nucleus segmentation. By contrast, both the KNN smoothing method and BayesSpace performed poorly across all spot sizes, with median NRMSE values of ~0.2 (Supplementary Fig. 1c). The KNN smoothing method consistently underperformed, underscoring the benefits of Thor's shared nearest neighbors (SNN) cell–cell graph and feature-preserving Markov diffusion approach.

To quantitatively evaluate Thor's performance under increasing spot complexity, we plotted the mean absolute error (MAE) of each cell against the Shannon entropy of cell type proportions. As spot heterogeneity increased, the MAE for the nearest spot method rose sharply; meanwhile, Thor accurately imputed gene expression for both low (Supplementary Fig. 3c, "A") and high (Supplementary Fig. 3c, "B", "C") heterogeneity spots. Although a subset of cells in highly heterogeneous spots showed a slight increase in MAE (Supplementary Fig. 3c, "C"), Thor's error remained much lower than that of the nearest spot method.

Finally, to evaluate Thor's imputation performance under varying dropout levels, an important challenge in high-resolution ST, we simulated 15 conditions with dropout ratios ranging from 5% to 60% and categorized them into three regimes: low dropout (<15%), moderate dropout (15–40%), and high dropout (>40%). We then measured cluster separations in principal component analysis (PCA) space using silhouette coefficients. As shown in the PCA plots (Supplementary Fig. 1d), introducing dropouts severely diminished cluster separations in the ground truth data, with silhouette coefficients reduced from 0.8 to near 0. In contrast, Thor-imputed data maintained the silhouette coefficient to 0.7–0.8 in the low-dropout regime, outperforming the KNN smoothing method and BayesSpace. When dropout ratios rose to the moderate regime, where the ground truth data's silhouette coefficients declined to 0.1–0.4, Thor-imputed data recovered the cluster separation successfully (silhouette coefficients 0.5–0.6). Even under high-dropout conditions (>40%), Thor's scores remained substantially above those of KNN smoothing and BayesSpace.

Collectively, these analyses highlight Thor's accuracy and robustness in various conditions, including missing cells, disrupted cell connections, varying spot sizes, and technical dropouts.

## Thor infers accurate gene expression at single-cell resolution

Next, we evaluated Thor on a mouse brain receptor map data acquired by MERFISH. The MERFISH data comprised 483 RNA targets from individual cells (Supplementary Fig. 4a). We simulated Visium-like data within the hippocampus region by creating a grid of evenly spaced "ST spots". The RNA molecule counts in a synthetic spot were aggregated over the cells covered by the "ST spot". These synthetic spots contained a mixture of cells of different cell types, particularly within the hippocampal subregions CA1/2/3 and the dentate gyrus (DG; Supplementary Fig. 4a). Thor connected cells of the same cell types by proximity in the morphological feature space and the spatial space, as illustrated by the cell–cell network in CA1 and DG (Supplementary Fig. 5a; note the cell type information was not provided to Thor). Thor successfully predicted cell-level gene expression in these heterogeneous regions evidenced by the profiles of selected marker genes (Supplementary Figs. 4b and 5b). For instance, Thor recovered *Adra1d* expression in CA1 and DG, which was missing in the spot-level data and the BayesSpace result. Furthermore, to gain a global view of the similarity between the in silico cells and the MERFISH cells, we projected the high-dimensional gene expression matrices to a joint uniform manifold approximation and projection (UMAP) embedding. The in silico cells inferred by Thor seamlessly mixed with the MERFISH cells on UMAP, and the distribution of cell type clusters of the in silico cells matched the ground-truth cell types (Supplementary Fig. 4c). As a baseline, mixtures of cell types were aggregated in the spot-level data, resulting in a low silhouette coefficient and Calinski-Harabasz index when mapped to the nearest cells. Thor substantially improved the cell type separation, achieving a silhouette score of 0.45 and a high Calinski-Harabasz index of 10,000, and outperformed BayesSpace by a large margin (Supplementary Fig. 4d).

We further applied Thor to a Visium dataset of human breast cancer tissue and compared the result against a Xenium reference dataset of the adjacent tissue section[28]. Using transcriptome data from the Visium dataset and the post-Xenium H&E image as input, Thor successfully inferred in silico cell-level gene expression. Visually, the spatial patterns of gene expression align closely with Xenium data (Fig. 2a). To gain a global view, we clustered the in silico cell-level gene expression using conventional scRNA-seq clustering. The same major cell types were identified from the in silico cells as from the Xenium data, evidenced by the spatial distribution of the cell types and the mean expression heatmaps of the marker genes (Fig. 2b). Additionally, integrating the predicted in silico cells with the Xenium cells showed that cells from the same cell types colocalize from both datasets (Supplementary Fig. 6), indicating Thor's ability to predict accurate and biologically meaningful cell-level gene expressions.

For a quantitative evaluation, we benchmarked Thor with three other methods of enhancing ST to near-cell resolution[14,15,17]. The spatial

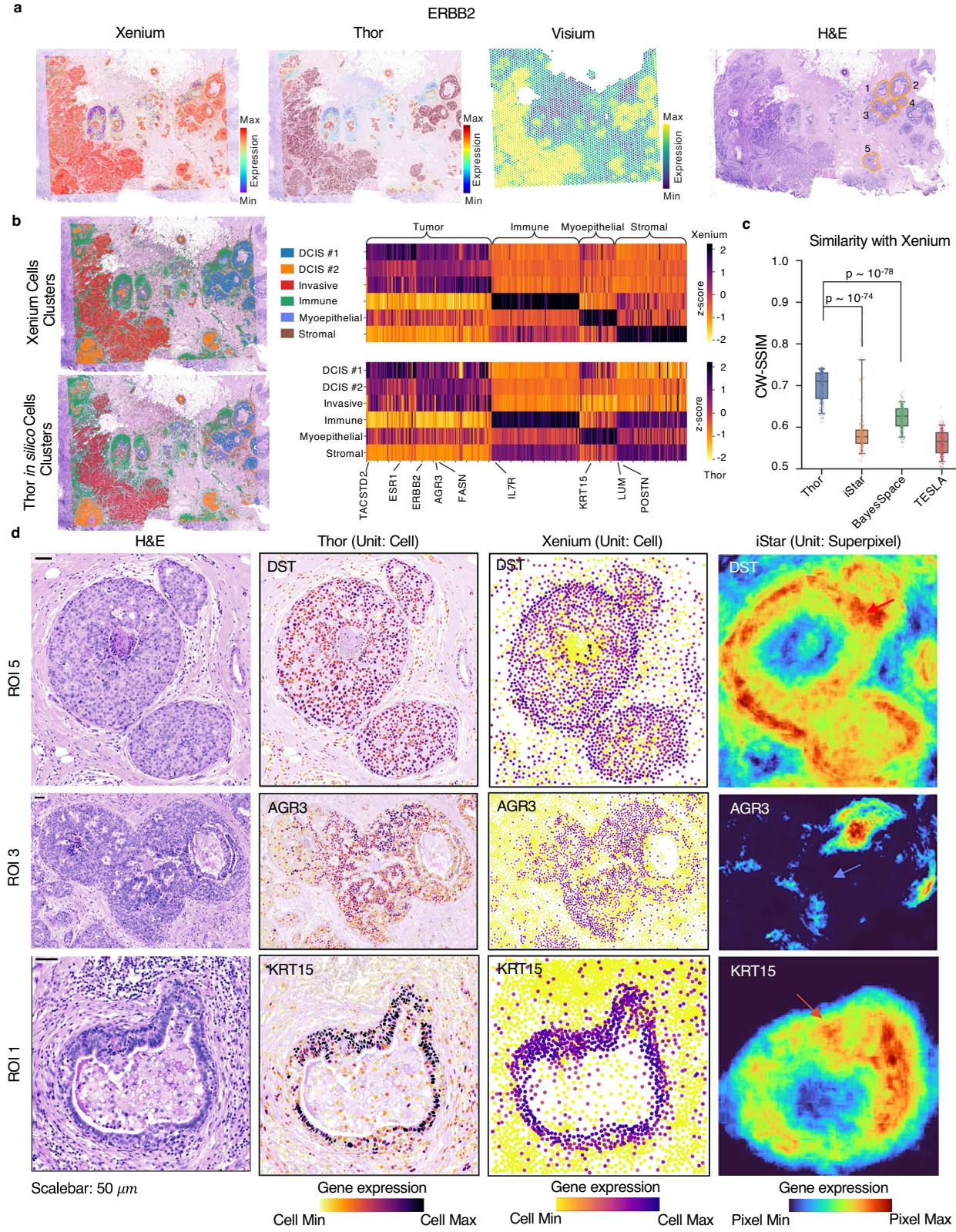

units vary among those tools (Thor: cell, iStar: superpixel, BayesSpace: subspot, and TESLA: superpixel), therefore, we calculated both image-centric and cell-centric metrics to provide a more complete evaluation. On the one hand, by converting spatial profiles of gene expression data into images, we compared the similarities between the predicted spatial patterns with the Xenium spatial patterns using the metrics structural similarity index measure (SSIM) and root mean squared error (RMSE) of pixel values. On the other hand, by mapping the pixel expression data to the cells using the nearest neighbors approach, we compared the deviations between the resulting cell-level gene expression with the Xenium data using cell-wise RMSE as an additional metric. Thor achieved the highest similarity with the Xenium data on all the metrics (Figs. 2c and S7). When using the cell-wise RMSE, the general trend remains, yet the difference between the four methods

**Fig. 2 | Thor accurately predicts single-cell spatial gene expression in human breast cancer. a** Spatial gene expression of in silico cells inferred from the Visium data and the H&E staining image of a breast cancer tissue by Thor align closely with Xenium data from the adjacent tissue section. The numbers on the H&E staining image mark DCIS regions of interest. **b** Thor-inferred spatial transcriptome of in silico cells demonstrates consistent cell clusters with Xenium using scRNA-seq clustering. The cluster annotations were adapted from the original study of the dataset[28]. The mean expression levels (normalized) of differentially expressed genes in each cluster were visualized using heatmaps. **c** Thor outperforms other methods in spatial gene expression prediction. The box plots summarize the similarity across 306 genes included in the Xenium panel. The middle line in the box plot, median; box boundary, interquartile range; whiskers, 5–95 percentile; minimum and maximum, not indicated in the box plot. One-sided Mann–Whitney U tests are used to compare Thor with the two next-best-performing tools; corresponding p-values are shown in the plot. **d** Spatial expression profiles of representative genes at the region of interest level are compared between Thor, iStar, and Xenium. Thor-inferred spatial gene expression closely aligns with the Xenium data, while iStar introduces artifacts at segment boundaries (the red arrows) and in regions with sparse cells (the blue arrow). Source data are provided in a Source Data file.

became less prominent. This is likely because all the gene expression levels, including Thor, needed to be mapped to the common cell positions (Xenium cells) using nearest neighbors before calculating cell-wise RMSE, which might have smoothed out some intricate details in the spatial pattern, as seen in Supplementary Fig. 7c, d. Overall, Thor demonstrated significantly better agreement with the Xenium data.

To gain more insights into Thor's unique advantage, we compared the expression profiles of representative genes with second best performing tool, iStar. Thor and iStar enhanced spatial resolution to (near) cell resolution, iStar at times introduced artifacts, including excessive fusion, for instance, at segment boundaries (Fig. 2d, red arrows), and in regions with sparse cells (Fig. 2d, blue arrow). For example, the spatial expression of myoepithelial marker *DST* inferred by Thor accurately outlined the boundaries of three DCIS regions in ROI 5 (Fig. 2d), as confirmed by the Xenium data and the H&E staining image. While Thor did not maintain the spatial gradient pattern due to misdetection of flat nuclei around certain region boundaries, iStar introduced excessive fusion in the tumor regions, as indicated by the red arrows in Fig. 2d. Additional examples are provided in Supplementary Figs. 8 and 9. These artifacts are likely due to that iStar predicts the expression of super-pixel patches of the WSI, rather than a cell. This approach may result in the omission of valuable cellular morphology information. In contrast, Thor takes a fundamentally different approach by considering a cell as the minimum biological unit and can accurately infer single-cell gene expression via a cell–cell network constructed from the transcriptomics and histology data.

### Thor unveils refined tissue structure in mouse olfactory bulb

We extended our evaluation of Thor-inferred gene expression levels on a MOB dataset collected by Visium. We compared the inferred molecular patterns with those acquired from high-resolution techniques, including the ISH images[33] and Stereo-seq data[27]. Results showed the spatial patterns of gene expression levels inferred by Thor aligned well with both data (Supplementary Figs. 10a and 11). For example, *Eomes* is a marker gene of cells in the glomerular layer and mitral layer[34], as observed in the ISH and Stereo-seq data. However, due to the limited spatial resolution, the spot-level Visium data failed to adequately capture the pattern in the mitral layer and exhibited discontinuities in the glomerular layer. By integrating the high-resolution H&E image with the spot-resolution ST, Thor recovered the spatial patterns marked by *Eomes* in glomerular and mitral layers (Supplementary Fig. 10a). Detailed gene expression profiles from Thor, ISH, Stereo-seq, and Visium were provided for comparison in Supplementary Fig. 11.

At the whole-transcriptome level, the in silico cell clusters dissected six main layers in the MOB, the subependymal zone, two granule layers, the mitral layer, the glomerular layer, and the olfactory nerve layer (Supplementary Fig. 10b). We further applied Cell-ID[35] to infer cell types (see "Method"; Signature genes are provided in Supplementary Data 1). By integrating ST with spatial locations and histological features, Thor resolved and refined neuron subtypes. For example, Thor distinguished granule cells (GCs) between GC-1 and GC-2 subtypes, with GC-1 concentrated in the internal plexiform layer and GC-2 predominantly in the GC layer. Additionally, Thor separated

mitral cells (M/TCs) into M/TC-1 and M/TC-2 subtypes, with M/TC-2 concentrated in the mitral layer and M/TC-1 extending into the glomerular layer. These results demonstrated Thor's capability to refine cell type classification by integrating histology and ST data.

Leveraging the cell-resolution spatial profiles, we next identified genes with spatially dependent activation patterns and coordinated gene modules using the package Hotspot[36]. The genes in the in silico cells formed 8 gene modules reflecting the primary structure of MOB (Supplementary Fig. 10c), with modules "2", "4", "7", and "8" capturing the glomerular layer, the mitral layer, the granule layers, and the olfactory nerve layers, respectively (Supplementary Fig. 10d). Remarkably, module "4" captured the thin mitral layer (thickness <40 μm), indicating successful resolution enhancement by Thor, enriching a thin layer of the M/TC-2 mitral cell subtype. The gene ontology (GO) pathway enrichment analysis of the layer-specific gene modules suggested a cascade of activities covering odor information sensory, processing, signal transmission, and memory formation in MOB layers. Together, Thor unveiled refined layers in the MOB tissue by accurately inferring cell-level gene expression data, aligning with various experimental measurements.

### Thor supports semi-supervised annotation of fibrotic regions in human myocardial infarction tissues

To better leverage the combinatory space of histological and transcriptomic features, we developed a human-in-the-loop tool for enhanced identification of tissue regions or spatial domains. The SSA tool operates within Mjolnir, enabling researchers to annotate small representative regions using marker gene expression and morphology of cells in gigapixel resolution images. These transcriptome- and morphology-guided annotations can then be quickly propagated across the entire tissue section based on Pearson correlation of the combinatory features, facilitating comprehensive tissue characterization.

We first quantitatively evaluated Thor's SSA tool using a cohort of heart tissue samples[37], which included high-resolution H&E images, high-quality ST data, and spot-level expert annotations for key tissue types in heart, including vessel, nodal tissue, adipose tissue, and fibrosis. Evaluated against spot-level expert annotations, SSA achieved accuracy ranges of 0.94–0.99 for vessels, 0.92–0.98 for nodal tissue, 0.84–0.92 for adipose tissue, and 0.92–0.93 for fibrosis (Supplementary Fig. 12). In contrast, spot-level clustering, even with optimized parameters, struggled to distinguish structures such as vessels (enriched with smooth muscle cells) from certain myocardium regions (Supplementary Fig. 13). These results suggest Thor enhances spatial tissue annotation by integrating histology with transcriptomics, surpassing spot-level clustering.

Next, we applied Thor to analyze six myocardial infarction patient samples, comprising two ischemic zones (IZ), two unaffected remote zones (RZ), and two late-stage fibrotic zones (FZ), to enable granular characterization of these distinct tissue zones in heart failure. Using the Mjolnir platform, we first defined an ROI based on fibroblast marker gene expression (*PDGFRA* and *FBLN2*) and morphological patterns in an H&E image. Thor then automatically extended the curated ROIs by identifying similar cells in the entire tissue. The

expression profiles of representative genes, including fibroblast marker genes and cardiac muscle-associated genes, displayed coherent patterns in the curated ROIs and the discovered cells (Fig. 3a and Supplementary Figs. 14–17). SSA revealed dense fibrotic areas and shallow areas which were otherwise difficult to identify manually (Fig. 3b and Supplementary Fig. 16). The resulting fractions of fibrotic areas in the six samples increased in the order of RZ, IZ, and FZ (Fig. 3c).

The precisely annotated fibrotic regions then enabled unbiased functional analysis. For each sample, we performed differential gene

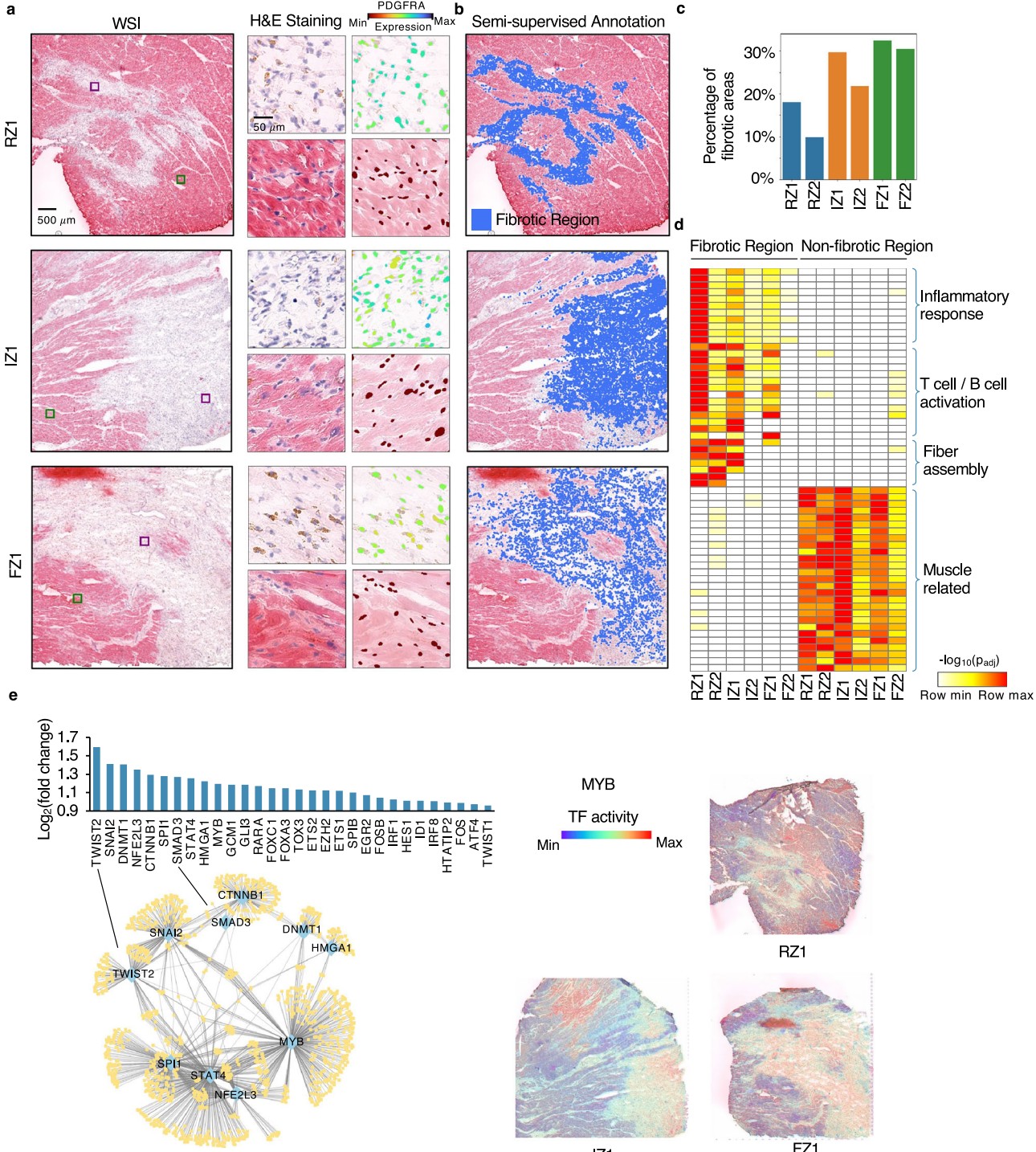

**Fig. 3 | Thor detects fibrotic regions in multiple human heart tissues with MI.** **a** H&E staining images of tissues from a remote zone (RZ1), an ischemic zone (IZ1), and a fibrotic zone (FZ1). Purple and green squares mark curated ROIs and are annotated as fibrotic and non-fibrotic regions, respectively. Close-up views of the cell morphology and inferred cellular expression of the fibroblast marker gene *PDGFRA* are provided for the curated ROIs. **b** Mjolnir-annotated fibrotic regions (blue) are visualized on the H&E staining images. **c** Bar plot of the percentages of the fibrotic regions in all six samples. **d** Heatmap of the GO pathway enrichment based on the up-regulated DEGs (fold change >2, adjusted *p*-value < 0.01 using two-sided Welch's *t*-test) in the fibrotic region compared to the non-fibrotic region in each sample, and the up-regulated DEGs (fold change >2, adjusted *p*-value < 0.01 using two-sided Welch's *t*-test) in the non-fibrotic region compared to the fibrotic region in each sample. **e** TF activity is inferred from the in silico cell spatial transcriptome. We used RTN (R package) for the transcriptional network inference and Cytoscape for network visualization. Source data are provided in a Source Data file.

expression analysis between cells in the fibrotic and non-fibrotic regions (lists of differentially expressed genes (DEGs) are provided in Supplementary Data 2), followed by GO pathway enrichment analysis. Irrespective of sample zones, the fibrotic regions showed significant enrichment of pathways such as fibroblast proliferation, stress fiber assembly, and collagen fibril organization, whereas myocardium-related pathways were enriched in non-fibrotic regions (Fig. 3a and Supplementary Fig. 18a). Interestingly, the fibrotic regions of RZ samples demonstrated more pronounced inflammation and fibrosis, likely reflecting heterogeneous progression of ischemic injury among the patient samples. After myocardial infarction, tissue in the immediate infarct area often undergoes rapid cell death and necrosis, whereas RZs may experience a delayed and prolonged inflammatory and fibrotic response[38,39]. While IZ and FZ contained the largest proportions of fibrotic regions at the whole tissue level (Fig. 3c), those findings demonstrate that functionally distinct fibrotic domains can exist outside necrotic regions.

To identify regulatory factors influencing those fibrotic regions, we estimated TF activities by utilizing a gene regulatory network database[40]. Compared to non-fibrotic regions, the most prominently activated TFs induced critical pathways, such as epithelial-mesenchymal transition (*TWIST2* and *SNAI2*) and immune response (*STAT4* and *MYB*; Fig. 3d and Supplementary Fig. 18b). The detected top-regulating TFs agreed with existing studies: *SMAD3* has been identified as a principal mediator of the fibrotic response to activate cardiac fibroblasts[41]; *SPI1* has been reported as an essential orchestrator of the pro-fibrotic gene expression program in multiple human organs[42].

Overall, Thor's SSA tool refines fibrotic tissue boundaries, highlights subtle variations in fibrotic progression, and facilitates functional insights into the molecular drivers of post-infarction cardiac fibrosis.

### Thor characterizes signature gene expression in heart failure

Spot-level STs are often inadequate to reveal intricate patterns in small or narrow regions due to limitations in spatial resolution. One such example is to identify the regenerative signatures in vascular regions. Thor allows for the exploration of gene expression in cell-resolution spatial contexts by predicting gene expression in cells detected from the histological images, thereby enhancing the ability to uncover intricate patterns.

In a patient with advanced heart failure, an LVAD is commonly implanted as a bridge to heart transplantation. Over the 6–12 months that the LVAD provides cardiac support, it has been observed that the structure and function of the heart improves to varying degrees[43,44]. Thus, we applied Thor to in-house heart tissues collected from post-LVAD implantation patients to identify genes that may be driving this regenerative remodeling. As the vasculature plays an important role in cardiac recovery[45], we prioritized our analysis in the vascular regions. Blood vessels typically consist of three layers, intima, media, and adventitia, from the lumen to the outer wall of the vessel. The media layer is mostly comprised of smooth muscle cells. In Mjolnir, based on the cell phenotypes and the expression levels of the smooth muscle marker *MYH11*, we annotated 29 and 11 vessel regions on two post-LVAD heart tissues (Fig. 4a and Supplementary Fig. 19a–c). We extracted highly expressed genes in these vessels, finding 56 genes common to both tissues (Fig. 4b and Supplementary Fig. 19d). *PLA2G2A* stood out when excluding known smooth muscle markers (such as *TAGLN*, *ACTA2*, *MYH11*, and *MYLK*). *PLA2G2A* was reported to promote cell proliferation, angiogenesis, and tissue regeneration[46] in several tumor types. *PLA2G2A* was reported to be preferentially expressed in donor heart fibroblasts in contrast to the failing heart fibroblasts[47]. We have previously found a subset of fibroblasts that are capable of transdifferentiating to endothelial cells to support microvascular recovery of an ischemic tissue[48,49]. Accordingly, we were

interested to determine if *PLA2G2A* expression represents a signature of the fibroblast subtype. We divided the vascular cells into *PLA2G2A*+ and *PLA2G2A*− groups based on the *PLA2G2A* expression levels (Fig. 4c). Notably, the *PLA2G2A*+ cells were enriched in GO pathways related to tube morphogenesis and blood vessel development (Fig. 4d). Furthermore, IF staining of tissues from these two post-LVAD patients confirmed preferential PLA2G2A protein expression in microvascular regions, which might be related to angiogenic transdifferentiation and expansion of the microvasculature (Fig. 4e). Of note, the expression of PLA2G2A may be dependent upon spatial context. We observed that in the conduit vessels, *PLA2G2A* expression was increased in vessel sections neighboring connective or adipose tissues, by comparison to those neighboring myocardium (Supplementary Fig. 19e). This difference could reflect an effect of the surrounding tissue on vascular wall gene expression. For example, perivascular adipose tissue in conduit arteries is associated with atherosclerosis, possibly due to the generation of inflammatory and angiogenic factors by the adipose tissue that may accelerate plaque neovascularization and plaque expansion[50,51]. Alternatively, differences in vascular expression of *PLA2G2A* may be due to vessel size, as it is well known that vascular gene expression changes along the course of a vessel and its branches, possibly due to developmental or hemodynamic differences in vascular segments[52,53]. Altogether, Thor's joint histology-transcriptome analysis revealed cell-resolution expression patterns of crucial molecular markers with spatial context in cardiovascular tissues.

### Thor enables multi-layered investigation of hallmarks in DCIS data

Thor offers rich layers of information through streamlined multimodal analyses within a unified platform. To showcase Thor's strengths and functions, we analyzed a well-validated DCIS dataset that has been used as benchmark widely[18,54]. DCIS is a potential precursor to invasive ductal carcinoma, a condition that can progress into a form requiring surgical intervention and radiotherapy. Understanding the heterogeneity of various DCIS regions is crucial for elucidating the factors driving their diverse behavior. The DCIS dataset comprises 18 pathologist-annotated major tumor regions (T1–T18; Fig. 5a). Histological features of segmented cells identified distinct clusters, underscoring their ability to distinguish between tissue regions (Fig. 5b and Supplementary Note 1). Through integrated histological features and ST analyses, Thor enabled a multi-layered investigation of breast cancer hallmarks.

First, Thor facilitates cell type annotation at single-cell level. The spatial distribution of annotated cell types aligned well with the results from SOTA methods such as CytoSPACE and RCTD[18,55] (Figs. 5c and S20; signature genes of each cell type are provided in Supplementary Data 3 for reference). While these methods require scRNA-seq reference data, Thor overcomes the limitation by integrating the underused histological features with ST. Additionally, Thor's advantage lies in providing gene expression for individual cells detected directly from the tissue image for additional analysis, maintaining spatial arrangement of the cells.

Second, Mjolnir enables interactive exploration of the spatial profiles of key molecules on the gigapixel histological images seamlessly at various zoom levels spanning from the whole tissue to the cellular scale. As an example, the visualization of *VEGFA*, a pivotal angiogenic factor influencing tumor growth and metastasis, highlighted distinct abundance levels within tumor subpopulations at the cellular resolution (Fig. 5d). Additional gene expression profiles at both spot and in silico cell levels were provided in Supplementary Fig. 21. A closer examination of the tumor region T1 using Thor revealed the morphological features and the nuanced expression patterns of the cancer cells. *VEGFA* exhibited the highest expression at the center of the tumor region T1, gradually decreasing in abundance towards the boundary; and was minimally expressed in the myeloid cell population outside of T1.

Third, Thor enables efficient search of similar cells in the combinatory space of histological and transcriptomic features. We curated a small set of tumor cells in T8 based on cell morphology and the key gene expression profiles. Cells in most tumor regions were

successfully identified (Fig. 5e; accuracy: 0.83). Interestingly, hardly any tumor cells in T7 matched the curated set, likely due to its distinct immune microenvironment. Instead, tumor cells in T7 were effectively identified using a separate set of curated cells within T7

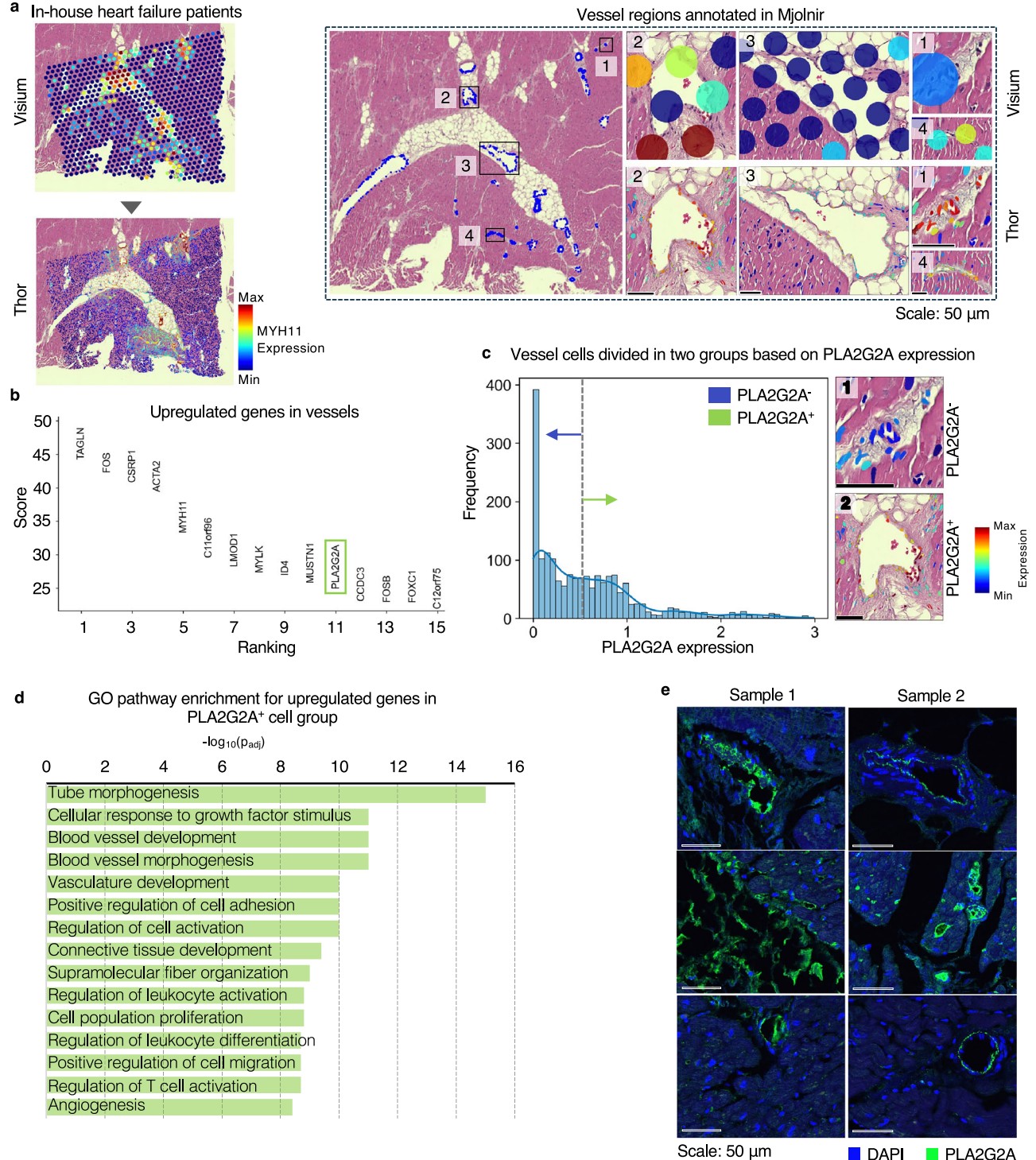

**Fig. 4 | Thor characterizes regenerative signatures in vessels in human heart failure. a** Thor infers cell-level gene expression, expression of the smooth muscle marker *MYH11* are visualized at the spot level and the cell level on sample I tissue. Utilizing Mjolnir, vessel regions are annotated. The expression levels of *MYH11* in selected vessels (labeled "1", "2", "3", "4") are recovered by Thor, where there exhibits low expression of *MYH11* at the spot resolution. **b** The upregulated genes in the vessels shared by two samples are ranked according to the gene scores. The green box marks the gene of interest. **c** Cells in the vessel regions are divided into

two groups according to *PLA2G2A* expression levels. The dotted vertical line separates *PLA2G2A*⁺ (green) and *PLA2G2A*⁻ (blue) cells at log expression = 0.5. **d** GO pathway enrichment using the top 500 upregulated DEGs (which was determined by the lowest adjusted *p*-values using two-sided Welch's *t*-test) in the *PLA2G2A*⁺ cells. **e** IF staining views of protein level *PLA2G2A* expression. The experiment was performed in two post-LVAD patient tissue samples. Source data are provided in a Source Data file.

(Supplementary Fig. 22). This demonstrates Thor's precision in identifying tumor cells through integrated analysis.

Using only the H&E image, the clustering-constrained-attention multiple-instance learning (CLAM) method[2] identified high-attention regions (Fig. 5e) that broadly overlapped with pathology-annotated tumor areas (Fig. 5a). However, CLAM also identified adipose tissue as high-attention region, which was not directly relevant to cancer (black box in Fig. 5e). These false positives happen for patterns which are not strongly represented in the negative samples[2], and may require additional training of CLAM on curated datasets of labeled WSIs for

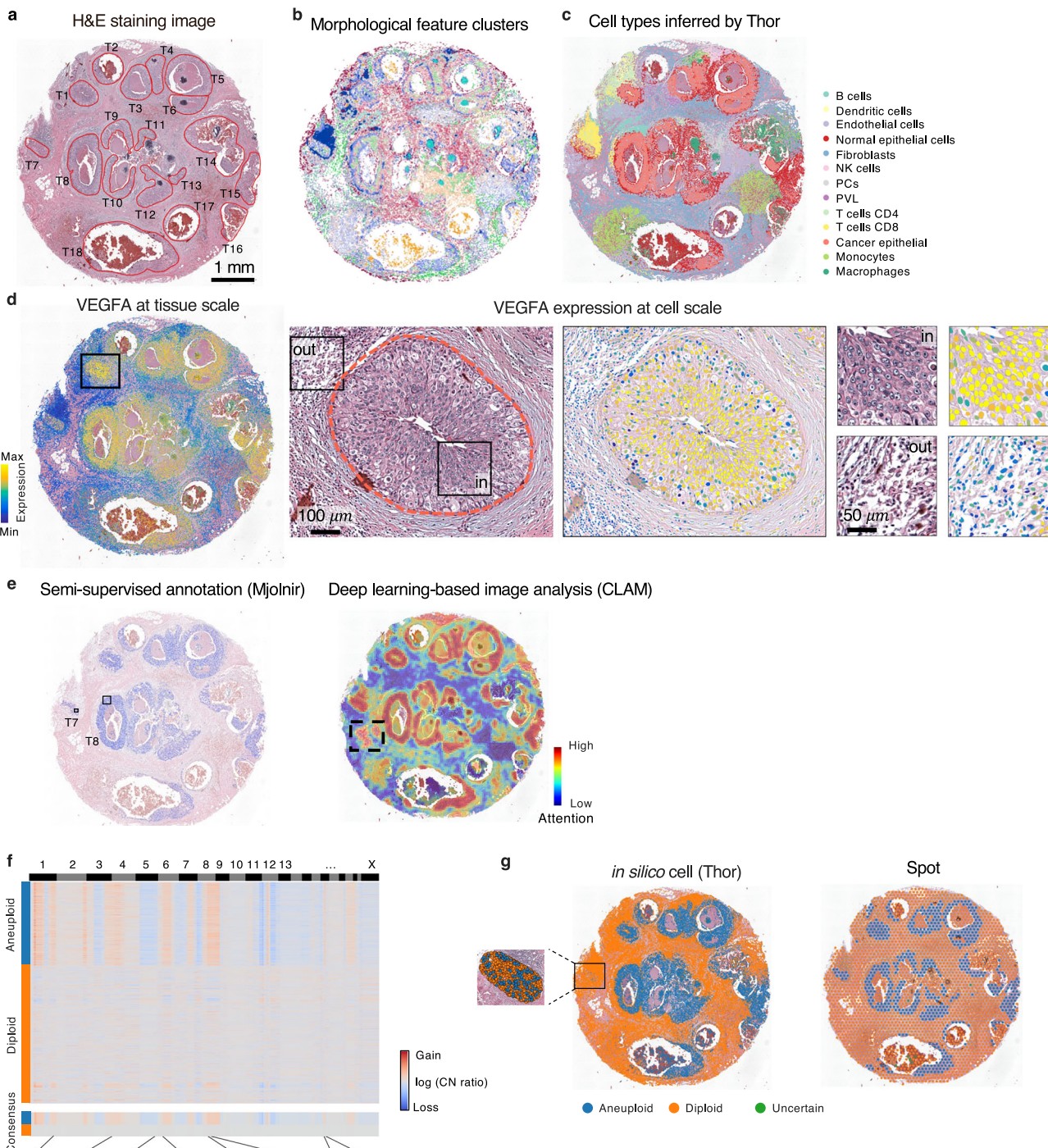

**Fig. 5 | Thor provides unbiased screening of hallmarks in cancer. a** H&E staining image of the DCIS tissue. The annotation of eighteen major tumor regions (T1–T18) in the DCIS tissue is adapted from the annotation by pathology experts (Agoko NV, Belgium). **b** Leiden clusters of the segmented cells using morphological features. The list of image features and details of Leiden clustering are provided in Supplementary Note 1. Colors represent cell clusters. **c** The spatial distribution of cell types. Cell types are obtained by Cell-ID using the Thor-inferred spatial transcriptome of the in silico cells and refined with cell type markers. **d** *VEGFA* gene expression pattern at tissue and cell scales in tumor region T1. The boxes mark regions of interest. **e** The tumor regions identified by high attention values in CLAM and semi-supervised annotation in Mjolnir. The solid black boxes mark the curated regions for semi-supervised annotation. The dotted black square marks a high-attention region where adipocytes are predominantly located. **f** Heatmap of the copy number profiles inferred by CopyKAT based on the in silico cell-level transcriptome predicted by Thor. Representative breast cancer-related genes are provided. **g** Aneuploid (tumor) and diploid (non-tumor) regions inferred by CopyKAT show consistent results between the in silico cell-level transcriptome and the spot data. Source data are provided in a Source Data file.

improved specificity. This demonstrated the value of tissue image analysis for tumor detection while highlighting the need for further multi-modal integration to reduce false positives.

Fourth, Thor's cell-level molecular signature and pathway enrichment analysis provided deeper insights into the heterogeneity of tumor progression. By examining spatial patterns of oncogenes and tumor suppressors, we observed a marked contrast between *ERBB2* (also known as *HER2*; an oncogene) and *ATM* (a tumor suppressor)[56]: *ERBB2* was highly expressed across all tumor regions, whereas *ATM* was upregulated exclusively in region T7 (Supplementary Fig. 23). An unbiased investigation of cancer hallmark pathways further high-lighted their complexity across different tumor regions at the cell level, including DNA repair, a crucial process for maintaining DNA integrity and preventing mutations (Supplementary Fig. 24). Notably, despite the low expression of *ESR1* (Supplementary Fig. 22), the estrogen response pathway still showed significant enrichment in tumor regions (Supplementary Fig. 24), emphasizing the power of pathway-based analyses to refine breast cancer classification.

Lastly, genomic CNV inference from Thor's cell-level tran-scriptome classified tumor and normal cells in DCIS. Thor uncovered genome-wide CNV profiles (Fig. 5f) and, when evaluating the dis-tribution of aneuploid cells, achieved an F1 score of 0.78 and a Jaccard index of 0.64 (Fig. 5g), aligning closely with pathology-annotated tumor regions and surpassed spot-level CNV analyses (F1 score: 0.73; Jaccard index: 0.58; Fig. 5g). Unlike spot-level CNV, which averages all cells within a spot, and can misrepresent regions containing both aneuploid and diploid cells, Thor's single-cell approach accurately detected mixed populations, as seen in tumor region T7. While spot-level analysis labeled the entire region as aneuploid, Thor-inferred and CytoSPACE-mapped single-cell data identified a mixture of aneuploid and diploid cells. These findings were further supported by external CNV profiles, validated through paired WGS data (Supplementary Fig. 25; see "Methods"). Moreover, Thor revealed key copy number aberrations across all tumor cells, including gains in 1, 2q, 8q, 12p, and 18p and losses in 5, 8p, 11q, and 12q. These aberrations highlighted well-known breast cancer-associated genes, such as *MDM4*, *ZNF595*, *FGFR4*, *HIST1H1B*, *TPD52*, *DECR1*, *GRB7*, and *JUP*[57]. CNV analyses pro-vide critical insights into the genomic alterations that underpin tumor heterogeneity and progression, offering potential biomarkers for prognosis and therapeutic targets. Altogether, as a unified platform of integrated analyses of histology and transcriptomics data, Thor offers an unbiased, multi-layered view of breast cancer hallmarks.

## Thor reveals heterogeneity of immune responses in tumor regions of DCIS

We further investigated cell-level immune responses in DCIS to quantitatively capture local immune activity around the tumor regions[57,58]. The tumor regions were ranked based on the median scores calculated from the expression levels of immune marker genes (see "Methods"). Regions T7, T1, and T17 exhibited the highest scores, indicative of robust immune activity (Fig. 6a, b).

To gain deeper insight into the molecular differences among these high- and low-scoring tumor regions, we performed differential gene expression analyses. Several immune-related genes exhibited marked variation: for example, *CD84* and *SMAD3* were abundant in T7 but almost undetectable in T15 (Fig. 6c and Supplementary Fig. 26a), whereas *KANK1*, often relevant in cancer prognosis, was highly expressed in T6 and T15 but absent in T7. We further examined func-tional distinctions and interactions between each tumor region and its immediate peritumoral neighbors (Supplementary Fig. 26b). T7 was enriched in pathways related to immune responses and T cell co-sti-mulation, whereas T15 showed enrichment for tumor-associated pathways, including hypoxia response and cell adhesion. Finally, an unbiased GO pathway enrichment analysis of genes upregulated in each tumor region highlighted Thor's ability to reveal immune-

response heterogeneity in DCIS. A global heatmap (Fig. 6d) showed that T7, T1, T4, and T14 were strongly enriched for inflammatory and immune pathways. Notably, high-scoring areas like T7 and T1 were enriched in pathways involving B cell activation, pointing to a more robust immune microenvironment with potential therapeutic relevance.

By mapping these immune landscapes at single-cell resolution, Thor elucidated functional heterogeneity among tumor regions, thereby refining our understanding of immune-tumor interactions in DCIS.

## Thor enhances gene expression signals in high-resolution Vis-ium HD data

Recent advances in ST technologies, such as Visium HD, offer cellular or even subcellular resolution. However, these high-resolution plat-forms still face technical challenges, including substantial dropout, transcript diffusion, and high background noise. To demonstrate Thor's effectiveness under these conditions, we generated a high-resolution dataset from an in-house bladder cancer sample using Vis-ium HD, which provides spatial resolution of up to $2\,\mu m$ square bins (aggregated into $8\,\mu m$ square bins for analyses, per 10x Genomics recommendations). In the Visium HD raw data, we observed high noise levels: for example, *PTPRC* (a lymphoid marker) was sparsely expres-sed in immune-rich areas, while *SPINK1* (a urothelium-associated gene) was erroneously detected in non-tissue regions (Supplementary Fig. 26a).

We first applied Thor to infer cell-level expression profile from the $2\,\mu m$ bin-level inputs by integrating ST with histology. Thor's cell-level imputation yielded more coherent expression patterns than $8\,\mu m$ square bins. Thor correctly localized *PTPRC* to immune areas and *SPINK1* to the tumor boundary, aligning with pathology annotations. Beyond single-gene assessments, Thor-imputed data captured distinct cell populations more accurately. For instance, cluster 7 in Thor's results precisely matched the pathology-annotated immune cell regions, whereas the raw bin-level data overestimated immune cell presence (Supplementary Fig. 26b). Similar overestimation of certain cell types was also reported recently in Visium HD data[59].

To contextualize Thor's performance, we further compared it with Bin2Cell which assigns bin-level expression to cells based on overlaps between transcriptomic bins and expanded nucleus masks[59]. Both methods successfully reduced expression artifacts in non-tissue regions and accurately localized *SPINK1* to epithelial compartments (Supplementary Fig. 26). However, Bin2Cell left approximately half of the detected cells without assigned transcripts, whereas Thor inferred expression profiles for a substantially larger cell population (by 37%). This difference may reflect Thor's use of histology-derived features to inform expression patterns beyond direct bin overlap, and Bin2Cell's reliance on bin-to-nucleus mapping can lead to under-assignment in dense or ambiguous regions.

While these proof-of-concept analyses demonstrate Thor's pro-mise for refining gene expression signals and enhancing biological interpretability in high-resolution ST datasets, we note that current analyses remain preliminary. Future work will involve systematic benchmarking across tissue types, staining protocols, and segmenta-tion pipelines to further validate and refine Thor's performance.

## Robustness of Thor to parameter settings

Thor is designed to be highly flexible, allowing customization of var-ious parameters that control the preprocessing of image/tran-scriptome data, cell–cell graph construction, and the Markov diffusion process. To evaluate Thor's robustness, we conducted a systematic sensitivity analysis of key parameters, including the diffusion step size $t$, the number of cell neighbors $k$, and the number of principal com-ponents $nPC$ of the transcriptome data. Thor constructs a SNN cell–cell graph based on the KNN in the combinatory space. First, we

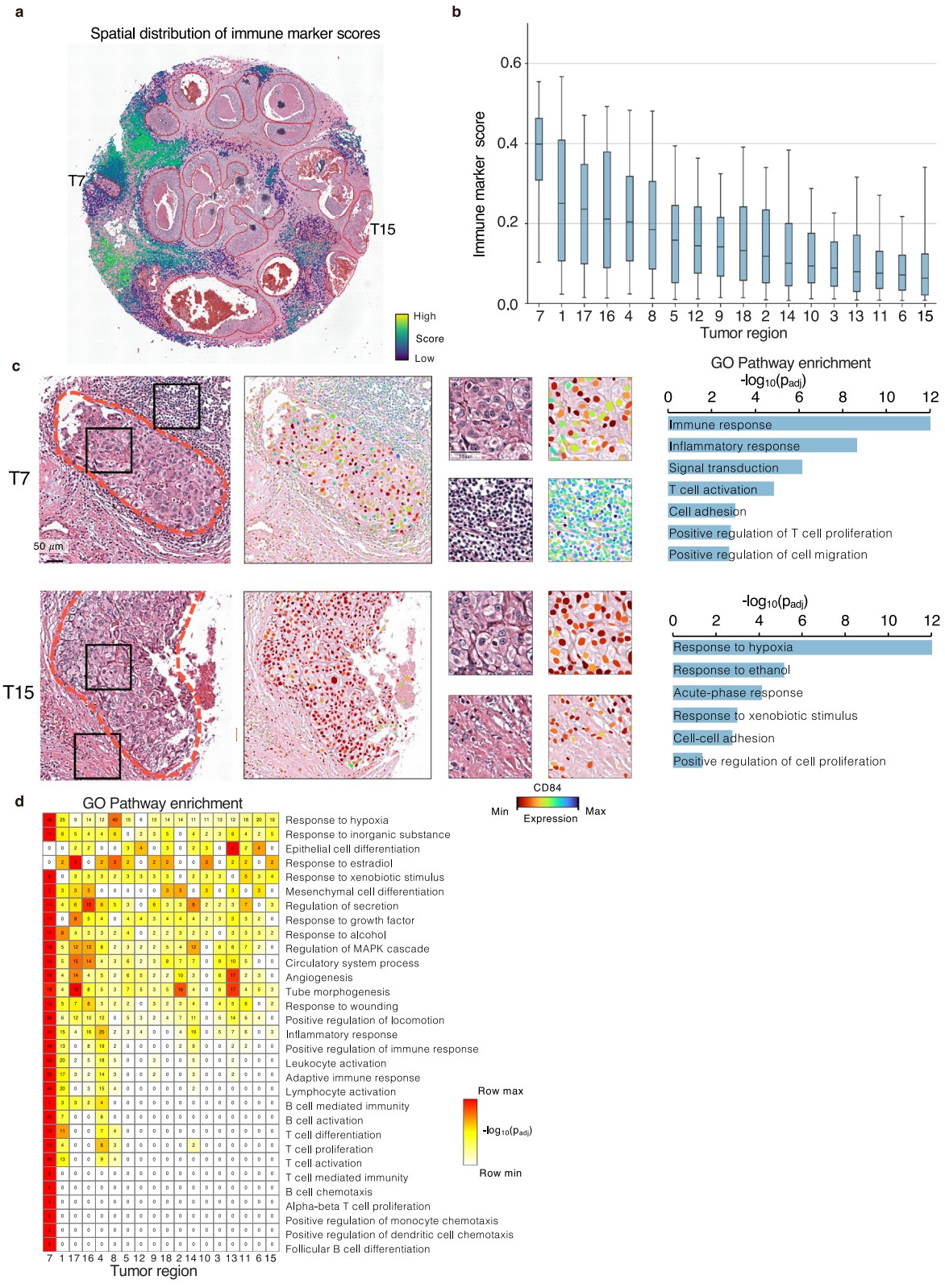

tested a range of $k$ values on the MOB dataset while keeping other parameters fixed ($t = 40$ and $nPC = 10$). To reduce bias from highly expressed genes, we applied $z$-score normalization for each gene. We then calculated the Pearson correlation coefficients ($r$) across each pair of $k$ settings. Thor demonstrated strong robustness for $k$ values between 4 and 10, with a mean $r = 0.88$ and standard deviation (std) = 0.09. However, very small $k$ values (<3) may produce disconnected cell graphs, whereas very large $k$ values (40–100) may lead to over-smoothing and weaker correlations with the results of other $k$ values (mean $r = 0.56$, std = 0.27). Second, we evaluated the impact of varying $nPC$ values while fixing $k = 5$ and $t = 40$. As shown in Supplementary Fig. 27a, Thor remains highly robust when $nPC >= 8$ (mean $r = 0.94$, std = 0.05). In contrast, $nPC < 4$ fails to capture sufficient complexity in the data, leading to lower correlations with high $nPC$ values. Third, we also

**Fig. 6 | Thor reveals mechanistic insights into the immune response of DCIS.**
**a** Spatial distribution of the cell-level scores based on 29 genes. The red lines encircle tumor regions from pathology annotation. **b** The tumor regions are ranked according to the median cell-level score. Source data are provided in a Source Data file. The middle line in the box plot, median; box boundary, interquartile range; whiskers, 5–95 percentile; minimum and maximum, not indicated in the box plot. **c** Zoom-in view of the tumor regions with highest/lowest (T7/T15) scores. The dotted orange lines encircle tumor regions. The expression level of one DEG, *CD84*,

is visualized in the inner and perimetral parts of the tumor regions (marked by the dotted black boxes). GO pathway enrichment is based on 300 up-regulated (fold change >2, adjusted *p*-value < 0.01 using two-sided Welch's *t*-test) and 300 down-regulated (fold change <0.5, adjusted *p*-value < 0.01 using two-sided Welch's *t*-test) DEGs between T7 and T15. **d** Heatmap of the GO pathway enrichment based on the up-regulated DEGs in each tumor region compared to the rest (fold change >1.5, adjusted *p*-value < 0.05 using two-sided Welch's *t*-test). Source data are provided in a Source Data file.

evaluated a range of diffusion time $t$ while keeping $nPC = 10$ and $k = 5$ fixed. Thor converged after ~10 diffusion steps, achieving a mean $r = 0.90$ (std = 0.10) for $t = 10$. However, large $t$ values (e.g., $t > 50$) may notably increase run time without significant performance gains (Supplementary Fig. 27b). Overall, our analyses show that Thor is robust to a broad range of $t$, $k$, and $nPC$ values. These findings indicate that minor adjustments within reasonable parameter ranges have minimal effect on Thor's results, which justifies keeping a common set of parameters across all case studies.

Moreover, variational autoencoder (VAE) is widely used for RNA-seq data analysis[60–62]. Thor can utilize the latent representation in VAE for faster predictions. In the fast mode, the Markov diffusion is conducted on the VAE latent embeddings. The hyperparameter tuning, such as adjusting the input and latent dimensions of VAE can affect the results of Thor and contributes to generalizability. The input dimension should depend on the genes of interest, such as highly variable genes or spatially variable genes. Moreover, a proper latent dimension should sufficiently capture the biological complexity in the data. For instance, a latent dimension of 10 is set by default in scvi-tools[61], with 20 or 30 being appropriate for more complex scRNA-seq datasets. We evaluated Thor's performance on the MOB dataset by varying the latent dimensions in separate VAE models (8, 16, 20, 32, 64, and 128), while keeping other parameters fixed ($nPC = 10$, $k = 5$, and $t = 40$). Thor-predicted gene expressions remained highly consistent with Pearson's $r > 0.85$ across all settings (Supplementary Fig. 27c). These results indicate that Thor is robust to a broad range of parameter settings.

## Discussion

Thor is an extensible and customizable platform detailed in the following aspects. First, the cell-level ST broadens the spectrum of downstream analyses to those originally designed for scRNA-seq data. Outputs from Thor interoperates with libraries such as Squidpy[24] and stLearn[25], and can be easily adapted for use with scRNA-seq tools. Thor has included submodules such as cell-specific pathway enrichment[63], inference of genomic CNV profiles[64], and ligand-receptor analysis[65]. Second, Thor supports customized cell features for building the cell–cell network. In this work, we highlight Thor's performance using intensity-based morphological features such as color intensities of the staining image patches. The inclusion of more task-relevant features elevates the quality of the cell–cell network. For example, research has shown that spatial cellular graphs built from multiplexed IF data enable better modeling of disease-relevant microenvironments[66]. In addition, Thor supports direct input of a cell–cell network adjacency matrix. Last, beyond ST, emerging omics technologies such as spatial metabolomics and proteomics are increasingly adopted to capture local metabolic or protein-level processes that underlie key tissue functions and disease mechanisms. While our current work focuses on applying Thor to ST, we envision that its underlying framework, which constructs a cell–cell graph from spot-level data, cell coordinates, and histological features, and then refines those data through graph diffusion, could be adapted for spatial metabolomics or proteomics as well. By substituting transcriptomic values with metabolomic or proteomic intensities, Thor could enable a more comprehensive, multi-omics view of tissue biology at single-cell resolution. We anticipate that future developments will provide deeper insights into complex

tissue characterization by integrating these additional modalities. In heart tissues from post-LVAD patients, we observed *PLA2G2A* expression in vessels embedded in myocardium as well as in those adjacent to connective or adipose tissues. While expression in the former may support beneficial regeneration[47], expression in conduit vessels near pericoronary adipose tissue may instead reflect pathological vascular inflammation linked to plaque formation and atherosclerosis[67]. These observations highlight the value of Thor's detailed spatial resolution in facilitating spatially aware analysis and supporting further investigation into spatially regulated processes in cardiovascular biology.

Thor integrates histological and transcriptomic features by inferring cell-level ST. Notably, Thor does not require any additional scRNA-seq data as a reference. This not only reduces the sequencing cost but is also practically advantageous in FFPE tissues. FFPE tissues serve as the most abundant specimens for longitudinal studies with preserved tissue morphological details, yet RNA-seq profiling encounters hurdles due to RNA crosslinking, modifications, and degradation. The Visium platform offers a solution for profiling mRNA levels in both fresh-frozen and FFPE tissues, employing a de-crosslinking process[68]. Nevertheless, it falls short of providing cellular-level resolution. In contrast, commonly used methods like chromogenic immunohistochemistry (IHC) for assessing in situ biomarker expression in FFPE tissues are limited by the number of analytes, non-linear staining intensity, and the subjective nature of quantitative analysis[69]. Thor strategically leverages the advantages of Visium and overcomes these challenges by delivering cell-level whole-transcriptome analysis, reducing cost and workload.

To better leverage the combinatory space of histological and transcriptomic features, we developed a human-in-the-loop tool SSA for targeted exploration. In Mjolnir, researchers select small representative regions guided by marker gene expression and cellular morphology; SSA then searches the joint feature space of geometry, histology, and transcriptomics to retrieve cells with matching profiles. Unlike unsupervised, global tissue segmentation tools which uncover large-scale tissue architecture without user input[17,70], SSA focuses on high-resolution, localized annotation. It is powered by Thor-inferred gene expression and histological features extracted from the high-resolution histology image, supporting sub-regional or sub-cell-type level search and labeling. Rather than replacing global tissue segmentation methods, SSA complements them by offering precise, user-guided analysis within complex tissues.

Thor offers several advantages over existing frameworks for studying histological structures. PROST uses spatial relationships and transcriptomics data to identify spatially variable genes and to cluster spatial domains, but it does not enhance the resolution of the original ST data[21]. Thus, with Visium data, PROST operates at the Visium-spot level and does not utilize histology images. By contrast, Thor integrates cell-level features from histology images with spot-level transcriptomics, enabling inference of gene expression at the single-cell level and providing a more granular analysis of histological structures from Visium data. METI, meanwhile, is an end-to-end framework tailored to cancer ST data, mapping tumor cells and the surrounding microenvironment primarily in oncology-focused contexts[22]. Thor, on the other hand, was conceived as a more generalizable approach applicable across various tissues, disease states, and organisms. Moreover, neither PROST nor METI directly outputs single-cell gene

expression. In contrast, Thor integrates histological and transcriptomic data in a task-agnostic manner to infer spatially resolved single-cell gene expression. This capability supports a broad range of downstream analyses and comes bundled with extensive analytical modules, including pathway enrichment, spatial gene module identification, differential gene expression, TF activity estimation, and interactive whole-slide data visualization. Taken together, these features allow Thor to complement and extend the capabilities of these frameworks by offering deeper spatial and molecular insights into tissue architecture.

Several cutting-edge tools released recently have advanced the analysis of spatial-omics data[70–72] in directions complementary with Thor's aims. Spotiphy, for example, generates cell-level gene expression from spot-level data by combining matched scRNA-seq data with cell locations segmented from histology images. In contrast, Thor achieves single-cell-level expression without external references by constructing a cell–cell graph that incorporates both spatial coordinates and rich histology-based features and refining expression via Markov diffusion on the graph. MISO, on the other hand, integrates spot-level omics features of multiple modalities into a single framework to achieve biologically relevant tissue domain segmentation. MISO operates at spot or patch level, whereas Thor focuses on single cell-level analyses on histology and transcriptomics. In addition, unlike deep learning-based tools that require extensive training data and advanced hardware, Thor functions as a light-weight model-based computational method that runs efficiently on a laptop. Thor presents a unified platform for streamlined tissue and molecular analyses, along with interactive visualization and annotation of gigapixel histology or generated omics images, for detailed cell-resolution data exploration. Thor can be applied to emerging high-resolution spatial platforms, such as Visium HD, underscoring its scalability and versatility. Across multiple tissue types and spatial resolutions (Visium and Visium HD), as well as extensive simulations of cell-segmentation miss-outs, erroneous cell–cell connections, and transcript drop-outs, Thor maintains low normalized-error scores and shows strong concordance with external ground-truth proxies such as WGS-validated CNV profiles and expert pathology annotations. Taken together, these results position Thor as a robust computational platform for single-cell-level analysis of histology and transcriptome data.

With rapid breakthroughs in deep-learning-based computer vision algorithms[7,8,73–75], accurately detecting cell nuclei has become increasingly viable, transforming the challenge of cell detection in high-density regions[9,76]. As an integrative ST analysis platform, Thor incorporates multiple SOTA tools for cell segmentation and also supports manually or strategically added missing cells, enabling enhanced flexibility and adaptability for diverse workflows. Thor is designed to stay aligned with ongoing advancements in cell segmentation technologies, ensuring that its methods remain cutting-edge. Thor extracts tile-based image features from an image patch centered at the segmented cell nucleus centroid to capture the local environment surrounding the nucleus. These features are not limited to the nucleus itself but include the tissue context within the image tile, providing a comprehensive representation of the cell's local environment from histology. Tile-based feature extraction is a practical strategy widely adopted in histology image analysis by deep-learning models and pathology foundation models. It facilitates a wide range of downstream tasks, including cell segmentation, cell type annotation, and tumor microenvironment profiling[15,17,77–80]. We recognize that certain cell types or microstructures may require more specialized descriptors. To address this, Thor provides an API (*thor.pp.image.WholeSlideImage.load_external_cell_features*) that supports morphological features generated by external tools (e.g., CellProfiler or CellViT). Researchers can extract customized metrics, such as cell shape, texture, or intensity profiles, and then input these features into Thor, effectively augmenting or replacing the default cell

detection and tile-based features. This flexible design allows Thor to accommodate a wide spectrum of histological analyses and cellular phenotyping tasks, ensuring that users can tailor the platform to their unique research objectives.

Recent advances in ST technologies have pushed spatial resolution toward cellular or even subcellular levels[27,32,81], yet each technology still faces practical hurdles that Thor can help address. For instance, although Visium HD offers sub-cellular bin sizes, it can suffer from high dropout rates, low gene coverage[59], and imperfect bin-to-cell alignment[82]. Meanwhile, Slide-seq may provide sparse transcript detection and limited capture size[83], and image-based platforms such as Xenium and CosMX rely on predefined gene panels and may omit genes of interest. Our results show that by integrating Visium HD data with histological features, Thor reduces technical noise and reveals spatially coherent expression patterns that match pathology annotations. Beyond improving data quality, Thor functions as a comprehensive downstream analysis platform capable of handling the computational and visualization demands posed by large-scale ST datasets, where a single slide can contain millions of bins or hundreds of thousands of cells. Thor's interactive visualization tool, Mjolnir, renders gigapixel images and facilitates responsive exploration on standard computing hardware. Moreover, Thor remains a cost-effective option for achieving single-cell level analyses with standard Visium data (~70% less expensive than Visium HD), benefiting labs with resource constraints or those seeking to reanalyze legacy ST datasets.

Thor has several limitations. First, it relies on high-resolution histology images (typically $0.25–0.5\,\mu m$ per pixel) for cell detection and histological feature extraction. Real-world scenarios may introduce complexities, such as loss of focus in imaging or improper staining across large tissue regions. Such conditions may lead to missed cell detection or unrepresentative image features, and Thor's performance may understandably suffer. Especially, Thor does not predict gene expression for undetected cells; in those cases, alternative tools which do not rely on cell detection (e.g., iStar) may be more appropriate. Second, Thor exhibits reduced performance in capturing gradient patterns for some genes, likely due to challenges in detecting flat cell nuclei in peripheral areas. Incorporating higher precision imaging techniques (e.g., cell-membrane staining) or alternative analysis methods (e.g., iStar) that do not rely on cell segmentation is recommended. Third, Thor does not support multi-sample integration that handles batch effects in transcriptomics data and histological variability between tissue sections, limiting the applicability of SSA across multiple samples. Fourth, Thor does not operate at subcellular resolution to provide finer level analysis. Due to the light diffraction limits in standard histology imaging and complex morphological variability of subcellular structures, robust and accurate segmentation of individual organelles or subcellular structures is highly challenging[84] and is feasible only for certain organelles in restricted platforms[75,85,86]. Thor's cell-level integrated analyses of transcriptomics and histology may complement nanoscale spatial omics technologies. Its modular architecture could, in future work, integrate with subcellular methods like Stereo-seq to bridge tissue, cellular, and subcellular-level insights.

In conclusion, Thor advances ST analysis by integrating histology and transcriptomics, refining gene expression to the single-cell level and enabling more precise characterization of tissue architecture. This approach provides a valuable foundation for future cross-modal integration, including highly multiplexed imaging techniques (e.g., CODEX or MIBI) to achieve a more comprehensive, multi-modal understanding of spatial-omics data. By enabling the exploration of cellular interactions across spatial landscapes, Thor not only facilitates the discovery of biological insights but also lays the foundation for the development of novel therapeutic modalities, thereby advancing the field of precision medicine for more effective and personalized patient care.

## Methods

### Ethics statement

Myocardial specimens from patients with heart failure supported by LVADs prior to transplantation were collected at Houston Methodist Hospital under institutional review board (IRB), with approved protocol PRO00006097:1 ("Congestive Heart Failure"). Written informed consent was obtained from all participants. Tissue blocks from patients with muscle-invasive bladder carcinoma treated at Houston Methodist Hospital were acquired through IRB protocol PRO00037670 ("Molecular Analysis of Archived Cancer Tissues"). All human data in this study were de-identified to ensure patient privacy. All procedures adhered to institutional and federal ethical guidelines.

### Overview of Thor

Thor integrates transcriptomics and histological information by faithfully inferring the whole transcriptome of in silico cells. Thor does not require training for the inference of cell-level gene expression. Instead, it operates per slide through a four-step modularized workflow (see Supplementary Fig. 30).

(i)   Identify cells and extract locations and morphological features of each cell in their spatial neighborhood from the histological image. Meanwhile, the ST data is preprocessed, and each cell's gene expression is initialized to their nearest spots.

(ii)  Compute multi-modal distances between cells and construct the cell–cell network based on their morphological features, geometrical locations, and the transcriptome collectively.

(iii) Convert the distances to affinities using an exponential kernel, so that the similarity between two cells decreases exponentially with the multi-modal distance.

(iv)  Infer gene expression of the cells by allowing information flow between similar cells while preventing transfers from cells covered by a heterogeneous spot.

Then the predicted cell-level gene expression can be applied to perform downstream analyses, including interactive analysis in ROIs. The modules are described in detail as follows.

### Nucleus segmentation and feature extraction from histological images

Nucleus segmentation is critical in the analysis of histological images, enabling quantitative assessment of the number of nuclei, density, and morphological characteristics. Thor integrates several SOTA tools[6,7,73] for nucleus segmentation and supports user-supplied segmentation results to ensure adaptability across diverse platforms.

For jointly analyzing the histological image and the transcriptomics, Thor employs two filtering processes: Thor eliminates out-of-context nuclei by superimposing segmented nuclei on the aligned spatial spots and removing nuclei whose centers are beyond a cutoff distance from the nearest spots. The default cutoff distance is the diameter of the spots. Furthermore, Thor detects and removes isolated cells or artifacts located away from the tissue boundaries.

Tile-level histological features are extracted to represent the local environment surrounding each cell. This local environment is defined by extending from the nucleus centroid to a given distance, typically twice the mean distance between the nearest nuclei centroids. In this study, we included image features such as the mean and standard deviation of color intensities, as well as image entropy, within a defined radius around each nucleus on the tissue. These features have proven to be effective in constructing cell–cell networks for Thor inference across all tested datasets. Additionally, Thor supports custom functions for feature extraction and allows the integration of user-supplied nucleus or cell-specific features, as well as deep-learning-derived features, offering flexibility and extensibility in the analysis process. Thor is designed to incorporate advances in segmentation toolkits to stay at the forefront of cell-segmentation field.

### Constructing the cell–cell network

Thor infers cell-level gene expression based on the cell–cell network. Connectivity between cells is determined by their distances in the combinatory feature space, formed by morphological features, geometrical locations, and the low-dimensional representation of the transcriptomic data. The features are standardized to a normal distribution $N(0,1)$ across all the cells. Nearest neighbors are included to construct KNN based on the distance metric $d_{ij}$ in the feature space, i.e.

$$d_{ij} = \sqrt{\left(w^{gen\_m}d_{ij}^{gen}\right)^2 + \left(w^{geo\_m}d_{ij}^{geo}\right)^2 + \left(d_{ij}^{mor}\right)^2} \quad (1)$$

where $d_{ij}^{gen}, d_{ij}^{geo}, d_{ij}^{mor}$ are the dimension-normalized Euclidean distances in the transcriptomic (reduced dimension), geometrical, and morphological feature space, respectively; $w^{gen\_m}$ and $w^{geo\_m}$ are the respective weights relative to the morphological feature distance. Increasing the $w^{geo\_m}$ value leads to a more localized network and increasing the $w^{gen\_m}$ value favors the distance in the transcriptomic space.

Next, to preserve local structure and account for the non-uniform density of the cells, the KNN cell graph is converted to a SNN graph. SNN prioritizes connections among cells that have multiple neighbors in common. This emphasis can unveil intricate data patterns and has demonstrated a reduced susceptibility to isolated noisy data points. Cells $i$ and $j$ are connected if the proportion of their shared neighbors $w_{ij}$ is beyond a given threshold.

$$w_{ij} = card(NN(i) \cap NN(j))/k \quad (2)$$

In Eq. (2), $NN(i)$ and $NN(j)$ refer to the sets of nearest neighbors of cell $i$ and cell $j$, respectively. $card$ refers to the cardinality of the overlap set. $k$ is the number of nearest neighbors.

### Feature-preserving Markov diffusion model

As the ST spot data represents the aggregate expression across enclosed cells, we hypothesize that gene expression in a homogeneous spot is more accurate than a heterogeneous spot. The heterogeneity of a spot is quantified by the coefficient of variation in cellular features of all cells mapped to the spot or by the Shannon entropy when cell type labels are available (e.g., from spot deconvolution methods). On the SNN cell graph, Thor ensures that more accurate gene expression data corrects the less accurate ones while inhibiting propagation of the less accurate information through modulation of node weights and edge weights.

Cells mapped to a more homogeneous spot carry a larger node weight $G_i$, making them more robust to variations. $G_i$ is calculated as an exponential kernel on the heterogeneity of the corresponding spot.

$$G_i = e^{-kS_i} \quad (3)$$

where $k$ is the inverse kernel width that controls the shape and $S_i$ is the heterogeneity of the spot enclosing cell $i$. The edge weight $\epsilon_{ij}$ between two cells $i$ and $j$ is computed as the product of the "bandwidth" $w_{ij}$ as shown in Eq. (5), proportion of their shared neighbors defined in Eq. (2), and the "latency" $L_{ij}$ defined in Eq. (4).

$$L_{ij} = \frac{1}{1 + e^{-\alpha(G_i - G_j)}} \quad (4)$$

$$\epsilon_{ij} = (1 - \delta_{ij})L_{ij}w_{ij} + \delta_{ij}G_i \quad (5)$$

where $\alpha$ controls the steepness of the scaled sigmoid function.

The transition matrix $F_{ij}$ is then computed as,

$$F_{ij} = (1 - \lambda)\delta_{ij} + \lambda\,\epsilon_{ij} = \delta_{ij} - \lambda(\delta_{ij} - \epsilon_{ij}) \tag{6}$$

where the constant $1 - \lambda \in (0, 1)$ is the probability of keeping the original (self) gene expression. As shown in Eqs. (4–6), the "latency" is a key parameter that turns the symmetric SNN into an asymmetric network in favor of incoming information flow in a connection from the cell with a lower heterogeneity score, or a larger node weight, to the cell with a higher heterogeneity score.

The transition matrix takes the same form as in a Laplacian smoothing method, likewise, the diffusion causes shrinking in the transcriptome space. Therefore, we employ a well-known feature-preserving technique in the field of surface smoothing[87] and introduce a reversed diffusion transition matrix $R_{ij}$ after the forward diffusion to inflate the transcriptome space.

$$R_{ij} = (1 - \mu)\delta_{ij} + \mu\,\epsilon_{ij} = \delta_{ij} - \mu(\delta_{ij} - \epsilon_{ij}) \tag{7}$$

where $\mu \in (-1, 0)$, $\delta_{ij}$ is the Kronecker delta. In practice, the absolute value of $\mu$ is set marginally larger than $\lambda$ for sufficient inflation.

The feature-preserving diffusion is composed of a forward diffusion step followed by a reversed diffusion step. Therefore, the effective transition matrix is computed as the matrix multiplication of the reversed diffusion transition matrix $R_{ik}$ and the forward diffusion transition matrix $F_{kj}$,

$$T_{ij} = \sum_k R_{ik} F_{kj} \tag{8}$$

The resulting Markov transition matrix $F_{ij}$ represents the probability distribution of transitioning from each cell to every other cell in a single step. The transition matrix $T_{ij}$ is normalized by rows to ensure that the probabilities of incoming signals sum up to 1.

Lastly, after obtaining the Markov transition matrix, Thor performs graph diffusion to infer the gene expression at the cellular level.

$$x^n = (T)^n x^0 \tag{9}$$

where $x^0$ is the input gene expression initialized by the nearest spot-level values, $x^n$ is the final inferred gene expression, $F$ is the feature-preserving Markov transition matrix, and $n$ is the number of diffusion steps. The Markov diffusion converges rapidly, typically within 10 steps[88].

Due to the substantial number of in silico cells, the diffusion can take hours. To speed up Thor, the Markov graph diffusion may be performed on the reduced-dimensional embedding, such as the latent variables of a VAE, and the transcriptome can be reconstructed from the latent variables. Finally, Thor rescales the gene expression to the same range as the input spot-level gene expression, and optionally samples cell-level gene expression considering stochasticity in scRNA-seq reads.

### Advanced analyses and dynamic visualization

Technical challenges arise when analyzing and visualizing systems containing a vast number of cells. The WSIs are gigapixel-scale and typically encompass from 10,000 to 100,000 in silico cells within a $6.5 \times 6.5$ mm tissue sample. These large-scale datasets present substantial difficulties in terms of computational resources and effective data visualization. To address these challenges, we adapted existing pipelines for analysis of cell-level multi-omics and imaging data, as well as developed a dedicated tool Mjolnir for interactive visualization of large biomedical images. Details for dynamic visualization and advanced analyses are as follows,

- Interactive visualization of histology and genomics. Mjolnir leverages image-tiling technologies used by Google Maps,

enabling seamless navigation through gigapixel images at a range of zoom levels. Mjolnir empowers users to visualize segmented components, including spots and cells/nuclei color-coded by gene expression or additional attributes, such as copy number profiles.
- ROI selection. Mjolnir supports drawing and editing regions of any shape on the staining image. A user can export the selected ROIs in common data formats such as annData for gene expression, TIFF for image patches, and JSON for polygon coordinates, facilitating further analyses.
- DEG analysis. DEGs are extracted between two specified groups of cells. Thor treats individual cells in a group as replicates and assesses the significance of changes in gene expression using statistical models.
- Pathway enrichment analysis. A pathway is represented by a group of specific molecules that collectively carry out vital functions within cells and organisms. Thor adapts the Python package decoupler[63] to compute the cellular enrichment of pathways.
- TF activity analysis. The activity of a TF is inferred by the expression levels of its regulated genes. Thor adapts decoupler[63] to compute cellular TF activity.
- CNV analysis. Thor integrates the R package CopyKAT[64] for CNV analysis with a wrapper function. Thor expedites the calculation of CNV by parallel computing.
- TLS score. The TLS score is calculated based on 29 signature genes, including markers of immune cells such as T cells, monocytes, macrophages, and fibroblasts[58]. The TLS score in the DCIS dataset was calculated with the *scanpy.tl.score_genes* function in SCANPY[89], as the averaged expression of a set of genes subtracted by the averaged expression of a set of randomly sampled genes.
- Cell–cell communication. Thor integrates the python package COMMOT[65] to analyze cell–cell communication, which accounts for competition among different ligand and receptor species as well as spatial distances between cells. Thor boosts the calculation by implementing a more efficient function to compute the cell–cell spatial distance matrix within the interaction cutoff distance in place of the original implementation.

### Post-LVAD heart failure ST data collection

**Sample collection and preparation.** Tissue sections (10 μm) obtained from the FFPE tissues were mounted on Visium spatial gene expression slides (10x Genomics, 1000520). The samples were processed as described in the manufacturer's protocols.

**ST by 10x Genomics Visium.** The tissue slides were permeabilized at 37 °C for 6 min, and polyadenylated mRNA was captured by oligonucleotides bound to the slides. Reverse transcription, second-strand synthesis, complementary DNA (cDNA) amplification, and library preparation proceeded using the Visium Spatial Gene Expression Slide & Reagent Kit (10× Genomics, 1000520) according to the manufacturer's protocol. After evaluation by real-time PCR, cDNA amplification included 13–14 cycles. Indexed libraries were pooled equimolarly and sequenced on a NovaSeq X Plus instrument in a PE28/150 run (Illumina). An average of 26,011 paired reads were generated per spot and the median genes per spot were 2277. Tissues were stained with H&E, and slides were scanned on a Pannoramic MIDI scanner (3DHISTECH) using a ×20, 0.8-NA objective.

**Spatial profiling of vascular protein.** To capture the spatial distribution of the candidate protein (PLA2G2A), we adapted an established protocol for spatial mapping using IF staining. This technique provides precise localization and analysis of the candidate gene's expression patterns across different tissue regions. First, paraffin-embedded sections were deparaffinized with xylene thrice for 5 min each. The sections were then rehydrated through a series of ethanol washes: twice in 100% ethanol for 2 min each, twice in 95% ethanol for 2 min each, and

once in 75% ethanol for 2 min. The slides were then rinsed in ultra-pure water for 5 min, followed by Tris-buffered saline (TBS) containing 0.0025% TritonX−100 for 5 min. For antibodies recognizing surface proteins, a rinse with 1x TBS alone was used. Subsequently, the slides were subjected to antigen retrieval by placing them in a sodium citrate solution heated to 85 °C on a hot plate for 10 min. The sections were then encircled with a pap pen, and the primary antibody, anti-PLA2G2A (Thermo Fisher Scientific, Cat# PA5-102403; dilution 1:200), was applied overnight in a dark, humidified chamber at 4 °C. The following day, the slides were washed twice with either 1x TBS containing TritonX-100 or 1x TBS alone, depending on the nature of the protein of interest. Next, the slides were incubated with the secondary antibody, Donkey anti-Rabbit Alexa Fluor 647 (Thermo Fisher Scientific, Cat# A-31573; dilution 1:200), in a dark chamber for 30 min. After incubation, the slides were washed twice and mounted using a DAPI-containing mounting medium. Microscopy images were obtained using an Olympus FV3000 Confocal microscope. A negative control slide was used to establish the threshold settings, which were consistently applied to all slides for image acquisition.

## Human bladder cancer Visium HD data collection

Pre-treatment FFPE tissue sections from muscle-invasive bladder cancer patients were processed using the 10x Genomics Visium HD Spatial Gene Expression platform according to the manufacturer's protocol (Visium HD Spatial Gene Expression Reagent Kits for FFPE, CG000663). Briefly, 5 μm-thick FFPE sections were mounted onto Visium HD slides and baked at 60 °C for 1 h. Following deparaffinization and heat-induced epitope retrieval, tissue sections were hybridized with probe sets targeting the human transcriptome and subjected to reverse transcription to capture spatial barcodes and unique molecular identifiers (UMIs). Slides were then permeabilized at 37 °C for 30 min to release the barcoded cDNA fragments, which were collected and amplified to generate sequencing libraries. Whole-transcriptome libraries were sequenced on a Novaseq X plus instrument (Illumina) to a target depth of ~200–300 million reads per capture area. Image alignment, UMI counting, and spatial mapping were performed using 10x Genomics' Loupe Browser and Space Ranger (version 3.1.0) pipeline. Quality-control metrics were reviewed to exclude low-quality bins prior to downstream analyses.

## Preprocessing the histology images

In order to accurately detect cells/nuclei from histology images, pre-processing steps including image normalization and augmentation of the histology images in this study adhered to recommended settings of the cell segmentation tools. For StarDist, pixel values were clipped at 1% and 99.8% for all the (red, green, blue) color channels, and the trained model "2D_versatile_he" was used. Cellpose internally includes data normalization in the neural network. Following the recommendations in Squidpy, we inverted the color values of the H&E images and used the blue channel for nuclei segmentation[24].

For the MOB dataset, nuclei were segmented from the H&E staining images with Cellpose[7] using the parameters (*min_size = 10, flow_threshold = 0.4, channel_cellpose = 0*). For the human MI datasets, the 10x Genomics human breast cancer Xenium & Visium datasets, the 10x Genomics human DCIS dataset, and the post-LVAD human heart tissues, StarDist[73] was used with the parameters (*prob_thresh = 0.05, nms_thresh = 0.2*). For the mouse brain MERFISH dataset, the cell segmentation downloaded along with the data was used.

## Preprocessing the spatial transcriptomics data

The initial preprocessing steps involved quality control and library size normalization, adhering to the SCANPY standard protocols[89]. Highly/spatially variable genes were identified by established protocols[89–91] for inference and the following downstream analyses. A low-dimensional representation was obtained through dimension reduction methods, including PCA, UMAP, or by utilizing the latent space of a VAE.

## Parameter settings in Thor inference

Thor inference demonstrated robust performance to the variations in parameter settings. Therefore, in all analyses of this study, default parameters in Thor were employed, with some specific configurations as outlined below. The construction of the cell neighborhood graph utilized an initial KNN approach, setting the number of neighbors to 5 (*n_neighbors = 5*). Additionally, the probability of retaining the original (self) gene expression, denoted as $1 - \lambda$ in Eq. (6), was set to 0.2 (equivalently in Thor, *smoothing_scale = 0.8*), and the total number of diffusion steps was specified as 20 (*n_iters = 20*).

## CytoSPACE

CytoSPACE was performed on the Human ductal carcinoma in situ by Visium dataset for two purposes. First, a breast cancer scRNA-seq atlas by Wu et al.[92] was used as the reference to project the cell type information to the ST spots. Second, a DCIS scRNA-seq dataset[64] with CNV profiles validated through WGS was used as the reference to project the CNV profiles to the ST spots. Default parameters were used in both.

A two-step approach was devised to validate Thor's CNV predictions using external data. In the first step, a reference scRNA-seq DCIS dataset was leveraged[64], whose CNV profiles were predicted by CopyKAT and validated by paired WGS data. In the second step, CytoSPACE was employed to map these reference single cells onto the Visium dataset, enabling the derivation of spot-level aneuploid cell proportions from the known CNV profiles.

## RCTD

RCTD was run on the Human ductal carcinoma in situ by Visium dataset to deconvolute the cell type proportions in the spots. An annotated breast cancer scRNA-seq atlas by Wu et al. was used as the reference data[92]. Default parameters were used.

## iStar

For iStar prediction of superpixel level gene expression on the Human breast cancer by Visium data, default settings recommended in the documentation in the GitHub repository (https://github.com/daviddaiweizhang/istar) were used. We applied iStar to the post-Xenium H&E image and the paired Visium dataset. We set the desired pixel size to 0.25 μm for high-resolution inference of the spatial gene expression.

## BayesSpace

For enhancing spatial features on the simulation data, we set the number of clusters to the ground truth number of clusters, the number of PCA components to 10, and spatial-enhancing Markov chain Monte Carlo (MCMC) rounds to 50,000. For enhancing spatial features on the mouse MERFISH-generated spot data, we set the number of clusters to 8, number of PCA components to 10, and the spatial-enhancing MCMC rounds to 50,000. For enhancing spatial features on the 10x human breast cancer by Visium dataset, we set the number of clusters to 6 (the number of major clusters identified in the reference Xenium dataset), number of PCA components to 10, and spatial-enhancing MCMC rounds to 50,000. Other parameters were set to their default values.

## Bin2Cell

Following the demonstration script from the official GitHub repository (https://github.com/Teichlab/bin2cell), Visium HD 2 μm bins to cells based on segmentation from both the H&E image and gene expression. The input image was rescaled to 0.5 micrometer per pixel (*mpp = 0.5*), and all other parameters were kept at their default values. For the bladder cancer dataset, a total of 202,482 cells were detected; after

filtering cells with less than 5 assigned bins, 73,597 cells were left for analysis.

### Cell type annotation by Cell-ID

Cell-ID was used to transfer the cell type information from an annotated reference scRNA-seq data to annotate single-cell data inferred by Thor[35]. Cell-ID was performed by a per-cell assessment in the query dataset evaluating the replication of gene signatures extracted from the reference dataset.

We followed the Cell-ID vignette and used the default parameters. For the MOB dataset from 10x Genomics, we used the cell type signatures from the scRNA-seq data[34]. For the human DCIS dataset from 10x Genomics, we used the cell type signatures from the scRNA-seq data for Cell-ID annotation[18]. We further refined the annotations by using expression levels of key gene signatures, including *EPCAM* and *CDH1*, to distinguish between normal and tumor epithelial cells. Similarly, monocytes and macrophages were separated by using marker genes *VCAN* (versican) and *CD14*, which were upregulated in circulating monocytes and reduced upon differentiation to macrophages (Supplementary Fig. 20a).

### Pathway enrichment analysis

Functional enrichment analysis was performed using the over-representation analysis (ORA) method implemented in the Python package *decoupler*[63,93]. For each cell, the top expressed genes were treated as the set of interest. For a given gene set (e.g., a GO term), a one-sided Fisher exact test was used to assess the significance of overlap between the gene sets. The resulting $p$ values were log-transformed to yield enrichment scores, where higher scores indicate greater significance.

### Transcription factor activity inference

The database CollecTRI and Python package *decoupler* were used for the TF activity inference. CollecTRI is a comprehensive resource comprising weighted transcriptional regulatory networks of TF–target gene interactions[40]. TF activities were estimated using the univariate linear model method implemented in *decoupler*[63], by predicting gene expression levels based on the TF–Gene interaction weights from CollecTRI. The resulting TF activity scores provide directional insights: positive scores indicate active TFs driving gene expression, and negative scores suggest inactivity or repression.

### Gene module identification

The package Hotspot[36] was used for the identification of informative genes in the single-cell level spatial transcriptome dataset. For the module assignment by Hotspot in the MOB dataset, the number of nearest neighbors was set to 30 for creating the KNN graph. A false discovery rate cutoff of 0.05 was applied, grouping 1688 of the 2781 highly variable genes into 8 modules.

### Datasets and preprocessing

**Human breast cancer by Xenium and Visium.** We mapped the Visium reads to the post-Xenium H&E staining image (In Situ Sample 1, Replicate 1) using 10x Genomics Space Ranger software (v2.1.0) for direct comparison between Thor-inferred result and Xenium data. The processed Visium gene expression matrix of the 306 genes, found commonly in the Xenium and the Visium datasets, and the post-Xenium H&E image were utilized as input for Thor/iStar/TESLA/BayesSpace. The Xenium data was employed as a reference for assessing the performance of Thor with the other software and was excluded during prediction.

**Human ductal carcinoma in situ by Visium.** The gene expression matrix was preprocessed and log-normalized expression of 2748 highly variable genes was used to train a VAE network for accelerating Thor inference. The dimension of the latent space of VAE was set to 20.

**Human healthy heart sample by Visium.** The gene expression matrices and spot level expert annotations were provided in the annData files from the original publication[37]. The sample IDs include "HCA-HeartST11702008" (vessel: S1), "HCAHeartST12992072" (vessel: S2), "HCAHeartST9383353" (vessel: S3), "HCAHeartST11290662" (node: S4), "HCAHeartST11702008" (node: S5), "HCAHeartST11702009" (node: S6), "HCAHeartST13228106" (adipose: S7), "HCAHeartST9383354" (adipose: S8), "HCAHeartST13228103" (adipose: S9), "HCAHeartST13228106" (fibrosis: S10), "HCAHeartST11350377" (fibrosis: S11), and "HCAHeart ST8795936" (fibrosis: S12).

**Human myocardial infarction by Visium.** Samples "10X0025" (RZ1), "ACH0019" (RZ2), "ACH0012" (IZ1), "ACH0014" (IZ2), "ACH008" (FZ1), "ACH006" (FZ2) were downloaded for analysis. To facilitate the comparison of tissues from ischemic, fibrotic, and remote zones, where the expressed genes exhibited substantial variations, we aimed to maximize the overlap of genes among the six samples. After filtering out genes which were not expressed in any spot, we inferred the expression of all the remaining genes.

**Human post-LVAD heart failure by Visium.** The gene expression matrices were obtained by using Space Ranger (v2.1.0), referencing the GRCh38 human Genome. The gene expression was preprocessed following SCANPY standard protocols. Log-normalized expression of all expressed genes was used as input.

**Human bladder cancer by Visium HD.** The gene expression matrices were obtained by using Space Ranger (v2.1.0), referencing the GRCh38 human Genome. Gene expression matrices of the 2 µm square bins and 8 µm square bins were preprocessed using SCANPY. Log-normalized expression of highly variable genes of 2 µm square bins was used as input. Log-normalized expression of 8 µm square bins were used for comparison.

**Mouse olfactory bulb by Visium.** The gene expression was pre-processed following SCANPY standard protocols. Log-normalized expression of highly variable genes was used as input. A VAE network was trained to allow inference in the latent space. For evaluation, we downloaded the ISH images of selected genes in MOB from Allen Brain Atlas[33] and the gene expression data from the Stereo-seq study[27].

**Mouse brain by MERFISH.** We used the Vizgen MERFISH mouse brain receptor map dataset that contains a MERFISH measurement of a 483 gene panel. Sample Slice 2 Replicate 1 was used. The DAPI image was used for extracting image features of single cells. 8597 cells in the hippocampus region were extracted (Supplementary Fig. 4a). After preprocessing, the log-normalized expression in 535 synthetic spots and the DAPI image features were used as input for Thor.

### Simulation details

Thor's accuracy, sensitivity, and limitations of Thor were evaluated on simulated ST data under conditions, including diverse sources of ground truth data, variation in spot sizes, missouts in cell identification, false connections in a cell–cell network, and technical dropouts in sequencing. We extracted the positions of 6579 cells in a mouse cerebellum Slide-seq data[32], including the Granular (Cluster 1), Oligodendrocyte (Cluster 2), and Purkinje (Cluster 3) cells. Those cell locations reliably reflect the spatial distribution of cells in the real tissue and the gene counts in the single cells were simulated using Poisson distributions. We simulated a single-cell ST dataset by generating 1000 genes of distinct spatial expression patterns, acting as markers for the three cell types. This included 350 genes for each of

the first two cell types and 300 genes for the third cell type. Specifically, for the marker genes, the mean values of the Poisson distributions ($\lambda$) were randomly sampled in the range of (100, 200); and for the non-marker genes, $\lambda$ values were randomly sampled in the range of (10, 20). Spots were then created on a grid and the spot-resolution gene expression levels were aggregated values of the enclosed cells.

(i) To assess the effect of different spot sizes, we simulated a series of spot diameters ranging from 25 to 150 μm, with nearby spots separated by 100 μm.

(ii) To assess the effect of cell missouts, we randomly dropped 10, 20, 30, and 40% of the cells.

(iii) To assess the effect of the false connections in the cell–cell network, we added randomized connections, until 10, 20, 30, and 40% of the cells contained randomized connections. In this evaluation, we did not directly use an image, instead, the cell positions were predefined, and the cell types were converted to one-hot vectors as image features. These features, combined with the generated spot-level gene expression, constituted the input for Thor, using default parameters to infer the cell-level ST data.

Additionally, to systematically assess the effect of technical dropouts, we simulated single-cell gene expression using the R package Splatter[94] with no dropouts and with variable levels of dropouts. Splatter models the probability of transcript dropouts using a logistic function based on the mean expression levels $P_{dropout}(x) = 1/(1 + e^{-k(x - x_0)})$, where $x$ is the mean expression level. The probability of transcript dropouts is controlled by two parameters, the midpoint parameter ($x_0$ or $dropout.mid$) and the shape parameter ($k$ or $dropout.shape$). The former is the expression level at which 50% cells are zero, and the latter controls how quickly the probabilities change from the midpoint. To simulate a wide range of dropout conditions, we used combinations of $dropout.mid$ values [1, 2, 3, 4, 5] and $dropout.shape$ values [−1, −2, −5], with percentages of zero reads up to 63%. This allowed a comprehensive assessment of the impact of varying dropout levels on Thor's performance. Spot-level gene expressions generated from these single-cell simulations with dropouts were then used as inputs for Thor.

We employed the Silhouette coefficient and Calinski-Harabasz index to measure separation of cell clusters. The scores were calculated on the PCA embeddings of the corresponding gene expression arrays using the functions from the library *scikit-learn*. We randomly sampled 3000 cells (without replacement out of all the 6579 cells) 10 times in the calculation of the mean Silhouette coefficients to access statistical robustness.

The MERFISH data of the mouse brain receptor map consists of 83,538 cells and 483 genes. We simulated Visium-like spot-level data by creating a grid of evenly spaced "spots". The molecule counts in a synthetic spot were aggregated over all the cells covered by the "spot". The spot size was set to 100 μm and a total of 4870 spots were simulated. We focused on the hippocampus region (Supplementary Fig. 4a), which consists of 535 spots covering 8597 cells. For visualization purposes, major Leiden cell clusters in the original data were annotated according to the cellular locations in the hippocampus components and a previous study[19]. Minor cell clusters were merged and labeled as "Others". A DAPI image in the dataset and the generated spot-level gene expression were jointly analyzed by Thor. For comparison, the positions of cells segmented from the source were used as our cell positions. Image features, including the mean and standard deviation of the grayness and the entropy of image patches surrounding the cells were calculated. The predicted transcriptome and the ground truth transcriptome were integrated by harmony[95].

## Evaluating cell-level spatial gene expression prediction accuracy
We used the root mean square error and structural similarity index to quantify the prediction accuracy for each gene. In the simulation

datasets, NRMSE was employed to calculate the mean deviation of the predicted gene expression from the ground truth data in all cells, as defined in Eq. 10.

$$\text{NRMSE} \overset{\text{def}}{=} \frac{\sqrt{\frac{\sum_{i=1}^{n}\left(x_i^{\text{pred}} - x_i^{\text{truth}}\right)^2}{n}}}{\frac{\sum_{i=1}^{n} x_i^{\text{truth}}}{n}} \tag{10}$$

where $x_i^{\text{pred}}$ ($x_i^{\text{truth}}$) is the predicted (ground truth) gene expression in cell $i$; and $n$ is the number of cells.

To assess the performance of predicted gene expressions by Thor and other methods, we calculated two pixel-centric metrics, including SSIM, RMSE (pixels), and a cell-centric metric RMSE (cells). For pixel-centric metrics, similar to the methodology from the iStar study[17], both the ground truth and predicted gene expression were treated as grayscale images. Considering spatial contexts within the images, we calculated SSIM between the spatial structures of the ground truth and predicted gene expression images. Practically, we observed that slight local distortions and shifts existed between the Visium and Xenium slides. For a more reliable measure of the prediction quality, we therefore calculated the Complex Wave SSIM, which is insensitive to consistent spatial translation[96]. SSIM values range from 0 to 1, with 1 indicating identical images and 0 indicating no similarity. For cell-centric metrics, we aggregated the nearby gene expression of superpixels to the ground truth cells.

## Comparison between Thor results and pathology annotation
To quantitatively assess Thor's SSA tool, we compared Thor against spot-level expert annotations. Because Thor assigns labels at the single-cell level, we employed majority voting to map these cell-level annotations to each spot. This allowed a direct comparison with expert-labeled spots. We report two commonly used classification metrics, accuracy, and area under the curve (AUC) as defined below. Accuracy is defined as the ratio of correct predictions to the total number of predictions. The receiver operating characteristic (ROC) curve plots the true positive rate (sensitivity) against the false positive rate (1−specificity) at various threshold settings; the area under this curve (AUC) summarizes the overall performance. A higher AUC suggests better overall classification performance.

To quantitatively assess Thor's prediction of aneuploid cells through CNV analysis, we compared against pathology-annotated tumor regions. We used two metrics F1 score and the Jaccard index. F1 score is calculated as the harmonic mean of precision and recall, and a higher F1 score indicates a better balance between precision (the proportion of predicted positives that are truly positive) and recall (the proportion of true positives that are correctly identified). Jaccard index measures the degree of overlap between predicted and reference sets by dividing the size of their intersection by the size of their union; values closer to 1 indicate a higher concordance between the two.

## Statistics and reproducibility
No statistical method was used to predetermine sample size. No data was excluded from the analyses and the experiments were not randomized.

## Data availability
The sequencing data generated in this study have been deposited in the Gene Expression Omnibus database. *Human post-LVAD heart failure by Visium:* accession code GSE300585. *Human bladder cancer by Visium HD:* accession code GSE300692. Published data used in this study were downloaded using the links provided in the publications. *The Human breast cancer by Xenium & Visium dataset:* https://www.10xgenomics.com/products/xenium-in-situ/preview-dataset-human-

breast. *Human ductal carcinoma* in situ *by Visium*: https://www.10xgenomics.com/resources/datasets/human-breast-cancer-ductal-carcinoma-in-situ-invasive-carcinoma-ffpe-1-standard-1-3-0. *Human healthy heart sample by Visium:* https://www.heartcellatlas.org/. *Human myocardial infarction by Visium*: https://zenodo.org/records/6580069#.ZHYP9OzMK3I. Samples "10X0025" (RZ1), "ACH0019" (RZ2), "ACH0012" (IZ1), "ACH0014" (IZ2), "ACH008" (FZ1), "ACH006" (FZ2) were used for analysis. *Mouse olfactory bulb by Visium*: https://www.10xgenomics.com/resources/datasets/adult-mouse-olfactory-bulb-1-standard-1. *Mouse brain by MERFISH*: https://info.vizgen.com/mouse-brain-map?submissionGuid=5606514b-5a81-4405-999e-327f908281cc. We used a MERFISH dataset with a 483 gene panel (Sample Slice 2 Replicate 1) and the DAPI image "mosaic_DAPI_z2.tif". Source data are provided with this paper.

## Code availability

The source code for the Thor inference framework, including all analysis modules and parameter settings, is available at: https://github.com/GuangyuWangLab2021/Thor. Scripts used to reproduce the results presented in this study are available at: https://github.com/biopzhang/thor_scripts_ncomm.

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

## Acknowledgements

This work was supported by Houston Methodist internal grant to G.W., National Institute of General Medical Sciences of the National Institutes of Health (R35GM150460 to G.W., R35GM151089 to Q.S.), National Heart, Lung, and Blood Institute (HL169204-01A1 to L.L), and National Cancer Institute (U54CA274375, R01CA175397 to K.S.C). This work utilized the Houston Methodist Neal Cancer Center Spatial Omics Core funded by Cancer Prevention and Research Institute of Texas (RR230010). We appreciate Drs. Qin Ma and Jordan Krull from the Ohio State University for critical reading and discussion.

## Author contributions

G.W. supervised the study. P.Z. and G.W. designed and developed the graph diffusion model. P.Z., W.C., T.N.T., I.K., and S.L. performed the data analyses. P.Z., T.N.T., and M.Z. developed the web platform. Y.S. contributed to the development of the web platform. L.L. and K.N.C. prepared the post-LVAD patient tissues and performed the IF staining. Y.Y. annotated the bladder cancer tissue image. X.H., F.N., and K.S.C. prepared the bladder cancer sample and performed Visium HD sequencing. P.Z., W.C., K.N.C., L.L., and G.W. wrote the manuscript. Q.S. helped supervise the development of the web platform. L.L. supervised the IF staining experiments. J.P.C., F.N., K.S.C., L.L., K.Y., Z.L., Y.Y., and Q.S. were involved in the discussions and helped improve the manuscript. All authors contributed to writing the manuscript and have approved the submitted version.

## Competing interests

The authors declare no competing interests.
