## [Transparent Peer Review file · Nature Communications]

Thor: a platform for cell-level investigation of spatial transcriptomics and histology

Corresponding Author: Dr Guangyu Wang

Version 0:

Reviewer comments:

Reviewer #1

(Remarks to the Author)

In this study, the authors presented Thor, a spatial analysis tool that infers cell-resolution transcriptomes from spot-level spatial transcriptomics (ST) data and integrates them with histology. The tool shows potential for ST data analysis and could provide valuable insights. However, there are several major limitations in the current manuscript that need to be addressed. Further benchmarking, analysis, and validation studies are necessary. Please see the detailed comments below:

1. While it is advantageous to have single-cell resolution imputation from Visium, the cell graph inferred by Thor does not accurately reflect the true cell morphology, as the "cells" primarily represent cell nuclei. Importantly, the density of cell nuclei predicted by Thor is quite different from what is observed in the histology images, particularly in regions with a high cell density. The authors should address these discrepancies and provide an explanation or adjustments to improve the accuracy of the inferred cell morphology.
2. With the introduction of the new Visium HD platform, what is the added value of Thor? Is there still a need for Thor, given the enhanced resolution of Visium HD? Furthermore, no Visium HD data is presented in this study. Including such data would help demonstrate Thor's relevance and utility in comparison to this newer platform.
3. In Fig. 2d, ROI5 (top row), Thor does not appear to maintain the spatial gradient pattern observed in the Xenium data and iStar results. For example, the outer layer of cells showed higher expression in both the Xenium data and iStar results, which was not captured by Thor. The authors should address this inconsistency and discuss its potential impact on the accuracy of the analysis.
4. It would be helpful to benchmark Thor against other methods, such as CytoSpace and other single-cell resolution deconvolution tools, to evaluate its performance. This comparison would provide a clearer understanding of Thor's strengths and limitations in producing single-cell resolution results.
5. The annotation of the fibrotic area in Fig. 3b does not seem to add significant value, as these areas are already visible from the H&E images. Additionally, this annotation does not provide insights into the phenotypic and functional states of the fibroblasts. Furthermore, how was T cell proliferation inferred in this analysis? Were the authors able to validate the T cell proliferation signals? Providing more details and validation would strengthen the findings.
6. The section on CNV analysis does not appear to be novel and add significant value, as there are already existing methods available for CNV analysis. The authors should clarify how their approach differs from or improves upon these established methods. How does Thor's spatial InferCNV analysis compare to spot-level spatial InferCNV, and what advantages does Thor offer?
7. The analysis of tertiary lymphoid structures (TLS) appears quite superficial and does not add value beyond existing methods. It also does not seem to perform well in accurately capturing TLSs. Proper pathology annotation and multi-plex staining are necessary to define TLS, its maturation status. The structure shown in Figure 6c is more likely a lymphoid aggregate (LA).
8. How does Thor's performance depend on the quality of histology images and transcriptomics data? It is suggested that the

authors demonstrate Thor's performance in scenarios where one of the two modalities is suboptimal.

9. How does the semi-annotation function align with spot-level pathology annotations? It is recommended that the authors conduct a quantitative analysis across a cohort of samples to evaluate this alignment. Additionally, how does Thor handle other key histological features in tissues, such as key microstructures within the tissues, other cell components such as macrophages, neutrophils, plasma cells, and cancer cell histology? Providing insights into these aspects would greatly enhance the utility and applicability of the tool.

10. Benchmarking is missing for both the TLS and CNV analyses. The authors should provide a more thorough evaluation and comparison with existing methods.

11. The analysis in Figures 5f and 5g is quite superficial, as checking gene expression and signature scores can be done on any spatial platform. The authors should emphasize the unique strengths of their algorithm in this analysis.

12. Additionally, it is unclear how much training data is required for Thor to achieve reliable results, and how is the accuracy of its predictions evaluated? The authors should provide details on the training data size and the methods used to assess prediction accuracy to ensure the robustness of the tool.

13. The data presented in Figure 5 require validation to strengthen the conclusions drawn from the analysis. The authors should provide additional experimental or independent validation to support these findings.

Minor comments:

1. the introduction and the first section of the results, the authors frequently use the term "genomic data," whereas the data utilized in the study is actually transcriptomic. It would be more accurate to adjust these descriptions to "transcriptomic data" to avoid potential confusion.

2. It would be helpful to assess Thor's performance with varying levels of dropouts in Figure S1. This evaluation will help readers better understand the robustness of the algorithm and the extent to which it can handle data dropouts.

3. Please provide quantitative measurements of the correlation between the imputed molecule expression and the MERFISH ground truth in their spatial locations for Figures S4b and S5b.

4. It would be beneficial to assess the relationship between cell type complexity within a spot and imputation accuracy. Specifically, does the accuracy decrease with increasing spot complexity, and if so, to what extent?

5. Please include quantitative measurements, such as the silhouette score, for Figure S4c to provide a more thorough analysis.

6. For Figure 2b, it would be useful to show cells from both Xenium and Thor in the joint embedding and assess the similarity of the same cell types defined from both modalities.

7. The benchmarking in Figure 2 is too limited, as only one published tool is tested. Please benchmark Thor against other existing tools such as cytospace to give readers a comprehensive view. For Figures S4 and S5, additional benchmarking with tools like iStar would be helpful.

8. In the section "Thor unveils refined tissue structure in mouse olfactory bulb," it would be beneficial to describe the tissue structures in combination with the H&E image, including the layers and cell types present. This will help readers better understand the complexity and results shown in Figures S8b-c.

9. Please label the regions in Figure 4a with numbers for clarity.

10. It would be useful to benchmark Thor against more spatial deconvolution methods, such as RCTD in Figure S16.

11. For Figure 5c, please provide a more comprehensive marker gene list for each cell type to assess the accuracy of the defined cell types. Additionally, distinguish between normal and tumor epithelial cells, allowing readers to check the correlation with the pathologically annotated tumor regions in Figure 5a. Showing classic tumor/epithelial markers (e.g., EPCAM, CDH1) in parallel with Figure 5d would also be helpful for comparison.

12. The results in Figure 5e indicate that tumor cell heterogeneity and their microenvironment might influence predictions. Since manual curation of cells is often impractical in high data volumes, the authors should assess how frequently such issues occur across different tumor samples. Additionally, the identification of adipose tissue in CLAM seems problematic, and a thorough assessment of potential circumstances where this mistake could occur would help inform users on proper tool selection and usage.

13. The results in Figures 6a-b are confusing, as tumor regions should indicate areas enriched with tumor cells, whereas TLS/LA should be independently identified in pathology. A detailed pathology annotation to distinguish tumor-enriched regions from TLS/LA regions is needed.

14. In Figure 5i, T7 did not show any CNVs, which is counterintuitive given the pathology annotation. Please provide explanations for this observation.

15. The analysis in Figures 6c-d doesn't add much value, as it merely compares cellular compositions in different regions. The authors should directly compare the different TLS regions identified, revealing differences in functions, maturity, relationships, and interactions with adjacent tumor regions to provide more informative insights.

16. How well does Thor handle the integration of multiple samples from a cohort, including samples from adjacent sections or from different tissue types? Additionally, could Thor enhance the integration of Visium data with other modalities, such as CODEX? Although this may be beyond the current scope, providing insights on this potential would be valuable.

(Remarks on code availability)

Reviewer #2

(Remarks to the Author)

(Remarks on code availability)

Reviewer #3

(Remarks to the Author)

This study introduced a new computational platform called Thor, specifically designed to integrate spatial transcriptomics data with histological images for cell-level analysis. Thor leverages an anti-shrinking Markov diffusion method to infer single-cell spatial transcriptomes from spot-level data, effectively combining spatial transcriptomics with histological images molecular information. Additionally, Thor encompasses a suite of modules for performing multi-modal genomic analyses, including gene expression, pathway enrichment, transcription factor activity, and copy number variation. The authors have also developed an interactive web tool named Mjolnir, which enables users to explore cellular organization and disease mechanisms through an intuitive interface.

The study employs advanced computational methods, such as the anti-shrinking Markov diffusion, which enhances the accuracy of inferring single-cell resolution gene expression from spatial transcriptomics data. Thor has been validated using multiple experimental datasets, demonstrating its significant contribution to current spatial biology analysis. This study provided a novel tool for comprehensive cellular and molecular analysis, thereby advancing the field of precision medicine. However, several issues needed to be addressed:

1. "Introduction" section, the author stated "Existing analysis platforms of spatial biology focus on -omics analysis without deeply analyzing tissue histological images." Actually, there are several analysis platforms such as iStar, it is recommended to review the existing platforms and point out the limitation of the existing platforms.

2. In the "Method" section, Parameter settings in Thor inference," it remains unclear whether the chosen parameter settings are based on specific rationale. How do different parameter choices impact the analysis results?

3. The accuracy of the Thor platform for histological imaging analysis is dependent on the quality of the input data. The quality control process of histological images is particularly important. How to ensure the consistency of histological data from different sources. Inconsistency and quality of data varying may will limit the application of Thor platform

4. In Figure S1, the authors compare the accuracy and stability of Thor, Nearest spot, and KNN smoothing for spatial transcriptomics-based cell clustering. How does Thor's performance compare to other commonly used methods, such as ClusterMap or BayesSpace?

5. In Figure 3c, UMAP was used to visualize clusters in fibrotic and non-fibrotic regions, but the separation between the two regions does not appear distinct. Furthermore, Figure 3e shows that the inflammatory response and fibrosis in distant regions are more pronounced than in infarct and ischemic zones. What could be the underlying reasons for this? Providing relevant marker genes or differential genes for these regions would be helpful.

6. In Figure 5C, the researchers used Cell-ID to obtain the spatial distribution of cell types in tumor tissue. However, tumor cell annotations are absent in Figure 5C. Additionally, monocytes and macrophages are grouped together. Can these two cell types be distinguished? If not, what could be the possible reasons for this?

7. Cells of the same type may exhibit different subtypes due to their environment or spatial location. Could the study improve upon the classification of cell subtypes?

8. The study infers cell types and gene expression from spatial transcriptomics data. Compared to single-cell transcriptomics, spatial transcriptomics offers significant advantages for studying cell-cell communication. Could Thor generate the cell communication module in this research?

9. Did the Tho have the capability of inferring cell-level information from metabolomics or proteomics.

10. Now, the Stereo-seq technology can provided the resolution of life-space expression at the subcellular level. Whether Thor can be compared to this technology?

11. Some data analysis frameworks that have enabled accurate identification of histological structures at cellular resolution such as PROST, and METI. Please discuss the advantages of Tho over current technologies?

(Remarks on code availability)

Reviewer #4

(Remarks to the Author)

The paper presents an algorithm, Thor, a comprehensive computational tool designed for spatial transcriptomics and histology analysis. Thor aims to bridge the gap between ST and image analysis, allowing for multi-modal exploration of tissue characteristics. The platform features an anti-shrinking Markov diffusion method that infers cell-resolution transcriptomes from spot-level data and integrates these with histological features. Thor is supplemented by Mjolnir a web-based tool for interactive tissue analysis. The platform offers functionalities like immune response analysis, transcription

factor activity estimation, pathway enrichment, and semi-supervised tissue annotation. Its performance is validated across different datasets, including heart failure tissues, mouse olfactory bulb, and breast cancer tissues. The topic is very interesting and it is noteworthy that it shows a possibility that overcomes current limitations of ST in terms of resolution. There are some limitations and drawbacks of this paper.

- Although Thor is compared to other tools like iStar, the scope of comparison is relatively narrow. It lacks a detailed, quantitative evaluation against a broader set of established tools in spatial transcriptomics, such as TESLA, BayesSpace, etc. More comparisons with varied metrics would solidify its advantages.

-The limitation of Thor's approach is that recent advancements in ST technologies, such as VisiumHD, Slide-seq, Xenium, and CosMX, are already enhancing resolution significantly. For instance, image-based ST platforms like Xenium and CosMX can provide cellular-level resolution while analyzing thousands of genes (>5k), potentially reducing the necessity for computational inference methods like Thor. If ST platforms can inherently deliver cellular resolution along with a sufficient number of gene sets for comprehensive transcriptomic analysis, the added value of Thor in terms of improving spatial resolution may be diminished. Therefore, Thor's utility under these conditions could be limited, and its impact may need to be reassessed in the context of increasingly advanced ST technologies that natively achieve high resolution.

-Regarding the use of CW-SSIM for evaluation, there are concerns about whether the comparison between Xenium, Thor, and iSTAR is entirely fair. Specifically, the output of Thor and Xenium is not at the pixel level, whereas iSTAR generates outputs at a pixel level, making the comparison potentially biased in favor of pixel-level evaluation. A more nuanced evaluation metric, tailored to the inherent characteristics of each method, might be more appropriate.

- Thor's evaluation could be extended to include comparisons with simpler approaches, such as nearest value assignment after nucleus segmentation or interpolation from surrounding spot values. This would help clarify whether Thor's more sophisticated approach offers a genuine advantage or if the improvements are merely due to incorporating basic nucleus segmentation. It is important to differentiate whether Thor's better performance results primarily from its advanced diffusion model or simply from the inclusion of nucleus segmentation. Adding these comparisons would provide a clearer perspective on the unique contributions of Thor.

- Thor heavily relies on histological images to infer cellular-level information. If high-quality histological images are unavailable or the images are noisy, the accuracy of in silico cell transcriptomes may be affected. In addition, sometimes MPP (maybe 0.25 um) cannot be obtained by other real-world situations. The paper does not discuss how Thor handles degraded or suboptimal images.

- While Thor identifies molecular signatures in different tissue zones and conditions, there are limited wet-lab biological validations or 'well known' acceptable validation for these findings. In terms of biological effects of pathological findings, it needs to be validated that Thor could result in discovering novel results which couldn't be discovered by Visium data alone.

(Remarks on code availability)

Graph Construction and Markov Diffusion part: The Markov diffusion method employed here involves constructing a connectivity matrix, followed by iterative graph-based smoothing. This complexity is appropriate given the need for nuanced spatial analysis. The code lacks sufficient parameter sensitivity analysis. For example, the choice of diffusion step size or the Gaussian kernel used for affinity computation may significantly impact the outcome. Providing grid search or comparison according to the hyperparameters of the results to different parameter values would add robustness to the analysis.

In VAE, latent dimension and input dimension (maybe depends on HVGs) could be changed. It also affect the results. To have more generalizability, hyperparameter tuning seems to be required.

Version 1:

Reviewer comments:

Reviewer #1

(Remarks to the Author)

The authors have addressed most of the minor comments but only a subset of the major concerns. While the manuscript has been improved, several key issues remain unresolved and should be addressed before publication.

Major Concerns:

1. Novelty and Comparison with Recent Methods

Several new methods have been published in the past few months, including MISO (Coleman et al., PMID: 39815104, Nature Methods, March 2025) and Spotiphy (Yang et al., PMID: 40074951, Nature Methods, March 2025), among others in high-impact journals. These methods offer robust and novel approaches, raising concerns regarding the novelty of Thor. A key question that remains unaddressed is: How does Thor improve upon these recently established methods? A more thorough comparison with these methods is necessary to justify Thor's contributions.

2. Comparison with Existing Methods (Original Comment #1 & #2)

The authors claim that Thor maintains robust accuracy despite cell dropout (Response Figure 1, Figure S1b), yet no direct comparison with existing methods is provided. Benchmarking against other state-of-the-art tools would significantly strengthen the claims.

Similarly, the performance of Thor on Visium HD data imputation has not been compared with existing approaches such as Bin2Cell (Response Figure 2, Figure S26). A side-by-side comparison with these methods is essential to contextualize Thor's effectiveness.

3. Discussion of Limitations

The Discussion section should be expanded to explicitly acknowledge Thor's limitations. For instance, Thor exhibits reduced performance in capturing gradient patterns for some genes, likely due to challenges in detecting flat cell nuclei. The authors should comment on this issue and propose potential solutions. Additionally, Thor's limitations in multi-sample integration should be discussed, as this remains a key challenge in the field.

4. Color Scheme in Response Figure 4 (Figure S20)

The interpretation of Response Figure 4 is challenging because different color keys are used for results from the three methods. To improve readability, the same or similar colors should be used to represent the same cell type across methods.

5. T Cell Proliferation Inference (Original Comment #5)

The authors have not adequately addressed the concern regarding T cell proliferation inference. Additional supporting evidence would be necessary to strengthen this claim.

6. Validation of Single-Cell CNV Inference (Original Comment #6)

Comparing Thor's single-cell CNVs to spot-level CNVs is not sufficient, as both are inferred from spatial transcriptomics data. Neither method directly reflects genomic aneuploidy levels, making external validation crucial.

To demonstrate accuracy, paired genomic data should be used to evaluate the consistency of Thor's CNV predictions.

Alternatively, a FISH assay could be performed to validate a subset of genes in regions where single-cell CNV gains were detected but missed by spot-level CNV analysis.

7. TLS Analysis (Original Comment #7)

The TLS analysis lacks novelty and does not introduce significant new findings. This section should be further condensed to maintain focus on Thor's core contributions.

8. Robustness with Imperfect Histology Images (Original Comment #8)

The claim that Thor is robust to imperfect histology images (Response Figure 6, Figure S1b) remains unsubstantiated. A comparison with existing methods that handle histological variations is necessary to support this claim.

9. Granularity of Semi-Supervised Annotation (Original Comment #9)

The quantitative assessment of Thor's semi-supervised annotation focuses primarily on large-scale tissue structures (e.g., vessel, node, adipose, fibrosis). However, it lacks fine-grained annotations, which are essential for a comprehensive evaluation.

For example, other methods, such as MISO (Coleman et al., PMID: 39815104) and iStar, reveal significantly more detailed tissue structures. A comparative analysis would help demonstrate the advantages and potential shortcomings of Thor's annotation approach.

While the manuscript has improved, the concerns above must be addressed to ensure the robustness and novelty of Thor's contributions. I encourage the authors to provide additional comparisons, validations, and refinements to strengthen their work.

(Remarks on code availability)

Reviewer #2

(Remarks to the Author)

(Remarks on code availability)

Reviewer #3

(Remarks to the Author)

The author provided a good explanation and appropriate analysis for our comments, without further questions.

(Remarks on code availability)

Reviewer #4

(Remarks to the Author)

Thank you for your thorough revisions. Overall, the previous concerns have been well addressed in this version of the manuscript.

With the emergence of methods that cover whole genes with higher resolution, which are widely applied in this field, like VisiumHD, it is good to see that these advancements have been incorporated into this revision. However, it would be beneficial to provide detailed documentation on parameter optimization when making the related code available on GitHub. This will help ensure reproducibility and facilitate further research based on your work.

(Remarks on code availability)

The code is well described and example notebooks are good for understanding how to run this code.

As a future work, after publishing this code to the public, detailed parameter settings and installation methods for various users could be beneficial.

Version 2:

Reviewer comments:

Reviewer #1

(Remarks to the Author)

The authors have made efforts to address several of the comments from the previous review round, including expanding the limitations section and rephrasing certain text. However, not all prior concerns have been fully resolved. The authors should ensure that all remaining points are thoroughly addressed, either in the discussion or limitations sections—particularly with regard to the method's novelty and robustness in the context of recently published tools that were not included in the benchmarking of this study. Additionally, Figure 1 could be enhanced to more effectively illustrate the workflow and highlight the novelty of the proposed method. A clearer visual representation would aid in conveying the unique aspects and contributions of this work.

(Remarks on code availability)

Reviewer #2

(Remarks to the Author)

(Remarks on code availability)

We thank the reviewers for their detailed feedback and constructive suggestions. In response, we have substantially revised the manuscript to improve the robustness, clarity, and utility of Thor—a platform designed to infer cell-level transcriptomes from spot-level spatial transcriptomics data through integration with histological images. Below is a summary of our main changes:

- **Enhanced Robustness & Parameter Sensitivity:**

We expanded our evaluations to demonstrate that Thor maintains high accuracy under challenging conditions, including simulations with up to 40% missing cells and high technical dropouts (Figures S1b and S1d).

Comprehensive sensitivity analyses of key parameters—including graph construction (cell neighbors, principal components), Markov diffusion steps, and VAE latent dimensions—confirm that our method is robust over a broad range of settings (Figure S27).

- **Extensive Benchmarking & Quantitative Evaluations:**

We added multiple benchmarking experiments comparing Thor against state-of-the-art methods (e.g., CytoSPACE, RCTD, BayesSpace, TESLA, and simple nearest-spot/KNN approaches) using metrics such as CW-SSIM, RMSE (both pixel- and cell-centric), silhouette scores, and Calinski-Harabasz indices (Figures 2c, S1c, S7, and S20).

These analyses consistently demonstrate Thor’s superior performance in reconstructing cell-level gene expression and preserving spatial patterns.

- **Integration of High-Resolution Visium HD Data:**

We incorporated new analyses on an in-house bladder cancer sample sequenced on the Visium HD platform. Despite challenges such as technical noise in high-resolution bins, Thor successfully imputed cell-level gene expression that aligns more closely with pathology annotations (Figure S26).

This addition underscores Thor’s cost-effectiveness and its ability to complement emerging high-resolution spatial platforms.

- **Improved Cell Segmentation & Feature Extraction:**

We clarified that Thor extracts tile-based image features from regions surrounding detected nuclei to capture local tissue context.

In addition, we now integrate multiple state-of-the-art cell detection tools and provide an API (`thor.pp.image.WholeSlideImage.load_external_cell_features`) to allow users to incorporate externally derived morphological features.

- **Refined Semi-Supervised Tissue Annotation & Cell Type Resolution:**

We enhanced our semi-supervised annotation module by quantitatively comparing Thor-derived cell annotations with expert pathology labels on heart tissue samples (achieving accuracies of 0.83–0.99 across tissue types).

Further refinements include using established marker genes to infer neuron subtypes in mouse olfactory bulb, and distinguish tumor from normal epithelial cells, and monocytes from macrophages, thereby improving the biological interpretability of the inferred cell types.

- **Advanced Downstream Analyses:**

We expanded our CNV analysis by comparing cell-level CNV profiles inferred by Thor with traditional spot-level approaches. Our results show improved discrimination between tumor and normal regions (with higher F1 scores and Jaccard indices).

Additional modules now facilitate pathway enrichment, transcription factor activity

inference, and also cell–cell communication analysis (via an API interfacing with COMMOT), positioning Thor as a comprehensive downstream analysis platform.

- **Contextualization with Existing Tools & Terminological Clarifications:**

The revised Introduction now places Thor within the context of existing spatial analysis platforms (e.g., iStar, PROST, METI), clearly highlighting its unique advantage in delivering true cell-level resolution without reliance on external scRNA-seq references. Minor issues, such as the use of “transcriptomic” (instead of “genomic”) data and enhanced figure labeling and joint embedding visualizations (e.g., integration of Xenium data), have also been addressed.

- **Comprehensive Workflow & Validation:**

We provided detailed descriptions of Thor’s four-step inference process—from histology image preprocessing and cell detection, through construction of a cell–cell graph, to the final Markov diffusion for gene expression imputation.

Validation across 35 simulation and experimental datasets (including ISH, Stereo-seq, MERFISH, Xenium, and immunofluorescence) further attests to Thor’s accuracy and versatility.

With these extensive revisions and additions, we believe that the manuscript has been significantly strengthened, offering a robust, versatile, and user-friendly framework for integrated spatial transcriptomics and histology analyses at single-cell resolution. Please see our detailed point-by-point responses below.

Response to Reviewer #1:

In this study, the authors presented Thor, a spatial analysis tool that infers cell-resolution transcriptomes from spot-level spatial transcriptomics (ST) data and integrates them with histology. The tool shows potential for ST data analysis and could provide valuable insights. However, there are several major limitations in the current manuscript that need to be addressed. Further benchmarking, analysis, and validation studies are necessary. Please see the detailed comments below:

Response: Thank you for thoroughly reading our manuscript and accurately summarizing our approach. We truly appreciate that you highlight the potential of Thor for ST data analysis. In response to your concerns, we have added analyses on in-house Visium HD sample to show Thor improves the analysis of high-resolution spatial data and validated with pathology annotations, performed quantitative evaluation of Thor's semi-supervised annotation tool on a cohort of heart tissue samples, benchmarked Thor's performance with additional methods on high-resolution gene expression prediction, cell type annotation, and CNV analysis, evaluated Thor's robustness to suboptimal conditions of omics/histology imaging, and conducted parameter sensitivity analysis. We hope these analyses adequately address your concerns. In the following, we provide point-by-point response to address all the concerns.

Major comments:

Question 1. While it is advantageous to have single-cell resolution imputation from Visium, the cell graph inferred by Thor does not accurately reflect the true cell morphology, as the "cells" primarily represent cell nuclei. Importantly, the density of cell nuclei predicted by Thor is quite different from what is observed in the histology images, particularly in regions with a high cell density. The authors should address these discrepancies and provide an explanation or adjustments to improve the accuracy of the inferred cell morphology.

Response: Thank you for raising this important concern.

First, regarding the cell features from histology images, Thor extracts tile-based image features from the image patch centered at the cell nucleus centroid to capture the local environment surrounding the nucleus. These features are not limited to the nucleus itself but include the tissue context within the image tile, providing a comprehensive representation of the cell's local environment from histology. Tile-based feature extraction is a practical strategy widely adopted in histology image analysis by deep-learning models and pathology foundation models. It facilitates a wide range of downstream tasks, including cell segmentation, cell type annotation, and tumor microenvironment profiling (Bannon, Moen et al. Nat Methods 2021, Hu, Coleman et al. Cell Syst 2023, Chen, Ding et al. Nat Med 2024, Wang, Zhao et al. Nature 2024, Xu, Usuyama et al. Nature 2024, Zhang, Schroeder et al. Nat Biotechnol 2024). In Thor's framework, these tile-based features provide a cell-cell graph that accurately characterizes each cell's neighborhood. Our analyses demonstrated that clustering these tile-based histological features effectively segments tissues at cell resolution (Supplementary Note 1).

Second, as you point out, cell detection may be compromised in certain regions with high cell density or suboptimal imaging conditions. To address this concern, we evaluated the robustness of Thor's prediction under compromised conditions where up to 40% of cell were undetected. We found that Thor's accuracy remained robust (Figure S1b). This robustness is attributed to

Thor's use of a cell-cell graph that integrates histological and transcriptomic features, in addition to the nuclei centroids, mitigating the impact of missing cells.

Response Figure 1 (Figure S1b). Thor is robust with cell missouts. Normalized Root Mean Squared Error (NRMSE) measures the normalized deviation of Thor-inferred gene expression from the ground-truth gene expression. The performance of Thor remains accurate when there are 40% of cells missing randomly, compared to the case where all the cells are detected (0% missing cells). The middle line in the box plot, median; box boundary, interquartile range; whiskers, 5-95 percentile.

Lastly, recent advancements in state-of-the-art (SOTA) cell detection tools are transforming the challenge of accurate cell detection in high-density regions (Greenwald, Miller et al. Nat Biotechnol 2022, Pachitariu and Stringer Nat Methods 2022). With breakthroughs in deep-learning-based computer vision algorithms, including StarDist (Schmidt, Weigert et al. Medical Image Computing and Computer Assisted Intervention 2018), CellProfiler (Stirling, Swain-Bowden et al. BMC Bioinformatics 2021), Cellpose (Stringer, Wang et al. Nat Methods 2021), CellViT (Hörst, Rempe et al. Medical Image Analysis 2024), and MedSAM (Ma, He et al. Nat Commun 2024), accurately detecting cell nuclei has become increasingly viable. As an integrative spatial transcriptomics analysis platform, Thor incorporates multiple SOTA tools for cell segmentation and also provides an API `thor.pp.image.WholeSlideImage.load_external_cell_features` to support manually/strategically added missing cells, enabling enhanced flexibility and adaptability for diverse workflows. Thor is designed to stay aligned with ongoing advancements in cell segmentation technologies, ensuring that its methods remain cutting-edge.

In summary, Thor extracts tile-based image features that effectively capture the cellular environment, demonstrates robustness to cell detection errors, and is designed to integrate ongoing advancements in cell segmentation tools. These capabilities ensure reliable and accurate analysis, even in challenging high-density tissue regions. We have revised the manuscript with these discussions on Lines 669-690 and reorganized the Method section of nucleus segmentation and feature extraction on Lines 758-781 for clarity.

Question 2. With the introduction of the new Visium HD platform, what is the added value of Thor? Is there still a need for Thor, given the enhanced resolution of Visium HD? Furthermore, no Visium HD data is presented in this study. Including such data would help demonstrate Thor's relevance and utility in comparison to this newer platform.

Response: Thank you for this important question. While the Visium HD platform enhances spatial resolution to the sub-cellular level, Visium HD faces challenges such as high dropout rates, low gene coverage and misalignment of bins with cells, complicating biologically relevant interpretations (Kamel, Song et al. bioRxiv 2024, Polanski, Bartolome-Casado et al. Bioinformatics 2024), as well as lack of comprehensive analysis platforms tailored to the large-scale datasets.

During the revision, we added an in-house bladder cancer sample data sequenced on Visium HD. In our experiment, despite the high spatial resolution (2 μm square bins, and they were aggregated into 8 μm square bins for analyses as recommended by 10x Genomics), the 8 μm bin data showed high technical noise. For instance, immune cell marker PTPRC is sparsely expressed in densely packed lymphoid regions (Figure S26, ROI 1) and the urothelium-associated gene SPINK1 showed high expression in non-tissue areas (ROI 3, arrow). Although Thor was developed before Visium HD was made publicly available, we adapted Thor (we provide a Jupyter Notebook as a tutorial) to infer cell-level gene expression through integration of the HD data with histology. Thor-imputed cell-level expression revealed stronger spatial patterns which align with histology annotations (Figure S26).

Response Figure 2 (Figure S26). Thor imputes Visium HD data and reconstructs gene expression patterns that align with pathology annotations in a bladder cancer sample. Pathology annotations (left) highlight immune cells (ROI 1), invasive carcinoma (ROI 2), and a tumor fragment (ROI 3). Gene expression patterns inferred by Thor (middle) and from Visium HD (right). The orange arrow points to a region with no cell.

Beyond improving data quality, Thor functions as a comprehensive downstream analysis platform for large-scale spatial data. Visium HD generates a massive number of spots/bins/cells (e.g. ~5 million 2 μm square bins on a 6.5 mm x 6.5 mm slide), posing computational and visualization challenges. Thor adapts state-of-the-art analysis tools and supports streamlined

analyses including clustering, differential gene expression, pathway enrichment analysis, transcription factor activity inference, CNV analysis, and semi-supervised annotation. Thor's interactive visualization tool, Mjolnir, also supports investigation of modalities (e.g. histology images, gene expression) as vivid gigapixel images. Leveraging image-tiling technologies akin to Google Maps, Mjolnir reduces memory burden and can be run on a standard laptop.

Additionally, Thor remains cost-effective by enabling single-cell resolution analysis on more affordable standard Visium data, which is roughly 70% cheaper than Visium HD. This makes Thor attractive for laboratories facing resource limitations or those wishing to reanalyze existing data generated from standard or legacy ST platforms.

In summary, Thor supplements Visium HD's fine-resolution data by imputing gene expression, providing deeper single-cell insights, and offering an extensive analytic suite that eases the computational and visualization burden inherent to high-resolution data. Our new Visium HD analysis (Lines 538-557) and discussion (Lines 692-707) demonstrate Thor's relevance for these next-generation platforms, while also highlighting its ongoing utility for cost-effective and large-scale ST investigations.

Question 3. In Fig. 2d, ROI5 (top row), Thor does not appear to maintain the spatial gradient pattern observed in the Xenium data and iStar results. For example, the outer layer of cells showed higher expression in both the Xenium data and iStar results, which was not captured by Thor. The authors should address this inconsistency and discuss its potential impact on the accuracy of the analysis.

Response: Thank you for your detailed investigation of the spatial profiles of genes and your valuable feedback. We acknowledge that iStar performed better than Thor in capturing gradient patterns for some genes, such as *DST* expression in the peripheral area in ROI5. Thor's reduced performance in these areas arose from challenges in detecting flat cell nuclei. While we have shown that misdetection of cells does not degrade Thor's performance on the detected cells (in our Response to Question 1), it limits the analysis of such regions. Incorporating higher precision imaging techniques, such as the DAPI imaging used in the Xenium data cell detection, could help address this issue. However, it is important to note that this limitation is localized and does not systematically affect Thor's predictions across the tissue.

Despite this localized limitation, Thor demonstrated best overall performance across all genes and the entire tissue among the four compared tools. To provide a comprehensive comparison, we calculated SSIM and RMSE metrics for all genes against the Xenium data. These analyses demonstrate that Thor significantly outperformed iStar in predicting spatial gene expression across the tissue (Figures 2c and S7). We have included the new analysis (Lines 258-274, and 282-283) and discussions of Thor's limitations (Lines 713-716) in the revised manuscript.

Response Figure 3 (Figures 2c, S7a-b). Quantitative comparison between Thor and ST spatial resolution-enhancement tools. (a) Using image-level metrics (b) Using cell-level metrics. One-sided Mann-Whitney tests were performed between Thor and other two best-performing tools (iStar and BayesSpace).

Question 4. It would be helpful to benchmark Thor against other methods, such as CytoSPACE and other single-cell resolution deconvolution tools, to evaluate its performance. This comparison would provide a clearer understanding of Thor's strengths and limitations in producing single-cell resolution results.

Response: Thank you for suggesting additional benchmarks to evaluate Thor's performance on cell type annotation against existing tools. In response to your feedback, we have included a comparison of Thor with CytoSPACE and RCTD (Figure S20). Thor demonstrated comparable performance to these single-cell deconvolution tools, as shown by the spatial distribution of cell types, such as epithelial cells, fibroblasts, monocytes/macrophages.

Response Figure 4 (Figure S20). Computationally inferred spatial distribution of cell types in the DCIS data by (a) Thor, (b) CytoSPACE, and (c) RCTD. For Thor, the annotations were assigned by Cell-ID based on the inferred cell-level gene expression; for CytoSPACE and RCTD, the per-spot proportions of cell types were inferred based on an external scRNA-seq data.

Thor presents two notable advantages over those spatial deconvolution tools. First, Thor does not require a matching scRNA-seq reference data; instead, Thor infers cell-level gene expression by integrating the underused paired full-resolution histology image and the spatial transcriptomics. Second, while other tools (e.g., CytoSPACE) project single cells onto ST spots, they cannot resolve the precise arrangement of those cells within each spot. Thor, by contrast, assigns gene expression profiles to the actual cells detected on the tissue, providing truly cell-level spatial resolution for higher-fidelity downstream analyses of tissue architecture, cell-cell interactions, and spatial heterogeneity. For example, in our analysis of the post-LVAD patient heart tissue, Thor identified key gene signatures enriched in thin vessels composed of a layer of cells. Such

precise spatial analysis may not be accomplished by deconvolution tools as they miss the spatial arrangement of the multiple cell layers covered by a Visium spot. One limitation of Thor is that it requires high-quality histology images (typically 0.25-0.5 μm per pixel in resolution), which may not always be accessible in Visium experiments. We have added the results (Lines 442-448) as well as discussions of this limitation (Lines 709-713) in the revised manuscript.

Question 5. The annotation of the fibrotic area in Fig. 3b does not seem to add significant value, as these areas are already visible from the H&E images. Additionally, this annotation does not provide insights into the phenotypic and functional states of the fibroblasts. Furthermore, how was T cell proliferation inferred in this analysis? Were the authors able to validate the T cell proliferation signals? Providing more details and validation would strengthen the findings.

Response: Thank you for these valuable comments. While we agree that fibrotic regions are often visible in H&E images, Thor's semi-supervised annotation integrates both morphological and transcriptomic data to yield a more nuanced and functionally informed delineation of fibrotic areas. Even when tissue morphology suggests fibrosis, early-stage or subtle fibrotic changes may be missed by conventional H&E-based inspection alone. Thor's human-in-the-loop design allows researchers to annotate small representative regions using fibroblast marker expression and tissue morphology at the full-resolution H&E image of about 0.25 microns per pixel, then apply these transcriptome & morphology-guided annotations across the entire tissue section. This integrated approach not only refines the fibrotic boundaries but also connects them to gene-expression patterns, offering deeper molecular insights.

In the subsequent analyses (Figures 3c-e and S18), we illustrate how these refined fibrotic annotations help reveal enriched pathways and transcriptional factor activities indicative of distinct cell states, including proliferation, collagen fibril organization, and inflammatory signaling. Thor's added value lies in integrating morphological inspection with transcriptomic evidence, thus revealing more nuanced fibrotic subregions and connecting histological observations to functional pathways. We have now rewritten this section to emphasize that semi-supervised annotation is not just about highlighting what is already visible in H&E, but also about uncovering molecular and functional details of the fibrotic niches (Lines 331-337, 363-387).

Regarding T cell proliferation activity, as part of an unbiased pathway screening, we applied over-representation analysis (ORA) across all Gene Ontology (GO) terms for the top-expressed genes in each cell. Among the pathways that emerged, "positive regulation of T cell proliferation" (GO:0042102) ranked highly in fibrotic regions—alongside other enriched terms like inflammatory response and fibroblast proliferation. To clarify our interpretation, we have updated both Figure 3b and the manuscript text to specify that these enrichment signals represent GO pathway scores rather than direct measurements of proliferative activity. ORA has been widely used in pathway-level signal assessments (Khatri, Sirota et al. PLoS Comput Biol 2012, Kuppe, Ramirez Flores et al. Nature 2022). While the tissue samples are unavailable (the data was downloaded from public resource) for direct validation, our finding aligns with previous evidence of T cell proliferation and fibroblast-mediated T cell activation in cardiac settings (Ramos, Hofmann et al. J Mol Cell Cardiol 2016, Learmonth, Corker et al. Am J Physiol Heart

Circ Physiol 2023). We have included these methodological details in the revised Methods section (Lines 1022-1030) and results (Lines 363-371) accordingly.

Question 6. The section on CNV analysis does not appear to be novel and add significant value, as there are already existing methods available for CNV analysis. The authors should clarify how their approach differs from or improves upon these established methods. How does Thor's spatial InferCNV analysis compare to spot-level spatial InferCNV, and what advantages does Thor offer?

Response: Thank you for the comment and suggestion. The unique advantage of Thor's CNV analysis is providing single cell level CNV analysis based on ST data. In contrast, spot-level CNV analysis may obscure heterogeneity in regions containing a mixture of aneuploid and diploid cells.

Following your suggestion, we compared CNV profiles on Thor-inferred single-cell data and the spot-level data to evaluate Thor's advantage. In the DCIS dataset, Thor successfully identified genome-wide CNV profiles, distinguishing tumor cells from normal cells with an F1 score of 0.78 and a Jaccard index of 0.64 (Figure 5g). The result closely aligned with pathology-annotated tumor regions and outperformed spot-level CNV analyses (F1 score: 0.73, Jaccard index: 0.58). We have added the analysis on Lines 490-497.

Response Figure 5 (Figure 5a, g). Spatial distribution of aneuploid cells. The annotation of eighteen major tumor regions (T1-T18) in the DCIS tissue is adapted from the annotation by pathology experts (Agoko NV, Belgium, left). Aneuploid (tumor) and diploid (non-tumor) regions inferred from Thor's *in silico* cell-level transcriptome and from spot-level data.

Question 7. The analysis of tertiary lymphoid structures (TLS) appears quite superficial and does not add value beyond existing methods. It also does not seem to perform well in accurately capturing TLSs. Proper pathology annotation and multi-plex staining are necessary to define TLS, its maturation status. The structure shown in Figure 6c is more likely a lymphoid aggregate (LA).

Response: Thank you for your feedback regarding the analysis of tumor region heterogeneity in DCIS and the use of TLS-related terminology. We apologize for any confusion caused by our descriptions in the manuscript. To clarify, our study does not aim to identify TLS or assess the maturation status of TLS. Instead, we employed the TLS score as a well-established metric to investigate the immune activity of tumor regions. The TLS score for each cell was calculated by comparing the average expression of a defined set of signature genes, including markers of immune cells, with the average expression of randomized genes. This methodology has been

widely validated and used in spatial transcriptomics analyses (Meylan, Petitprez et al. Immunity 2022, Wei, He et al. Nat Biotechnol 2022).

To evaluate tumor region heterogeneity, we ranked tumor regions based on the median TLS scores of cells located in the respective tumor region up to one spot-size outward from the region boundary. This ranking enabled a comparative analysis of immune activity between regions. As suggested in your Minor Comment 15, we examined functional distinctions and interactions between tumor regions and their immediate peritumoral neighbors. T7 was enriched for pathways linked to immune responses and T cell co-stimulation, whereas T15 was enriched for tumor-related pathways such as hypoxia response and cell adhesion. Those results provided valuable insights into the functional heterogeneity among tumor regions, supporting a refined understanding of immune-tumor interactions in DCIS. While we used the TLS score as a proxy for immune activity, we did not investigate TLS structures directly or aimed at differentiating TLS from lymphoid aggregates. To avoid confusion, we have revised the manuscript to clarify our focus on tumor region heterogeneity rather than TLS identification (Lines 506-514) and to include additional tumor-immune functional analysis (Lines 521-525).

Question 8. How does Thor's performance depend on the quality of histology images and transcriptomics data? It is suggested that the authors demonstrate Thor's performance in scenarios where one of the two modalities is suboptimal.

Response: Thank you for the valuable suggestion. As suggested, we systematically assessed Thor's performance under suboptimal conditions of histology images or transcriptomics, as outlined below.

For suboptimal histology images, two common issues were identified that could potentially affect performance: (i) missed detection of cell nuclei, particularly in out-of-focus or densely packed regions, and (ii) erroneous cell-cell connections due to poor histological features. Despite these challenges, our evaluation demonstrates that Thor maintains high prediction accuracy even under substantial cell loss and misrepresented cell-cell connections (up to 40%), as shown in Figure S1b. This highlights the method's robustness in handling imperfect histology images.

Response Figure 6 (Figure S1b). Thor is robust with cell missouts or erroneous connections caused by suboptimal histology images. NRMSE provides a quantitative measure of the normalized deviation of Thor-inferred gene expression from the ground-truth gene expression. The performance of Thor remains accurate when there are 40% of cells missing or 40% of cells with randomized connections. The dash lines mark the baseline level when raw spot data are

mapped to nearest cells. The middle line in the box plot, median; box boundary, interquartile range; whiskers, 5-95 percentile.

Regarding suboptimal transcriptomics data, we conducted a series of simulations with varying dropout rates (ranging from 5% to 60%) to assess how Thor performs under different levels of data sparsity. We employed silhouette coefficients to quantitatively evaluate how well Thor preserves cluster structure and separation. In our analysis (Figure S1d), Thor demonstrated consistent performance, with silhouette coefficients remaining in the 0.5–0.6 range under moderate dropout conditions (compared to a significant decline in performance from the ground truth data). Even under high dropout conditions (>40%), Thor’s silhouette coefficients remained substantially higher than those of competing methods such as KNN smoothing and BayesSpace, showcasing its ability to recover cluster separation and handle data sparsity effectively.

Response Figure 7 (Figure S1d). Thor imputes gene expression with technical dropouts and recovers cluster separation. The error bars for the mean Silhouette coefficients are omitted as they are too small to visualize. The colors in the PCA plots represent the ground truth cell type information. “Drop %” in the table is calculated as the ratio of zeros in the count matrix of the simulated scRNA-seq data.

In summary, our analysis confirms that Thor performs robustly under a wide range of suboptimal conditions in both histology and transcriptomics modalities. We have incorporated these findings

into the revised manuscript (Lines 168-177 and Lines 204-216) to provide a more comprehensive assessment of Thor's capabilities in challenging data scenarios.

Question 9. How does the semi-annotation function align with spot-level pathology annotations? It is recommended that the authors conduct a quantitative analysis across a cohort of samples to evaluate this alignment. Additionally, how does Thor handle other key histological features in tissues, such as key microstructures within the tissues, other cell components such as macrophages, neutrophils, plasma cells, and cancer cell histology? Providing insights into these aspects would greatly enhance the utility and applicability of the tool.

Response: Thank you for your constructive feedback. We have now incorporated additional analyses to quantitatively evaluate Thor's semi-supervised annotation function. In the analysis of the DCIS benchmark dataset, only region-level pathology annotations were available, so we calculated the accuracy score between Thor-annotated tumor cells and region-level pathology labels (accuracy: 0.83).

To quantitatively evaluate Thor's semi-supervised annotation more broadly, we downloaded a cohort of heart tissue samples from the literature (Kanemaru, Cranley et al. Nature 2023). These samples included high-resolution H&E images, high-quality ST data, and precise spot-level expert-annotations for key tissue types in heart including vessel, node, adipose, and fibrosis. Using Thor's semi-supervised annotation tool, we observed strong alignment with the expert annotations, with accuracy ranging from 0.94 to 0.99 for vessels, 0.92 to 0.98 for nodes, 0.84 to 0.92 for adipose, and 0.92 to 0.93 for fibrosis (Figure S12).

Regarding additional histological features and diverse cell types such as macrophages, neutrophils, plasma cells, and various cancer cell morphologies, while Thor's default tile-based features already capture general morphological context, we recognize that certain cell types or microstructures may require more specialized descriptors. To address this, we have developed an API (`thor.pp.image.WholeSlideImage.load_external_cell_features`) that supports seamless inclusion of morphological features generated by external tools (e.g., CellProfiler or CellViT). Researchers can extract customized metrics, such as cell shape, texture, or intensity profiles and feed to Thor, effectively augmenting or replacing the default tile-based features. We provide detailed documentation on accepted feature formats and usage instructions in our code repository and on our website (currently as supplementary materials for your review: website_htmls/_autosummary/thor.pp.WholeSlideImage.html). This flexible design allows Thor to accommodate a wide spectrum of histological analyses and cellular phenotyping tasks, ensuring that users can tailor the platform to their unique research objectives. We have added the new analysis (Lines 339-348) and discussion (Lines 682-690) in the revised manuscript.

Response Figure 8 (Figure S12) Quantitative assessment of Thor's semi-supervised annotation using spot-level expert annotations. (left panel) H&E image, (second panel) spot-level pathology annotations, (third panel) Thor's cell-level annotations, and (right panel) Thor's cell-level annotation mapped to the spots using majority voting (>50%) of representative samples in each tissue type including vessel, node, adipose, and fibrosis. The metrics are calculated based on spot-level annotations.

Question 10. Benchmarking is missing for both the TLS and CNV analyses. The authors should provide a more thorough evaluation and comparison with existing methods.

Response: Thanks for this important suggestion. As suggested, we have benchmarked Thor's CNV inference (using CopyKAT) by comparing cell-level results from Thor-imputed data to spot-level CNV analyses. In the DCIS dataset, Thor successfully identified genome-wide CNV profiles, accurately distinguishing tumor cells from normal cells with an F1 score of 0.78 and a Jaccard index of 0.64. The result closely aligned with pathology-annotated tumor regions and outperformed spot-level approach (F1 score: 0.73, Jaccard index: 0.58). These results highlight

Thor's advantage in detecting CNVs from cell-level data. However, for TLS-related analyses, our study does not seek to identify or assess TLS structures or their maturation status—a point we clarified in our response to Question 7. As such, we have not conducted a benchmarking analysis specific to TLS detection.

It is also important to emphasize that in addition to infer cell-level gene expression, Thor's primary strength lies in offering a unified, user-friendly platform rather than introducing new algorithms for each downstream task. By seamlessly wrapping established methods within a single interface, Thor minimizes data format conversions and technical overhead, thus streamlining multi-modal spatial omics analyses. For instance, Thor integrates the widely validated tool CopyKAT to perform transcriptomics-based CNV calls. We believe these design considerations address your concerns with the validation of downstream analysis modules while underscoring Thor's utility as an accessible and integrative solution. We have included the benchmark of CNV analysis on Lines 490-497 in the revised manuscript.

Question 11. The analysis in Figures 5f and 5g is quite superficial, as checking gene expression and signature scores can be done on any spatial platform. The authors should emphasize the unique strengths of their algorithm in this analysis.

Response: Thank you for this constructive feedback. To address this concern, we have reorganized Figure 5 to more clearly emphasize the two main advantages of Thor: (1) its unique approach to inferring cell-level gene expression through integration of histological and transcriptomic data, and (2) its streamlined framework for downstream spatial omics analyses. Therefore, we moved the basic demonstration of gene and signature scores (previously in Figures 5f and 5g) to the Supplementary figures (now Figures S23b and S24).

We have now reorganized the result section (Lines 430-504) to demonstrate how Thor improved cell type annotation and enabled interactive exploration of gene profiles at the gigapixel histological images, search of similar cells in the combined histological and transcriptomic space, and CNV detection at cell resolution. By doing so, we emphasize the advantages of added resolution gained from the histology-guided imputation and the unified platform facilitating multi-layer investigation in the DCIS dataset.

Question 12. Additionally, it is unclear how much training data is required for Thor to achieve reliable results, and how is the accuracy of its predictions evaluated? The authors should provide details on the training data size and the methods used to assess prediction accuracy to ensure the robustness of the tool.

Response: Thank you for the suggestions. Thor does not require training for the inference of cell-level gene expression. Instead, it operates per histology slide through a four-step workflow. (i) Preprocess the full resolution histology image and obtain the cell nuclei and image features surrounding the nuclei; meanwhile, preprocess the spot-level spatial transcriptomics data and map the spot-level data to the cell level using nearest neighbors mapping method. (ii) Construct a cell-cell graph based on the cell-level image features, cell positions, and the transcriptomic features. (iii) Create a Markov transition matrix based on the cell-cell graph, representing the probabilities of transitioning from a cell to every other cell (iv) Perform Markov diffusion using the transition matrix per slide to acquire histology-refined gene expression.

We assessed the accuracy of the Thor-inferred cell-level gene expressions on 35 simulation data and experimental data including ISH, Stereo-seq, MERFISH, Xenium, and immunofluorescent staining data. We used metrics including root mean squared error (RMSE), complex-wave structure similarity index measure (CW-SSIM), and Pearson correlation coefficient. for gene expression. For example, in the human breast cancer data downloaded from 10x Genomics, the Thor-inferred spatial ST aligned well with the coupled Xenium data in cell types and gene expressions; in the mouse olfactory bulb (MOB) dataset, the Thor-inferred spatial expression profiles of genes matched the ISH staining and Stereo-seq data; in the mouse brain data, the Thor-inferred cell-level gene expressions agreed with MERFISH data at both single-gene and cluster levels.

To evaluate the model robustness, we have conducted new sensitivity analyses on important model parameters including the number of neighbors, number of principal components of the transcriptomic features, and number of diffusion steps. We found our feature-preserving Markov diffusion model remained robust across varying parameter settings (Figure S27). We have clarified the workflow of Thor inference (Lines 742-756) and the description of evaluation metrics for gene expression prediction (Lines 1171-1190), and added sensitivity analyses (Lines 559-595) in the revised manuscript.

Response Figure 9 (Figure S27a-b). Sensitivity analyses of Thor inference on (a) graph construction parameters, and (b) the diffusion steps. Mean Pearson correlation coefficients between every pair of parameter settings for all genes are plotted. Spatial distributions of a representative gene *Penk* is provided to illustrate the influence of diffusion steps on Thor's inference.

Question 13. The data presented in Figure 5 require validation to strengthen the conclusions drawn from the analysis. The authors should provide additional experimental or independent validation to support these findings.

Response: Thank you for this comment. We understand the concern about the validation. However, in this case study, we aim to showcase Thor's functions on a well-validated benchmark dataset from 10x Genomics, which has high data quality and was widely used as benchmarks in many spatial transcriptomics analysis studies (Vahid, Brown et al. Nat Biotechnol 2023, Zhu, Kubota et al. Nat Commun 2024). We are not aiming to provide biological findings or conclusions, but to showcase how Thor works as a comprehensive analysis platform of histology and spatial transcriptomics data. All the analyses present in Figure 5 are based on the outcomes of applying well-established methods, including differential gene expression (SCANPY), GO pathway enrichment analysis (decoupler), and CNV analyses (CopyKAT) to the Thor-inferred gene expression. The prediction of cell-level gene expression has been evaluated on 35 simulation datasets and experimental data including ISH, Stereo-seq, MERFISH, Xenium, and our in-house immunofluorescent staining data, as elaborated in our response to Question 12.

Furthermore, we have added benchmarks of baseline analyses including CNV analysis on the spot-level data, and benchmarks of other tools including with image-based tumor annotation tool CLAM, cell type deconvolution tools CytoSPACE and RCTD. To avoid any potential overstatement on the biological findings, we have rewritten the result section regarding Figure 5 for clarification on Lines 430-504.

Minor comments:

Comment 1. the introduction and the first section of the results, the authors frequently use the term "genomic data," whereas the data utilized in the study is actually transcriptomic. It would be more accurate to adjust these descriptions to "transcriptomic data" to avoid potential confusion.

Response: Thank you for the suggestion. We have changed “genomic data” to “transcriptomic data” in all the instances.

Comment 2. It would be helpful to assess Thor's performance with varying levels of dropouts in Figure S1. This evaluation will help readers better understand the robustness of the algorithm and the extent to which it can handle data dropouts.

Response: Thank you for the great suggestion. As suggested, we have expanded our evaluation by including additional levels of technical dropouts. We assessed Thor's performance under low (<15% zeros), medium (15%-40% zeros), and high noise (>40% zeros) conditions. We then measured cluster separations in principal component analysis (PCA) space using silhouette coefficients. As shown in the PCA plots (Figure S1d), introducing dropouts severely diminished cluster separations in the ground truth data, with silhouette coefficients reduced from 0.8 to near 0. In contrast, Thor-imputed data maintained the silhouette coefficient to above 0.7-0.8 in the low-dropout regime, outperforming the KNN smoothing method and BayesSpace. When dropout ratios rose to the moderate regime, where the ground truth data's silhouette coefficients declined to 0.1-0.4, Thor-imputed data recovered the cluster separation successfully (silhouette coefficients 0.5-0.6). Even under high-dropout conditions (>40%), Thor's scores remained substantially above those of KNN smoothing and BayesSpace. We have added the analysis in the revised manuscript on Lines 204-216.

Response Figure 10 (Figure S1d). Thor robustly imputes gene expression with technical dropouts. The error bars for the mean Silhouette coefficients are omitted as they are too small to visualize. The colors in the PCA plots represent the ground truth cell type information. “drop %” in the table is calculated as the ratio of zeros in the count matrix.

Comment 3. Please provide quantitative measurements of the correlation between the imputed molecule expression and the MERFISH ground truth in their spatial locations for Figures S4b and S5b.

Response: Thanks for this suggestion. Following your suggestion, we have added the Pearson correlation coefficients between the Thor-inferred molecule expression and the MERFISH ground truth data in Figures S4b and S5b, which ranged from 0.32 to 0.66.

Comment 4. It would be beneficial to assess the relationship between cell type complexity within a spot and imputation accuracy. Specifically, does the accuracy decrease with increasing spot complexity, and if so, to what extent?

Response: Thanks for raising this important question. Yes, the accuracy of Thor decreases with increasing spot complexity to some extent. To evaluate Thor's performance with varying levels of spot complexity, we plotted the mean absolute error (MAE) of each cell against spot heterogeneity on simulation data, where the spot heterogeneity was quantified by the Shannon entropy of cell type proportions in the ground truth data. As expected, with increasing spot heterogeneity, nearest-spot mapping exhibited a sharp increase in MAE. In contrast, Thor effectively recovered gene expression levels in both low (Figure S3c, A) and high (Figure S3c, B and C) heterogeneity spots. Specifically, Thor accurately imputed gene expression for the majority of cells within highly heterogeneous spots (Figure S3c, B). For a smaller subset of cells

(Figure S3c, C), the MAE increased slightly but remained substantially lower than that with nearest-spot mapping.

Response Figure 11 (Figure S3c). Thor demonstrates robust performance across varying levels of spot heterogeneity. The mean absolute error (MAE) is calculated between Thor-predicted and ground-truth gene expression levels. A, B, C mark one low-heterogeneity, and two high-heterogeneity clusters of Thor data.

Thor's accurate imputation is attributed to integration of spatial cell positions, histological features, and transcriptomic profiles in a cell-cell graph. We have added the analysis in the revised manuscript (Lines 197-203).

Comment 5. Please include quantitative measurements, such as the silhouette score, for Figure S4c to provide a more thorough analysis.

Response: Thanks for this suggestion. Following your suggestion, we have added silhouette score and Calinski-Harabasz index (Figure S4d) for quantitative evaluation of cluster separation. Thor outperformed BayesSpace and nearest spot mapping in separating the cell type clusters. We have included the analyses in the revised manuscript on Lines 239-243.

Response Figure 12 (Figures S4d). Thor recovers cell-level gene expression in mouse brain. The Silhouette coefficient and Calinski-Harabasz index were calculated based on the embeddings and the ground truth cell annotations. For BayesSpace, as the native output was sub-spot level gene expression, we added comparison for both the sub-spot level and cell-level (mapping the closest sub-spots to the cells).

Comment 6. For Figure 2b, it would be useful to show cells from both Xenium and Thor in the joint embedding and assess the similarity of the same cell types defined from both modalities.

Response: We totally agree with suggestion. In response, we have integrated Thor and Xenium cells (Figure S6). We used the tool harmony with all the genes from Xenium panel for integration. The same cell types from both modalities co-localize in the UMAP embedding of the integrated data. This result has been included in the revised manuscript on Lines 253-256.

Response Figure 13 (Figure S6). UMAP embeddings of the Thor and Xenium integrated cells colored by modalities (left) and by cell types (right).

Comment 7.1 The benchmarking in Figure 2 is too limited, as only one published tool is tested. Please benchmark Thor against other existing tools such as CytoSPACE to give readers a comprehensive view.

Response: Thanks for this suggestion. We have conducted a comprehensive benchmark by comparing Thor with additional tools BayesSpace and TESLA (as requested by Reviewer #3), both enhancing spatial resolution without dependence on external scRNA-seq data. We did not include CytoSPACE in this benchmark because it may be not an entirely fair comparison, as CytoSPACE requires additional reference scRNA-seq data (nevertheless, we included benchmark of CytoSPACE in cell type deconvolution in the DCIS dataset). Our results demonstrate that overall Thor significantly outperformed existing tools in the breast cancer dataset, using the Xenium data as a reference. We have included the results in the revised manuscript on Lines 258-274.

Response Figure 14 (Figures 2c and S7a-b). Quantitative comparison between Thor and ST spatial resolution-enhancement tools. (a) Using image-level metrics and (b) Using cell-level metrics. One-sided Mann-Whitney tests were performed between Thor and other two best-performing tools. For the box plots, the middle line in the box plot, median; box boundary, interquartile range; whiskers, 5-95 percentile.

Comment 7.2 For Figures S4 and S5, additional benchmarking with tools like iStar would be helpful.

Response: Thanks for this suggestion. Actually, we have compared Thor with iStar in the breast cancer case study (using the Xenium data as ground truth). We didn't compare Thor with iStar in this case study because iStar takes an H&E image as input, while this MERFISH dataset only provided a DAPI image. To further benchmark Thor, we have added comparison with BayesSpace (Figure S4 for metrics and S5 for representative gene profiles from all tools). Thor outperformed BayesSpace and nearest spot mapping in separating the cell type clusters. We have included the analyses on Lines 239-243.

Response Figure 15 (Figures S4d). Thor recovers cell-level gene expression in mouse brain. The Silhouette coefficient and Calinski-Harabasz index were calculated based on the embeddings and the ground truth cell annotations. For BayesSpace, as the native output was sub-spot level gene expression, we added comparison for both the sub-spot level and cell-level (mapping the closest sub-spots to the cells).

Comment 8. In the section "Thor unveils refined tissue structure in mouse olfactory bulb," it would be beneficial to describe the tissue structures in combination with the H&E image, including the layers and cell types present. This will help readers better understand the complexity and results shown in Figures S8b-c.

Response: Thank you for the suggestion. We have added cell type information in the revised Figure S10b (formerly Figure S8b). We annotated the mouse olfactory bulb by using the cell type markers from an external MOB scRNA-seq data (Tepe, Hill et al. Cell Reports 2018).

Response Figure 16 (Figure S10b). The spatial distribution of neuron subtypes in the mouse olfactory bulb predicted from Thor-inferred single-cell gene expression. OSN: Olfactory sensory neuron, PGC: Periglomerular cell, GC: Granule cell, M/TC: Mitral/Tufted cell.

Thor successfully resolved finer cell subtypes. For example, Thor distinguished granule cells into GC-1 (mainly localized in the internal plexiform layer) and GC-2 (predominantly in the granule cell layer), while also identifying two mitral cell subtypes, M/TC-1 and M/TC-2, with M/TC-2

concentrated in the mitral layer and M/TC-1 extending into the glomerular layer. These results demonstrate Thor's capability to refine cell type classification by leveraging histology and spatial transcriptomic data. We have enriched the description by including both MOB layers as well as neuron subtypes in the manuscript on Lines 307-315 and 323.

Comment 9. Please label the regions in Figure 4a with numbers for clarity.

Response: Thank you for the suggestion. We have added labels in Figure 4a.

Comment 10. It would be useful to benchmark Thor against more spatial deconvolution methods, such as RCTD in Figure S16.

Response: Thank you for the suggestion to include more spatial deconvolution methods in our benchmarking. This question is related to Question 4. We have added comparisons with CytoSPACE and RCTD (Figure S20, previously S16). Thor's cell-type assignments aligned closely with those derived from spatial deconvolution tools, as well as morphological expectations (e.g. epithelial cells and fibroblasts).

Thor presents two notable advantages over those spatial deconvolution tools. First, Thor does not require a matching scRNA-seq reference data; instead, Thor infers cell-level gene expression by integrating the underused paired full-resolution histology image and the spatial transcriptomics. Second, while other tools (e.g., CytoSPACE) project single cells onto ST spots, they cannot resolve the precise arrangement of those cells within each spot. Thor, by contrast, assigns gene expression profiles to the actual cells detected on the tissue, providing truly cell-level spatial resolution for higher-fidelity downstream analyses of tissue architecture, cell-cell interactions, and spatial heterogeneity. One limitation of Thor is that it requires high-quality histology images (typically 0.25-0.5 μm per pixel in resolution), which may not always be accessible in Visium experiments. We have added the results (Lines 442-448) as well as a discussion of this limitation (Lines 709-713) in the revised manuscript.

Response Figure 17 (Figure S20). Computationally inferred spatial distribution of cell types in the DCIS data by (a) Thor, (b) CytoSPACE, and (c) RCTD. For Thor, the annotations were assigned by Cell-ID based on the inferred cell-level gene expression; for CytoSPACE and RCTD, the per-spot proportions of cell types were inferred based on an external scRNA-seq data.

Comment 11. For Figure 5c, please provide a more comprehensive marker gene list for each cell type to assess the accuracy of the defined cell types. Additionally, distinguish between normal and tumor epithelial cells, allowing readers to check the correlation with the pathologically annotated tumor regions in Figure 5a. Showing classic tumor/epithelial markers (e.g., EPCAM, CDH1) in parallel with Figure 5d would also be helpful for comparison.

Response: Thank you for the suggestion. As suggested, we have provided the signature genes of each cell type in Supplementary Table S3 for reference. We used an established tool Cell-ID (Cortal, Martignetti et al. Nat Biotechnol 2021) to transfer the cell type information from an annotated reference scRNA-seq data of breast cancer to our inferred *in silico* cells. For clarity, we have added more details on how the cells were annotated in the revised manuscript in the Methods section under subtitle “Cell type annotation by Cell-ID”.

Furthermore, we used *EPCAM* and *CDH1* as markers to further distinguish normal epithelial and tumor epithelial cells (revised Figure 5c and Figure S20a). Specifically, the mean and standard deviation of the *EPCAM* and *CDH1* expression levels were applied to separate the normal and tumor epithelial cells. Expression profile of an additional marker gene *TACSTD2* is also provided for additional reference.

Response Figure 18 (Figures 5c and S20a). Refined cell type annotations by using marker gene expression to separate normal and cancer epithelial cells.

Comment 12. The results in Figure 5e indicate that tumor cell heterogeneity and their microenvironment might influence predictions. Since manual curation of cells is often impractical in high data volumes, the authors should assess how frequently such issues occur across different tumor samples. Additionally, the identification of adipose tissue in CLAM seems problematic, and a thorough assessment of potential circumstances where this mistake could occur would help inform users on proper tool selection and usage.

Response: We appreciate your insight into the sensitivity of our semi-supervised annotation tool. A key advantage of Thor’s semi-supervised search tool is its design to allow users to quickly and precisely select similar cells with user-curated cells, thus incorporating user’s knowledge of cell morphology and specific molecular markers. However, we acknowledge that manual curation becomes challenging with high data volumes, and Thor currently does not support search across multiple tissue samples due to batch effects in both histology and transcriptomics. We have included discussion of these limitations of Thor’s semi-supervised annotation in its scalability for extensive datasets on Lines 716-720.

Regarding CLAM’s false identification of adipose tissue as high-attention regions, we went through the original publication and the code repository thoroughly. CLAM outputs tile-level attention scores to indicate the relative importance of the tiles in determining the slide-level tumor type prediction. The authors of CLAM stated in the original publication that “*the attention heatmaps exhibit a high level of agreement with the pathologist annotations of tumor regions*” with dice scores of 0.7. False positive annotations of tumor by CLAM can happen with patterns

which are not strongly present in the negative samples. The CLAM authors gave an explanation on a similar issue on CLAM's GitHub repository, which we found reasonable and reposted it here for reference, "if all tumor slides in your training set consistently have adipose tissue or certain patterns of stromal tissue that are not present in the negative slides - then from the perspective of the learning algorithm - it would be perfectly natural to use that as a means to stratify positive vs negative samples". In our case, CLAM accurately predicted tumor regions, while it also included adipose, this likely resulted from that the pretrained weights of CLAM associates both tumors and adipose with the cancer type of the tissues. Therefore, users may need to train the CLAM model with their own sets of labelled WSIs for more accurate determination of tumor regions. A more in-depth interpretation of CLAM's performance can be found in the original publication (Lu, Williamson et al. Nat Biomed Eng 2021). We have added discussion including cautions to inform users on proper tool selection and usage in the revised manuscript on Lines 469-476.

Comment 13. The results in Figures 6a-b are confusing, as tumor regions should indicate areas enriched with tumor cells, whereas TLS/LA should be independently identified in pathology. A detailed pathology annotation to distinguish tumor-enriched regions from TLS/LA regions is needed.

Response: We apologize for the confusion. In Figures 6a-b, we sought to quantitatively characterize the heterogeneity of tumor regions according to their surrounding environments. We are not investigating the TLS. Specifically, we ranked the tumor regions by calculating the TLS scores for cells up to 1 spot-size outwards from the tumor region boundary. The ranking highlighted the heterogeneity across tumor regions and was further supported by differential gene expression and pathway enrichment analyses. For clarity, we have modified the caption and legend of Figure 6 and the manuscript (Lines 506-514).

Comment 14. In Figure 5i, T7 did not show any CNVs, which is counterintuitive given the pathology annotation. Please provide explanations for this observation.

Response: Thank you for pointing this out. We are sorry for the misleading visualization in Figure 5g (formerly Figure 5i) due to the high density of cells in region T7. Actually, T7 contains a mixture of aneuploid and diploid cells. To clarify this, we have added a zoomed-in view of T7 in the inset. This detailed visualization highlights the presence of aneuploid cells, addressing the apparent discrepancy with the pathology annotation.

Response Figure 19 (Figure 5g, formerly Figure 5i) Spatial distribution of aneuploid and diploid cells in the DCIS dataset. Inset shows the zoomed-in view of cells in region T7.

Comment 15. The analysis in Figures 6c-d doesn't add much value, as it merely compares cellular compositions in different regions. The authors should directly compare the

different TLS regions identified, revealing differences in functions, maturity, relationships, and interactions with adjacent tumor regions to provide more informative insights.

Response: Thank you for this thoughtful comment. To provide a more informative comparison of tumor regions with different TLS scores, we expanded our analysis to explore the functional distinctions and interactions between tumor regions and their immediate peritumoral neighbors (Figure S25b). Specifically, for each tumor region, we conducted a differential gene expression analysis between cells within the tumor region and those in a few layers outwards the tumor boundary. In the region with highest TLS score (T7), gene ontology (GO) enrichment highlighted pathways related to immune responses and T cell co-stimulation. Conversely, the region with the lowest TLS score (T15) exhibited enrichment in tumor-related pathways such as hypoxia response and cell adhesion.

Response Figure 20 (Figure S25b) Interaction between tumor regions and the surrounding environments. The orange dashed lines mark the pathology-annotated tumor region boundaries.

These results underscore the functional heterogeneity of tumor regions associated with varying TLS scores by illuminating how tumor cells interact with their adjacent neighborhood. We have reorganized this section (Lines 506-537) in the revised manuscript to better articulate these new results.

Comment 16. How well does Thor handle the integration of multiple samples from a cohort, including samples from adjacent sections or from different tissue types? Additionally, could Thor enhance the integration of Visium data with other modalities, such as CODEX? Although this may be beyond the current scope, providing insights on this potential would be valuable.

Response: Thank you for the questions. Thor is not originally designed to handle cohort-level integration. Integrating multi-sample ST data presents inherent challenges due to batch effects in transcriptomics data and variations in histology images across samples. In practice, Thor relies on multi-sample/-omics integration by existing preprocessing platforms such as Seurat and SCANPY. For a harder task of multi-omics integration, we have not tested whether Thor enhances the integration of Visium with other omics like CODEX, which is a potential limitation of Thor and a valuable future direction to achieve a more comprehensive, multi-modal understanding of spatial -omics data. We have added these discussion in the revised manuscript (Lines 716-720 and 731-733).

Response to Reviewer #2:

Response: Thank you for reviewing our manuscript. Please find our point-by-point responses in the combined report.

Response to Reviewer #3:

This study introduced a new computational platform called Thor, specifically designed to integrate spatial transcriptomics data with histological images for cell-level analysis. Thor leverages an anti-shrinking Markov diffusion method to infer single-cell spatial transcriptomes from spot-level data, effectively combining spatial transcriptomics with histological images molecular information. Additionally, Thor encompasses a suite of modules for performing multi-modal genomic analyses, including gene expression, pathway enrichment, transcription factor activity, and copy number variation. The authors have also developed an interactive web tool named Mjolnir, which enables users to explore cellular organization and disease mechanisms through an intuitive interface.

The study employs advanced computational methods, such as the anti-shrinking Markov diffusion, which enhances the accuracy of inferring single-cell resolution gene expression from spatial transcriptomics data. Thor has been validated using multiple experimental datasets, demonstrating its significant contribution to current spatial biology analysis. This study provided a novel tool for comprehensive cellular and molecular analysis, thereby advancing the field of precision medicine. However, several issues needed to be addressed:

Response: Thank you for giving a comprehensive review for our manuscript. We are glad that you consider Thor as a novel tool for comprehensive cellular and molecular analysis in the field. We provide point-by-point responses to all the issues as follows.

Comments:

Question 1. "Introduction" section, the author stated, "Existing analysis platforms of spatial biology focus on -omics analysis without deeply analyzing tissue histological images." Actually, there are several analysis platforms such as iStar, it is recommended to review the existing platforms and point out the limitation of the existing platforms.

Response: Thank you so much for pointing out these additional platforms. In the revised Introduction section, we have broadened our discussion to include a wide range of current spatial analysis platforms and frameworks (Lines 75-88).

Specifically, we discuss that while tools such as BayesSpace leverage spatial neighborhood information (Zhao, Stone et al. Nat Biotechnol 2021) and others such as iStar integrate histology with ST (Bergenstrahle, He et al. Nat Biotechnol 2022, Hu, Coleman et al. Cell Syst 2023, Zhang, Schroeder et al. Nat Biotechnol 2024), they operate at the spatial units that do not correspond to individual cells. As a result, this may hinder biologically relevant insights – particularly in contexts requiring cell-level data, such as analyzing ligand-receptor interactions. We also discuss that existing cell-type decomposition tools, which estimate the cell-type proportions in each spot using scRNA-seq reference datasets (Biancalani, Scalia et al. Nat Methods 2021, Kleshchevnikov, Shmatko et al. Nat Biotechnol 2022, Vahid, Brown et al. Nat Biotechnol 2023), do not infer gene expression and are further restricted by the quality and availability of scRNA-seq reference data, especially for formalin-fixed paraffin-embedded (FFPE) tissues where transcriptomic data quality is often compromised. Recent methods capable of cellular-level histological structure analysis (Jiang, Liu et al. Nat Commun 2024, Liang, Shi et al. Nat Commun 2024), also do not generate single-cell gene expression matrices for additional downstream functional or molecular analyses.

Moreover, those platforms are mostly specialized for particular tasks, while comprehensive analysis platforms (e.g. Seurat) emphasize -omics-driven analyses and do not deeply integrate with the histological image. Therefore, we develop Thor platform to fill this gap, featuring extensible modules for differential gene expression, functional pathway enrichment, transcription factor activity inference, and copy number variation analysis, alongside tissue analyses such as semi-supervised tissue annotation and nucleus detection. Additionally, we develop Mjolnir, a user-friendly web-based platform that enables interactive exploration of tissue organization and pathogenesis via vivid gigapixel images on a standard laptop with no coding required.

Question 2. In the “Method” section, “parameter settings in Thor inference”, it remains unclear whether the chosen parameter settings are based on specific rationale. How do different parameter choices impact the analysis results?

Response: Thank you for the important comment. In response, we have added parameter sensitivity analysis of Thor inference to provide rationale in the parameter settings. Thor remains robust in a wide range of parameter settings, including the diffusion step size t , the number of cell neighbors k , and the number of principal components nPC used for the transcriptome representation (Figure S27). Therefore, in all analyses of this study, common default parameters were employed. We have included the analysis (Lines 559-581) and justifications for parameter settings (Lines 969-971) in the revised manuscript.

Response Figure 21 (Figure S27a-b). Sensitivity analyses of Thor inference on (a) graph construction parameters, and (b) the diffusion steps. Mean Pearson correlation coefficients between every pair of parameter settings for all genes are plotted. Spatial distributions of a representative gene *Penk* at different diffusion steps are provided to illustrate the influence of diffusion steps on inference.

Question 3. The accuracy of the Thor platform for histological imaging analysis is dependent on the quality of the input data. The quality control process of histological images is particularly important. How to ensure the consistency of histological data from different sources. Inconsistency and quality of data varying may limit the application of Thor platform.

Response: We totally agree with this comment. To ensure different quality images can be processed accurately, we implemented APIs for multiple state-of-the-art cell segmentation tools such as StarDist and Cellpose for cell detection, both providing quality control functions such as image normalization for effective detection of cell nuclei. Importantly, Thor integrates the histology image and spatial transcriptomics per tissue. We will also keep updating the cell segmentation toolkits to make sure Thor stay on the front of cell segmentation field. We have included the description of image normalization for nuclei detection from H&E images on Lines 945-959.

Question 4. In Figure S1, the authors compare the accuracy and stability of Thor, Nearest spot, and KNN smoothing for spatial transcriptomics-based cell clustering. How does Thor's performance compare to other commonly used methods, such as ClusterMap or BayesSpace?

Response: Thanks for this question. We acknowledge the importance of benchmarking Thor with existing methods and have added BayesSpace for comparison. Our result shows that Thor outperforms BayesSpace in predicting cell-level gene expression in various conditions of spot sizes (Figure S1c) and in cluster separations with various levels of technical dropouts (Figure S1d).

Response Figure 22 (Figure S1c). Thor shows robust performance with different spot sizes. The nearest spot method maps the expression of the closest spot to the cell; KNN smoothing takes the average of the twenty nearest neighbors; the subspot-level gene expression from BayesSpace is mapped to the identified cells using nearest cell neighbors.

Response Figure 23 (Figure S1d). Thor imputes gene expression with technical dropouts and recovers cluster separation. The error bars for the mean Silhouette coefficients are omitted as they are too small to visualize. The colors in the PCA plots represent the ground truth cell type information. “Drop %” in the table is calculated as the ratio of zeros in the count matrix of the simulated scRNA-seq data.

We did not include ClusterMap in this comparison for the following reasons. ClusterMap integrates RNA spatial locations and gene identities to identify expression patterns, primarily designed for high-resolution in situ transcriptomics like STARmap and MERFISH. While it can technically be applied to Visium-like spot-level data, its output focuses on spot clustering rather than cell-level expression. Thus, a direct comparison with Thor, which infers cell-level expression by integrating histology, would not be appropriate.

We have included the analyses in the revised manuscript on Lines 179-195 and Lines 204-216.

Question 5. In Figure 3c, UMAP was used to visualize clusters in fibrotic and non-fibrotic regions, but the separation between the two regions does not appear distinct. Furthermore, Figure 3e shows that the inflammatory response and fibrosis in distant regions are more pronounced than in infarct and ischemic zones. What could be the underlying reasons for this? Providing relevant marker genes or differential genes for these regions would be helpful.

Response: Thanks for pointing this out! The UMAP shows a joint region of fibrotic and non-fibrotic which may partially be due to the low sequencing quality in the original data. As suggested, we removed this UMAP to avoid confusion in this analysis, and provided a table of differentially expressed genes (DEGs) between cells in the fibrotic and those in the non-fibrotic regions (Supplementary Table S2). We also have performed pathway enrichment analysis. The results showed that regardless of sample zones (ischemic, remote, or fibrosis), the semi-annotated fibrotic regions, showed significant enrichment in pathways such as fibroblast proliferation, stress fiber assembly, and collagen fibril organization. In contrast, myocardium-related pathways were enriched in non-fibrotic regions.

We appreciate that you pointed out the pronounced inflammatory response and fibrosis observed in fibrotic regions from remote zone samples. The reason could be attributed to the heterogeneous progression of ischemic injury in the patient samples. Following myocardial infarction or ischemia, the immediate infarct area typically undergoes acute cell death and necrosis, while the distant/remote zone may experience a delayed and prolonged inflammatory and fibrotic response (Frangogiannis Cardiovasc Res 2021, Kuppe, Ramirez Flores et al. Nature 2022). Although ischemic and fibrosis zones contained the largest proportions of fibrotic regions at the whole tissue level, those findings demonstrate that functionally distinct fibrotic domains can exist outside necrotic regions.

We have included the discussion on Lines 371-378 and provided the full table of differential genes in Supplementary Table S2 for additional reference.

Question 6. In Figure 5C, the researchers used Cell-ID to obtain the spatial distribution of cell types in tumor tissue. However, tumor cell annotations are absent in Figure 5C. Additionally, monocytes and macrophages are grouped together. Can these two cell types be distinguished? If not, what could be the possible reasons for this?

Response: We apologize for any confusion regarding the annotations in Figure 5c. To address this, we refined the annotations by incorporating key gene signatures. Specifically, *EPCAM* and *CDH1* were used together to separate tumor cells from normal epithelial cells, while *CD14* and *VCAN* (versican) were used to separate monocytes and macrophages. For instance, the distinct expression patterns of *CD14* and *VCAN* (both upregulated in circulating monocytes and downregulated upon differentiation into macrophages) allowed us to distinguish monocytes from macrophages. These refinements provide a more detailed understanding of the cellular compositions within the tumor tissue. We have updated Figure 5c and added the methodological details of cell type annotation on Lines 1006-1020.

We would like to further note that Cell-ID was used to transfer cell type information from a reference scRNA-seq dataset, in which both normal and tumor epithelial cells were labelled as epithelial cells, and monocytes and macrophages were grouped together. As a result, the annotations for Thor-inferred cells reflected the cell type classifications in the reference data. However, these groupings were not due to limitations in Thor's cell type resolution but rather a direct consequence of the reference labels.

Response Figure 24 (Figure S20a). Monocytes and macrophages are separated by expression levels of marker genes. (a) Spatial distribution of the two clusters of cells in the ‘Monocytes/Macrophages’ in original annotation. (b) Density plot expression levels of the marker genes. (c) Spatial expression profiles of the marker genes.

Response Figure 25 (updated Figure 5c). Spatial distribution of refined cell types predicted from Thor-inferred gene expression data.

Question 7. Cells of the same type may exhibit different subtypes due to their environment or spatial location. Could the study improve upon the classification of cell subtypes?

Response: Thank you for the important suggestion. To further demonstrate how Thor reveals cell subtypes due to their environment, we provided refined annotations of cells in the mouse olfactory bulb (MOB) by transferring the cell type information from an external MOB scRNA-seq data (Tepe, Hill et al. *Cell Reports* 2018) to Thor-inferred cell-level data with Cell-ID.

Thor’s cell type distributions accurately depict the layered architecture of the MOB. Moreover, its integrated approach, leveraging both spatial context and transcriptomic information, enabled the identification of subtle subtype distinctions tied to specific tissue layers. For example, Thor distinguished granule cells into GC-1 (mainly localized in the internal plexiform layer) and GC-2 (predominantly in the granule cell layer). Similarly, it identified two mitral cell subtypes, M/TC-

1 and M/TC-2, with M/TC-2 concentrated in the mitral layer and M/TC-1 extending into the glomerular layer.

These findings underscore Thor's capability to enhance cell classification at a finer resolution, capturing meaningful spatial and transcriptomic heterogeneity. We have incorporated these additional analyses into the revised manuscript to highlight Thor's ability to refine cell-type classification based on local environment on Lines 307-315.

Response Figure 26 (Figure S10b). The spatial distribution of neuron subtypes in the mouse olfactory bulb predicted from Thor-inferred single-cell gene expression. OSN: Olfactory sensory neuron, PGC: Periglomerular cell, GC: Granule cell, M/TC: Mitral/Tufted cell.

Question 8. The study infers cell types and gene expression from spatial transcriptomics data. Compared to single-cell transcriptomics, spatial transcriptomics offers significant advantages for studying cell-cell communication. Could Thor generate the cell communication module in this research?

Response: Thank you for the suggestion. We agree that cell-cell communication analysis is a powerful way to leverage the single-cell resolution spatial transcriptomics inferred by Thor. With this in mind, we have developed an API to interface Thor-inferred cell-level data with COMMOT (*thor.analy.ccc*), a state-of-the-art tool for cell-cell communication analysis in spatial transcriptomics data (Cang, Zhao et al. *Nat Methods* 2023). While our implementation is still under active development—particularly because sequence-based single-cell spatial data often exhibit limited coverage, or high noise-to-signal ratios, and cannot directly measure protein abundance (Zhu, Wang et al. *Nat Methods* 2024).

We modified the original COMMOT package (which calculates pairwise distances for all cells, resulting in prohibitive RAM usage and long run time when analyzing several tens of thousands of cells) by restricting the calculation of the cell-cell distance to a user-defined cutoff (typically ~100 μm) and use of sparse matrix operations. Those modifications reduce the computational complexity from $O(N^2)$ to $O(N)$, offering manageable runtime and memory requirements for large-scale datasets. We have created a Jupyter notebook to show how to run cell-cell communication analyses with Thor-inferred gene expression data (provided as a supplementary file for review).

In addition, Thor outputs cell-level gene expression in AnnData format, which integrates seamlessly with other platforms for cell-cell communication studies such as stLearn and Squidpy. Our API documentation and GitHub repository also invite community contributions toward integrating novel or updated algorithms for spatially resolved cell-cell communication,

ensuring that Thor remains a flexible and forward-compatible framework for advanced spatial transcriptomics analysis.

Question 9. Did Thor have the capability of inferring cell-level information from metabolomics or proteomics?

Response: Thank you for this thoughtful question. While Thor has not been applied to spatial metabolomics or proteomics data in our current study, its underlying framework (building a cell-cell graph informed by spatial coordinates and histological features and performing Markov diffusion to infer cell level information) could, in principle, be adapted for these modalities.

In such a scenario, Thor's approach to refining spatial measurements to the cell level would remain conceptually similar, merely substituting transcriptomic values with metabolomic or proteomic intensities. However, extending Thor to these omics layers would require appropriate validation and potentially modifications tailored to the distinct characteristics of metabolic or proteomic measurements.

We anticipate that future developments will provide deeper insights into complex tissue characterizations by integrating these additional modalities. We have included the discussions in the revised manuscript (Lines 611-621).

Question 10. Now, the Stereo-seq technology can provide the resolution of life-space expression at the subcellular level. Whether Thor can be compared to this technology?

Response: Thank you for raising this insightful comparison between Thor and Stereo-seq, a cutting-edge technology achieving subcellular resolution (~220 nm) via DNA nanoball-patterned arrays. In contrast, Thor is designed to infer cell-level gene expression based on cell-level features extracted from histology images and patch-level transcriptomics data, and thus does not operate at subcellular resolution.

This limitation is primarily due to the light diffraction limits in standard histology imaging and complex morphological variability of subcellular structures, which make robust and accurate segmentation of individual organelles or subcellular structures highly challenging (Sekh, Opstad et al. Nature Machine Intelligence 2021). Although specialized organelle-specific computational models may enable subcellular segmentation, these approaches remain limited to certain organelles or highly specialized imaging platform such as electron microscopy (Glancy Cell Syst 2023, Lu, Christensen et al. Nat Methods 2023, Ma, He et al. Nat Commun 2024).

Furthermore, Thor is optimized for cell-level integrated analyses of transcriptomics and histology and complements, rather than competes with, nanoscale spatial omics technologies. Its modular architecture could, in future work, integrate with subcellular methods like Stereo-seq to bridge tissue, cellular, and subcellular-level insights. We have added a discussion in the revised manuscript (Lines 720-727) clarifying Thor's current limitations at the subcellular analyses and contrasting its resolution with that of Stereo-seq, thereby positioning Thor within the evolving spatial omics landscape.

Question 11. Some data analysis frameworks that have enabled accurate identification of histological structures at cellular resolution such as PROST, and METI. Please discuss the advantages of Thor over current technologies.

Response: Thank you for the great suggestion. We have expanded our discussion to highlight Thor's advantages over existing frameworks for studying histological structures.

PROST uses spatial relationships and transcriptomics data to identify spatially variable genes and to cluster spatial domains, but it does not enhance the resolution of the original ST data. Thus, with Visium data, PROST operates at the Visium-spot level and does not utilize histology images. By contrast, Thor integrates cell-level features from histology images with spot-level transcriptomics, enabling inference of gene expression at the single-cell level and providing a more granular analysis of histological structures from Visium data.

METI, meanwhile, is an end-to-end framework tailored to cancer ST data, mapping tumor cells and the surrounding microenvironment primarily in oncology-focused contexts. Thor, on the other hand, was conceived as a more generalizable approach applicable across various tissues, disease states, and organisms. Moreover, neither PROST nor METI directly output single-cell gene expression. In contrast, Thor integrates histological and transcriptomic data in a task-agnostic manner to infer spatially resolved single-cell gene expression. This capability supports a broad range of downstream analyses and comes bundled with extensive analytical modules, including pathway enrichment, spatial gene module identification, differential gene expression, transcription factor activity estimation, and interactive whole-slide data visualization. Taken together, these features allow Thor to complement and extend the capabilities of frameworks like PROST and METI by offering deeper spatial and molecular insights into tissue architecture. We have included these discussions in the revised manuscript on Lines 637-654.

Response to Reviewer #4:

The paper presents an algorithm, Thor, a comprehensive computational tool designed for spatial transcriptomics and histology analysis. Thor aims to bridge the gap between ST and image analysis, allowing for multi-modal exploration of tissue characteristics. The platform features an anti-shrinking Markov diffusion method that infers cell-resolution transcriptomes from spot-level data and integrates these with histological features. Thor is supplemented by Mjolnir a web-based tool for interactive tissue analysis. The platform offers functionalities like immune response analysis, transcription factor activity estimation, pathway enrichment, and semi-supervised tissue annotation. Its performance is validated across different datasets, including heart failure tissues, mouse olfactory bulb, and breast cancer tissues. The topic is very interesting, and it is noteworthy that it shows a possibility that overcomes current limitations of ST in terms of resolution. There are some limitations and drawbacks of this paper.

Response: We sincerely thank you for the accurate and comprehensive summary of our work. We are particularly grateful for your comments on the novelty of Thor and its potential to address the current limitations of spatial transcriptomics in terms of resolution. Below, we address your questions and concerns point by point.

Question 1. Although Thor is compared to other tools like iStar, the scope of comparison is relatively narrow. It lacks a detailed, quantitative evaluation against a broader set of established tools in spatial transcriptomics, such as TESLA, BayesSpace, etc. More comparisons with varied metrics would solidify its advantages.

Response: Thank you for the suggestion. Following your suggestion, we have improved the quantitative evaluation of Thor against two more tools (TESLA and BayesSpace) using two more metrics, in addition to the existing comparison. We compared the predicted gene expression with the paired Xenium data (ground truth) on the breast cancer data from 10x Genomics. We introduced two additional metrics, including pixel-centric metric root mean squared error (RMSE, pixels) and cell-centric metric RMSE (cells) for a more solid evaluation. Our quantitative comparison showed that Thor achieved the highest accuracy among the four tools, with significantly higher CW-SSIM, and lower RMSE at pixel and cell levels (Figures 2c and S7). We have included the new results and analysis in the revised manuscript on Lines 258-274.

Response Figure 27 (Figures 2c and S7a-b). Quantitative comparison between Thor and ST spatial resolution-enhancement tools. (a) Using image-level metrics and (b) Using cell-level metrics. One-sided Mann-Whitney tests were performed between Thor and other two best-

performing tools. For the box plots, the middle line in the box plot, median; box boundary, interquartile range; whiskers, 5-95 percentile.

Question 2. The limitation of Thor's approach is that recent advancements in ST technologies, such as VisiumHD, Slide-seq, Xenium, and CosMX, are already enhancing resolution significantly. For instance, image-based ST platforms like Xenium and CosMX can provide cellular-level resolution while analyzing thousands of genes (>5k), potentially reducing the necessity for computational inference methods like Thor. If ST platforms can inherently deliver cellular resolution along with a sufficient number of gene sets for comprehensive transcriptomic analysis, the added value of Thor in terms of improving spatial resolution may be diminished. Therefore, Thor's utility under these conditions could be limited, and its impact may need to be reassessed in the context of increasingly advanced ST technologies that natively achieve high resolution.

Response: Thank you for the very important comment. While emerging ST platforms are pushing the resolution and throughput boundaries, each technology still faces practical limitations that Thor can help address. For instance, although Visium HD provides sub-cellular bin sizes, it can suffer from high dropout rates and low gene coverage (Polanski, Bartolome-Casado et al. Bioinformatics 2024), as well as imperfect alignment between bins and cells (Kamel, Song et al. bioRxiv 2024). Slide-seq may exhibit sparse transcript detection and limited capture size (You, Fu et al. Nat Methods 2024), and image-based platforms such as Xenium and CosMX rely on predefined gene panels and may yield incomplete transcriptomic coverage.

First, to show Thor's utility on high-resolution spatial data, we added an in-house bladder cancer sample sequenced on Visium HD. In our experiment, despite the high spatial resolution (up to 2 μm square bins), even the 8 μm bin data showed high technical noise. For instance, immune cell marker PTPRC is sparsely expressed in densely packed lymphoid regions (Figure S26, ROI 1) and the urothelium-associated gene SPINK1 showed high expression in non-tissue areas (ROI 3, arrow). Thor was originally developed before Visium HD became available, we adapted Thor to integrate HD data with histology and infer cell-level gene expression. This morphology-guided approach enhanced spatial gene expression patterns, improving alignment with pathology annotations (Figure S26).

Second, Thor functions as a comprehensive downstream analysis platform for large-scale spatial data. High-resolution technologies generate a massive number of spots/bins/cells (e.g. ~ 5 million 2 μm square bins on a 6.5 mm x 6.5 mm Visium HD slide), posing computational and visualization challenges. Thor adapts state-of-the-art analysis tools and supports streamlined analyses including clustering, differential gene expression, pathway enrichment analysis, transcription factor activity inference, CNV analysis, and semi-supervised annotation. Thor's interactive visualization tool, Mjolnir, also supports investigation of modalities (e.g. histology images, gene expression) as vivid gigapixel images on a standard laptop.

Additionally, Thor remains cost-effective by enabling single-cell resolution analysis on more affordable standard Visium data. This makes Thor attractive for laboratories facing resource limitations or those wishing to reanalyze existing data generated from standard or legacy ST platforms.

Response Figure 28 (Figure S26). Thor imputes Visium HD data and reconstructs gene expression patterns that align with pathology annotations in a bladder cancer sample.

Pathology annotations (left) highlight immune cells (ROI 1), invasive carcinoma (ROI 2), and a tumor fragment (ROI 3). Gene expression patterns inferred by Thor (middle) and from Visium HD (right). The orange arrow points to a region with no cell.

In summary, Thor’s flexible framework remains relevant for bridging morphological and transcriptomic data across a variety of resolutions and experimental protocols. Thor provides an extensive suite of downstream analyses modules, serving both as a transitional tool and a complementary approach that enhances the interpretability of even the most cutting-edge ST platforms. This versatility ensures Thor’s continuing impact in an era of rapidly advancing spatial transcriptomics technologies. We have included these analyses (Lines 538-557) and discussions (Lines 692-707) in the revised manuscript.

Question 3. Regarding the use of CW-SSIM for evaluation, there are concerns about whether the comparison between Xenium, Thor, and iSTAR is entirely fair. Specifically, the output of Thor and Xenium is not at the pixel level, whereas iSTAR generates outputs at a pixel level, making the comparison potentially biased in favor of pixel-level evaluation. A more nuanced evaluation metric, tailored to the inherent characteristics of each method, might be more appropriate.

Response: Thank you for raising the critical concern. For a more thorough and fair evaluation of pixel-level data (image) and cell/subspot-level data, we have added two more metrics, including pixel-centric metric RMSE (pixels) and cell-centric metric RMSE (cells). We also expanded our comparison with two more existing methods, following your suggestion in Question 1.

The output basic units of gene expression levels in Thor, iStar, BayesSpace, and TESLA are cell, superpixel, subspot, and superpixel, respectively. Therefore, we adopted metrics for comparing

image pixels and cells to provide a more complete evaluation of the performance. On the one hand, by converting spatial profiles of gene expression data into images, we compared the similarities between the predicted spatial patterns with the Xenium spatial patterns using the metrics CW-SSIM and RMSE of pixel values. On the other hand, by mapping the pixel expression data to the cells using the nearest neighbors approach (we used the nearest 5 neighbors; by varying this number from 5 to 20, the trend did not change), we compared the error between resulted cell-level gene expression data with the Xenium data using RMSE (cell wise) as an additional metric.

Thor achieved the highest similarity with the Xenium data on all the metrics. When using the cell-wise RMSE, the general trend remains, yet the difference between the four methods became less prominent. This is likely because all the gene expression levels including Thor needed to be mapped to the common cell positions (Xenium cells) using nearest neighbors before calculating cell-wise RMSE, which might have smoothed out some intricate details in the spatial pattern, as seen in Figures S7c-d. We consider CW-SSIM a more suitable metric, as it is widely used in ST data analysis that takes spatial information into account.

Altogether, Thor demonstrated significantly better agreement with the Xenium data (Figure S7). The results are added to the revised manuscript on Lines 258-274.

Response Figure 29 (Figure S7). Quantitative comparison between Thor and ST spatial resolution-enhancement tools. (a) Using image-level metrics, and (b) Using cell-level metrics. One-sided Mann-Whitney tests were performed between Thor and other two best-performing tools. For the box plots, the middle line in the box plot, median; box boundary, interquartile range; whiskers, 5-95 percentile. (c) Spatial profiles of representative genes inferred by Thor and other tools. The data from Xenium are provided for reference. The CW-SSIM scores are included. (d) Spatial profiles of representative genes inferred by Thor and other tools. The RMSE of Min-Max normalized cell level expressions are provided. Nearest cell/superpixel/subspot expression levels were mapped to the Xenium cell positions.

Question 4. Thor's evaluation could be extended to include comparisons with simpler approaches, such as nearest value assignment after nucleus segmentation or interpolation from surrounding spot values. This would help clarify whether Thor's more sophisticated approach offers a genuine advantage or if the improvements are merely due to incorporating basic nucleus segmentation. It is important to differentiate whether Thor's better performance results primarily from its advanced diffusion model or simply from the inclusion of nucleus segmentation. Adding these comparisons would provide a clearer perspective on the unique contributions of Thor.

Response: Thank you for the suggestion. To address this, we compared Thor's performance with simpler approaches, including the nearest spot and KNN smoothing methods. The nearest spot method assigns the gene expression of a cell directly from its nearest spot. Both KNN smoothing method and Thor start with the nearest value assignment data. The KNN smoothing approach assigns gene expression levels by averaging the values of a cell's spatially nearest neighbors. In contrast, Thor uses a Markov diffusion process on a cell-cell graph constructed in the combined space of transcriptomics, histological features, and cell positions.

In our experiments, we varied the spot sizes and calculated the normalized error in expression levels across all genes. We observed that when the spot size was close to the cell size ($< 25 \mu\text{m}$), the nearest spot method and Thor were both accurate. The accuracy of the nearest spot method decreased as spot size increased, while Thor still remained accurate with the spot size up to $100 \mu\text{m}$. This indicates that Thor's superior performance is not primarily due to incorporating nucleus segmentation. By contrast, both the KNN smoothing method and BayesSpace performed poorly across all spot sizes, with median NRMSE values of approximately 0.2 (Figure S1c). The KNN smoothing method consistently underperformed, underscoring the benefits of Thor's shared nearest neighbors cell-cell graph and feature-preserving Markov diffusion approach. We have included those findings in the revised manuscript on Lines 179-195, providing additional clarity on the unique contributions of Thor.

Response Figure 30 (Figure S1c). Thor shows robust performance with different spot sizes. The nearest spot method maps the expression of the closest spot to the cell; KNN smoothing takes the average of the twenty nearest neighbors; the subspot-level gene expression from BayesSpace is mapped to the identified cells using nearest cell neighbors. For the box plots, the middle line in the box plot, median; box boundary, interquartile range; whiskers, 5-95 percentile.

Question 5. Thor heavily relies on histological images to infer cellular-level information. If high-quality histological images are unavailable or the images are noisy, the accuracy of in silico cell transcriptomes may be affected. In addition, sometimes MPP (maybe 0.25 um) cannot be obtained by other real-world situations. The paper does not discuss how Thor handles degraded or suboptimal images.

Response: Thank you for raising this important point regarding Thor's reliance on high-quality histological images and its performance under suboptimal imaging conditions. To address this concern, we have conducted simulations to assess Thor's performance when handling suboptimal histology images. Two common issues in suboptimal histology images that could affect Thor's performance were considered: (i) missed detection of cell nuclei, particularly in out-of-focus or densely packed regions, and (ii) erroneous cell-cell connections due to poor histological features. Despite these challenges, our evaluation demonstrates that Thor maintains high prediction accuracy even under substantial cell loss and misrepresented cell-cell connections (up to 40%), as shown in Figure S1b. This highlights the method's robustness in handling imperfect histology images.

Response Figure 31 (Figure S1b). Thor is robust with cell missouts or erroneous connections caused by suboptimal histology images. NRMSE provides a quantitative measure of the normalized deviation of Thor-inferred gene expression from the ground-truth gene expression. The performance of Thor remains accurate when there are 40% of cells missing or 40% of cells have randomized connections. The dash lines mark the baseline level when raw spot data are mapped to nearest cells. The middle line in the box plot, median; box boundary, interquartile range; whiskers, 5-95 percentile.

We further discuss that real-world scenarios can introduce additional complexities, such as loss of focus in imaging or improper staining across large tissue regions. Such conditions may lead to missed cell detection or unrepresentative image features and Thor's performance may understandably suffer. We have added the analysis of Thor's robustness in suboptimal image conditions (Lines 168-177) and discussions (709-713) about those limitations in the revised manuscript.

Question 6. While Thor identifies molecular signatures in different tissue zones and conditions, there are limited wet-lab biological validations or 'well known' acceptable validation for these findings. In terms of biological effects of pathological findings, it

needs to be validated that Thor could result in discovering novel results which couldn't be discovered by Visium data alone.

Response: Thank you for raising this critical point about validating Thor's ability to uncover novel molecular insights beyond standard Visium data.

To demonstrate Thor's biological relevance, we validated its inferred cell-level gene expression through extensive computational simulations and high-resolution experimental modalities, including MERFISH, Xenium, ISH, and Stereo-seq. These comparisons confirm that Thor accurately refines spatial gene expression and improves tissue annotation at single-cell resolution. Additionally, during the revision, we introduced an in-house bladder cancer sample data sequenced on Visium HD and applied Thor to integrate the bin data with the histology image. Our result showed that Thor imputed gene expression, and the spatial distribution of marker genes aligned more closely with pathology annotations than did bin-level Visium HD data (Figure S26).

Response Figure 32 (Figure S26). Thor imputes Visium HD data and reconstructs gene expression patterns that align with pathology annotations in a bladder cancer sample.

Pathology annotations (left) highlight immune cells (ROI 1), invasive carcinoma (ROI 2), and a tumor fragment (ROI 3). Gene expression patterns inferred by Thor (middle) and from Visium HD (right). The orange arrow points to a region with no cell.

Furthermore, we performed large-scale semi-supervised annotations to validate Thor's strength over spot-level data in characterizing tissue regions and capturing biological insights from pathological findings. Thor's semi-supervised annotation precisely annotated critical structures such as vessels, nodes, adipose tissue, and fibrotic regions (Figure S12). In contrast, spot-level clustering, even with optimized parameters, struggled to distinguish structures such as vessel-associated regions enriched with smooth muscle from certain myocardium regions (Figure S13).

These results underscore Thor's ability to capture cellular heterogeneity and enhance spatial tissue annotation by integrating histology with transcriptomics, surpassing spot-level clustering.

Response Figure 33 (Figure S13). Comparison between Thor's semi-supervised annotation and the spot-level clustering. (a) H&E staining image and expert-annotations on the heart tissue sample. (b) Cells annotated via Thor's semi-supervised annotation. Annotation of cells are mapped to corresponding spots according to majority voting for quantitative evaluation against expert annotation. (c) K-means clusters solely based on Visium ST data. Regardless of the number of clusters used, Visium ST data alone fails to accurately distinguish vessel-associated spots (enriched with smooth muscle and endothelial cells) from some myocardium spots with high TAGLN (a smooth muscle marker) expression.

Crucially, Thor's single-cell resolution facilitates the discovery of biological phenomena that would remain hidden at the spot level. For example, the 55 μm -diameter spots in Visium data often encompass multiple cell types, making it infeasible to identify key regulators in thin vascular regions where distinct cell types are closely interspersed (Response Figure 34). We identified *PLA2G2A* as a potential regulator of vascular regeneration by isolating gene expression patterns unique to thin vessel structures. We then validated *PLA2G2A* expression through immunofluorescence on heart tissues from the same patients, confirming its localization in thin vessels of post-LVAD heart failure patients and thereby providing direct, wet-lab evidence of Thor's capacity for generating actionable biological hypotheses.

Response Figure 34. Visualization of Visium spots and Thor-inferred cells on vessel regions from in-house heart failure patient samples.

In summary, Thor’s integration of histological features and spatial transcriptomics not only improves the resolution and accuracy of tissue annotations but also enables the identification of previously unrecognized molecular regulators in specific tissue niches. By complementing spot-level data with refined single-cell insights, Thor can reveal new targets and pathways that standard Visium data alone would likely obscure. We have included these findings in the revised manuscript (Lines 339-348 and Lines 538-557).

Question 7. Graph Construction and Markov Diffusion part: The Markov diffusion method employed here involves constructing a connectivity matrix, followed by iterative graph-based smoothing. This complexity is appropriate given the need for nuanced spatial analysis. The code lacks sufficient parameter sensitivity analysis. For example, the choice of diffusion step size or the Gaussian kernel used for affinity computation may significantly impact the outcome. Providing grid search or comparison according to the hyperparameters of the results to different parameter values would add robustness to the analysis.

Response: Thank you for the suggestion. We have conducted a sensitivity analysis of key parameters involved in cell-cell graph construction and the Markov diffusion process, including the number of neighbors k (which controls the kernel size), the number of principal components nPC used for the transcriptome representation, and the diffusion steps t . We systematically evaluated the robustness of Thor to these parameters as follows.

Response Figure 35 (Figure S27a-b). Sensitivity analyses of Thor inference on (a) graph construction parameters, and (b) the diffusion steps. Mean Pearson correlation coefficients between every parameter settings for all genes are plotted. Spatial distributions of a representative gene *Penk* at different diffusion steps are provided to illustrate the influence of diffusion steps on inference.

(1) Number of cell neighbors (k):

Thor constructs a shared nearest neighbors (SNN) cell-cell graph based on the k -nearest neighbors in the combinatory space. We tested a range of k values from 3 to 100 on the MOB dataset while keeping other parameters fixed ($t = 40$ and $nPC = 10$). To reduce bias from highly expressed genes, we applied z-score normalization for each gene. We then calculated the Pearson correlation coefficients (r) across each pair of k settings. Thor demonstrated strong robustness for k values between 4 and 10, with a mean $r = 0.88$ and standard deviation (std) = 0.09. However, very small k values (< 3) may produce disconnected cell graphs, whereas very large k values (40-100) may lead to over-smoothing and weaker correlations with results of other k values (mean $r = 0.56$, $\text{std} = 0.27$).

(2) Number of principal components of the transcriptome data (nPC):

We next evaluated the impact of varying nPC values while fixing $k = 5$ and $t = 40$. As shown in Figure S27a, Thor remains highly robust when $nPC \geq 8$ (mean $r = 0.94$, $\text{std} = 0.05$). In contrast, $nPC < 4$ fails to capture sufficient complexity in the data, leading to lower correlations with high nPC values.

(3) Diffusion step size (t):

We also evaluated a range of diffusion time t while keeping $nPC = 10$ and $k = 5$ fixed. Thor converged after approximately 10 diffusion steps, achieving a mean $r = 0.90$ (std = 0.10) for $t = 10$. However, large t values (e.g. $t > 50$) may notably increase run time without significant performance gains (Figure S27b).

Overall, our analyses show that Thor is robust to a broad range of t , k , and nPC values. These findings indicate that minor adjustments within reasonable parameter ranges have minimal effect on Thor's results, which justifies that we kept a common set of parameters across all case studies. We have added the analysis to the revised manuscript on Lines 559-581.

Question 8. In VAE, latent dimension and input dimension (maybe depends on HVGs) could be changed. It also affect the results. To have more generalizability, hyperparameter tuning seems to be required.

Response: Thank you for the suggestion. We agree that hyperparameter tuning, such as adjusting the input and latent dimensions in variational autoencoder (VAE) models, can affect the results and is an important consideration for generalizability. Thor requires training of a VAE, where Markov diffusion is performed on the reduced representation of the transcriptomics data (i.e., the VAE latent embeddings) in fast mode.

The input dimension of VAE depends on the genes of interest, such as highly variable genes or spatially variable genes. For example, in the mouse olfactory bulb (MOB) dataset, we used the 2,781 highly variable genes detected by SCANPY as the input dimension. VAE has been widely used for RNA-seq data analysis, and detailed guidelines for training VAE models can be found in popular frameworks (Lopez, Regier et al. Nat Methods 2018, Gronbech, Vording et al. Bioinformatics 2020, Xu, Wang et al. Nat Commun 2023). A proper latent dimension should sufficiently capture the biological complexity in the data. For instance, a latent dimension of 10 is set by default in *scvi-tools* for common scRNA-seq data, with 20 or 30 being appropriate for more complex datasets.

We evaluated Thor's performance on the MOB dataset by varying the latent dimensions in separate VAE models (8, 16, 20, 32, 64, and 128) while keeping other parameters fixed. Thor-predicted gene expressions remained highly consistent with Pearson's $r > 0.85$ across all settings (Figure S27c). These results indicate that Thor is robust to a broad range of latent dimensions. The results are included in the revised manuscript on Lines 583-595.

Response Figure 36 (Figure S27c). Sensitivity of Thor inference to the dimension of the latent layer in VAE. Mean Pearson correlation coefficients between every parameter settings for all genes are plotted. Spatial distributions of a representative gene *Penk* are provided to illustrate the influence of latent dimension on inference.

References

- Bannon, D., E. Moen, M. Schwartz, E. Borba, T. Kudo, N. Greenwald, V. Vijayakumar, B. Chang, E. Pao, E. Osterman, W. Graf and D. Van Valen (2021). "DeepCell Kiosk: scaling deep learning-enabled cellular image analysis with Kubernetes." Nat Methods **18**(1): 43-45.
- Bergenstrahle, L., B. He, J. Bergenstrahle, X. Abalo, R. Mirzazadeh, K. Thrane, A. L. Ji, A. Andersson, L. Larsson, N. Stakenborg, G. Boeckxstaens, P. Khavari, J. Zou, J. Lundeberg and J. Maaskola (2022). "Super-resolved spatial transcriptomics by deep data fusion." Nat Biotechnol **40**(4): 476-479.
- Biancalani, T., G. Scalia, L. Buffoni, R. Avasthi, Z. Lu, A. Sanger, N. Tokcan, C. R. Vanderburg, A. Segerstolpe, M. Zhang, I. Avraham-Davidi, S. Vickovic, M. Nitzan, S. Ma, A. Subramanian, M. Lipinski, J. Buenrostro, N. B. Brown, D. Fanelli, X. Zhuang, E. Z. Macosko and A. Regev (2021). "Deep learning and alignment of spatially resolved single-cell transcriptomes with Tangram." Nat Methods **18**(11): 1352-1362.
- Cang, Z., Y. Zhao, A. A. Almet, A. Stabell, R. Ramos, M. V. Plikus, S. X. Atwood and Q. Nie (2023). "Screening cell-cell communication in spatial transcriptomics via collective optimal transport." Nat Methods **20**(2): 218-228.
- Chen, R. J., T. Ding, M. Y. Lu, D. F. K. Williamson, G. Jaume, A. H. Song, B. Chen, A. Zhang, D. Shao, M. Shaban, M. Williams, L. Oldenburg, L. L. Weishaupt, J. J. Wang, A. Vaidya, L. P. Le, G. Gerber, S. Sahai, W. Williams and F. Mahmood (2024). "Towards a general-purpose foundation model for computational pathology." Nat Med **30**(3): 850-862.
- Cortal, A., L. Martignetti, E. Six and A. Rausell (2021). "Gene signature extraction and cell identity recognition at the single-cell level with Cell-ID." Nat Biotechnol **39**(9): 1095-1102.
- Frangogiannis, N. G. (2021). "Cardiac fibrosis." Cardiovasc Res **117**(6): 1450-1488.
- Glancy, B. (2023). "MitoNet: A generalizable model for segmentation of individual mitochondria within electron microscopy datasets." Cell Syst **14**(1): 7-8.
- Greenwald, N. F., G. Miller, E. Moen, A. Kong, A. Kagel, T. Dougherty, C. C. Fullaway, B. J. McIntosh, K. X. Leow, M. S. Schwartz, C. Pavelchek, S. Cui, I. Camplisson, O. Bar-Tal, J. Singh, M. Fong, G. Chaudhry, Z. Abraham, J. Moseley, S. Warshawsky, E. Soon, S. Greenbaum, T. Risom, T. Hollmann, S. C. Bendall, L. Keren, W. Graf, M. Angelo and D. Van Valen (2022). "Whole-cell segmentation of tissue images with human-level performance using large-scale data annotation and deep learning." Nat Biotechnol **40**(4): 555-565.
- Gronbech, C. H., M. F. Vording, P. N. Timshel, C. K. Sonderby, T. H. Pers and O. Winther (2020). "scVAE: variational auto-encoders for single-cell gene expression data." Bioinformatics **36**(16): 4415-4422.
- Hörst, F., M. Rempe, L. Heine, C. Seibold, J. Keyl, G. Baldini, S. Ugurel, J. Siveke, B. Grünwald and J. Egger (2024). "Cellvit: Vision transformers for precise cell segmentation and classification." Medical Image Analysis **94**: 103143.
- Hu, J., K. Coleman, D. Zhang, E. B. Lee, H. Kadara, L. Wang and M. Li (2023). "Deciphering tumor ecosystems at super resolution from spatial transcriptomics with TESLA." Cell Syst **14**(5): 404-417 e404.

Jiang, J., Y. Liu, J. Qin, J. Chen, J. Wu, M. P. Pizzi, R. Lazcano, K. Yamashita, Z. Xu, G. Pei, K. S. Cho, Y. Chu, A. Sinjab, F. Peng, X. Yan, G. Han, R. Wang, E. Dai, Y. Dai, B. A. Czerniak, A. Futreal, A. Maitra, A. Lazar, H. Kadara, A. A. Jazaeri, X. Cheng, J. Ajani, J. Gao, J. Hu and L. Wang (2024). "METI: deep profiling of tumor ecosystems by integrating cell morphology and spatial transcriptomics." *Nat Commun* **15**(1): 7312.

Kamel, M., Y. Song, A. Solbas, S. Villordo, A. Sarangi, P. Senin, S. Mathew, L. C. Ayestas, S. Wang, M. Classe, Z. Bar-Joseph and A. P. Planas (2024). "ENACT: End-to-End Analysis of Visium High Definition (HD) Data." *bioRxiv*: 2024.2010.2017.618905.

Kanemaru, K., J. Cranley, D. Muraro, A. M. A. Miranda, S. Y. Ho, A. Wilbrey-Clark, J. Patrick Pett, K. Polanski, L. Richardson, M. Litvinukova, N. Kumasaka, Y. Qin, Z. Jablonska, C. I. Semprich, L. Mach, M. Dabrowska, N. Richoz, L. Bolt, L. Mamanova, R. Kapuge, S. N. Barnett, S. Perera, C. Talavera-Lopez, I. Mulas, K. T. Mahbubani, L. Tuck, L. Wang, M. M. Huang, M. Prete, S. Pritchard, J. Dark, K. Saeb-Parsy, M. Patel, M. R. Clatworthy, N. Hubner, R. A. Chowdhury, M. Nosedá and S. A. Teichmann (2023). "Spatially resolved multiomics of human cardiac niches." *Nature* **619**(7971): 801-810.

Khatri, P., M. Sirota and A. J. Butte (2012). "Ten years of pathway analysis: current approaches and outstanding challenges." *PLoS Comput Biol* **8**(2): e1002375.

Kleshchevnikov, V., A. Shmatko, E. Dann, A. Aivazidis, H. W. King, T. Li, R. Elmentaite, A. Lomakin, V. Kedlian, A. Gayoso, M. S. Jain, J. S. Park, L. Ramona, E. Tuck, A. Arutyunyan, R. Vento-Tormo, M. Gerstung, L. James, O. Stegle and O. A. Bayraktar (2022). "Cell2location maps fine-grained cell types in spatial transcriptomics." *Nat Biotechnol* **40**(5): 661-671.

Kuppe, C., R. O. Ramirez Flores, Z. Li, S. Hayat, R. T. Levinson, X. Liao, M. T. Hannani, J. Tanevski, F. Wunnemann, J. S. Nagai, M. Halder, D. Schumacher, S. Menzel, G. Schafer, K. Hoefft, M. Cheng, S. Ziegler, X. Zhang, F. Peisker, N. Kaesler, T. Saritas, Y. Xu, A. Kassner, J. Gummert, M. Morshuis, J. Amrute, R. J. A. Veltrop, P. Boor, K. Klingel, L. W. Van Laake, A. Vink, R. M. Hoogenboezem, E. M. J. Bindels, L. Schurgers, S. Sattler, D. Schapiro, R. K. Schneider, K. Lavine, H. Milting, I. G. Costa, J. Saez-Rodriguez and R. Kramann (2022). "Spatial multi-omic map of human myocardial infarction." *Nature* **608**(7924): 766-777.

Learmonth, M., A. Corker, S. Dasgupta and K. Y. DeLeon-Pennell (2023). "Regulation of cardiac fibroblasts by lymphocytes after a myocardial infarction: playing in the major league." *Am J Physiol Heart Circ Physiol* **325**(3): H553-H561.

Liang, Y., G. Shi, R. Cai, Y. Yuan, Z. Xie, L. Yu, Y. Huang, Q. Shi, L. Wang, J. Li and Z. Tang (2024). "PROST: quantitative identification of spatially variable genes and domain detection in spatial transcriptomics." *Nat Commun* **15**(1): 600.

Lopez, R., J. Regier, M. B. Cole, M. I. Jordan and N. Yosef (2018). "Deep generative modeling for single-cell transcriptomics." *Nat Methods* **15**(12): 1053-1058.

Lu, M., C. N. Christensen, J. M. Weber, T. Konno, N. F. Laubli, K. M. Scherer, E. Avezov, P. Lio, A. A. Lapkin, G. S. Kaminski Schierle and C. F. Kaminski (2023). "ERnet: a tool for the semantic segmentation and quantitative analysis of endoplasmic reticulum topology." *Nat Methods* **20**(4): 569-579.

Lu, M. Y., D. F. K. Williamson, T. Y. Chen, R. J. Chen, M. Barbieri and F. Mahmood (2021). "Data-efficient and weakly supervised computational pathology on whole-slide images." Nat Biomed Eng **5**(6): 555-570.

Ma, J., Y. He, F. Li, L. Han, C. You and B. Wang (2024). "Segment anything in medical images." Nat Commun **15**(1): 654.

Meylan, M., F. Petitprez, E. Becht, A. Bougouin, G. Pupier, A. Calvez, I. Giglioli, V. Verkarre, G. Lacroix, J. Verneau, C. M. Sun, P. Laurent-Puig, Y. A. Vano, R. Elaidi, A. Mejean, R. Sanchez-Salas, E. Barret, X. Cathelineau, S. Oudard, C. A. Reynaud, A. de Reynies, C. Sautes-Fridman and W. H. Fridman (2022). "Tertiary lymphoid structures generate and propagate anti-tumor antibody-producing plasma cells in renal cell cancer." Immunity **55**(3): 527-541 e525.

Pachitariu, M. and C. Stringer (2022). "Cellpose 2.0: how to train your own model." Nat Methods **19**(12): 1634-1641.

Polanski, K., R. Bartolome-Casado, I. Sarropoulos, C. Xu, N. England, F. L. Jahnsen, S. A. Teichmann and N. Yayon (2024). "Bin2cell reconstructs cells from high resolution Visium HD data." Bioinformatics **40**(9).

Ramos, G., U. Hofmann and S. Frantz (2016). "Myocardial fibrosis seen through the lenses of T-cell biology." J Mol Cell Cardiol **92**: 41-45.

Schmidt, U., M. Weigert, C. Broaddus and G. Myers (2018). Cell detection with star-convex polygons. Medical Image Computing and Computer Assisted Intervention, Springer.

Sekh, A. A., I. S. Opstad, G. Godtliebsen, Å. B. Birgisdottir, B. S. Ahluwalia, K. Agarwal and D. K. Prasad (2021). "Physics-based machine learning for subcellular segmentation in living cells." Nature Machine Intelligence **3**(12): 1071-1080.

Stirling, D. R., M. J. Swain-Bowden, A. M. Lucas, A. E. Carpenter, B. A. Cimini and A. Goodman (2021). "CellProfiler 4: improvements in speed, utility and usability." BMC Bioinformatics **22**(1): 433.

Stringer, C., T. Wang, M. Michaelos and M. Pachitariu (2021). "Cellpose: a generalist algorithm for cellular segmentation." Nat Methods **18**(1): 100-106.

Tepe, B., M. C. Hill, B. T. Pekarek, P. J. Hunt, T. J. Martin, J. F. Martin and B. R. Arenkiel (2018). "Single-Cell RNA-Seq of Mouse Olfactory Bulb Reveals Cellular Heterogeneity and Activity-Dependent Molecular Census of Adult-Born Neurons." Cell Reports **25**(10): 2689-2703.e2683.

Vahid, M. R., E. L. Brown, C. B. Steen, W. Zhang, H. S. Jeon, M. Kang, A. J. Gentles and A. M. Newman (2023). "High-resolution alignment of single-cell and spatial transcriptomes with CytoSPACE." Nat Biotechnol **41**(11): 1543-1548.

Wang, X., J. Zhao, E. Marostica, W. Yuan, J. Jin, J. Zhang, R. Li, H. Tang, K. Wang, Y. Li, F. Wang, Y. Peng, J. Zhu, J. Zhang, C. R. Jackson, J. Zhang, D. Dillon, N. U. Lin, L. Sholl, T. Denize, D. Meredith, K. L. Ligon, S. Signoretti, S. Ogino, J. A. Golden, M. P. Nasrallah, X. Han, S. Yang and K. H. Yu (2024). "A pathology foundation model for cancer diagnosis and prognosis prediction." Nature **634**(8035): 970-978.

Wei, R., S. He, S. Bai, E. Sei, M. Hu, A. Thompson, K. Chen, S. Krishnamurthy and N. E. Navin (2022). "Spatial charting of single-cell transcriptomes in tissues." Nat Biotechnol **40**(8): 1190-1199.

Xu, H., N. Usuyama, J. Bagga, S. Zhang, R. Rao, T. Naumann, C. Wong, Z. Gero, J. Gonzalez, Y. Gu, Y. Xu, M. Wei, W. Wang, S. Ma, F. Wei, J. Yang, C. Li, J. Gao, J. Rosemon, T. Bower, S. Lee, R. Weerasinghe, B. J. Wright, A. Robicsek, B. Piening, C. Bifulco, S. Wang and H. Poon (2024). "A whole-slide foundation model for digital pathology from real-world data." Nature **630**(8015): 181-188.

Xu, H., S. Wang, M. Fang, S. Luo, C. Chen, S. Wan, R. Wang, M. Tang, T. Xue, B. Li, J. Lin and K. Qu (2023). "SPACEL: deep learning-based characterization of spatial transcriptome architectures." Nat Commun **14**(1): 7603.

You, Y., Y. Fu, L. Li, Z. Zhang, S. Jia, S. Lu, W. Ren, Y. Liu, Y. Xu, X. Liu, F. Jiang, G. Peng, A. Sampath Kumar, M. E. Ritchie, X. Liu and L. Tian (2024). "Systematic comparison of sequencing-based spatial transcriptomic methods." Nat Methods **21**(9): 1743-1754.

Zhang, D., A. Schroeder, H. Yan, H. Yang, J. Hu, M. Y. Y. Lee, K. S. Cho, K. Susztak, G. X. Xu, M. D. Feldman, E. B. Lee, E. E. Furth, L. Wang and M. Li (2024). "Inferring super-resolution tissue architecture by integrating spatial transcriptomics with histology." Nat Biotechnol.

Zhao, E., M. R. Stone, X. Ren, J. Guenthoer, K. S. Smythe, T. Pulliam, S. R. Williams, C. R. Uyttingco, S. E. B. Taylor, P. Nghiem, J. H. Bielas and R. Gottardo (2021). "Spatial transcriptomics at subspot resolution with BayesSpace." Nat Biotechnol **39**(11): 1375-1384.

Zhu, J., Y. Wang, W. Y. Chang, A. Malewska, F. Napolitano, J. C. Gahan, N. Unni, M. Zhao, R. Yuan, F. Wu, L. Yue, L. Guo, Z. Zhao, D. Z. Chen, R. Hannan, S. Zhang, G. Xiao, P. Mu, A. B. Hanker, D. Strand, C. L. Arteaga, N. Desai, X. Wang, Y. Xie and T. Wang (2024). "Mapping cellular interactions from spatially resolved transcriptomics data." Nat Methods **21**(10): 1830-1842.

Zhu, S., N. Kubota, S. Wang, T. Wang, G. Xiao and Y. Hoshida (2024). "STIE: Single-cell level deconvolution, convolution, and clustering in in situ capturing-based spatial transcriptomics." Nat Commun **15**(1): 7559.

We truly appreciated each reviewer's comments, which are crucial for us to improve our study. In this revision, we have made several key updates to address the reviewers' comments:

1. **Discussion of newly published methods:** We expanded our comparison to include recently introduced methods (MISO, Spotiphy) in the Discussion and clarified how Thor differs from and complements these tools, focusing on where Thor's value lies.
2. **Enhanced benchmarking:** We compared Thor's performance to baseline approaches for cell missout simulations and Bin2Cell for Visium HD data imputation.
3. **Refined claims and figures:** We revised and refined the claims about suboptimal histology handling. Additionally, we refined the use cases for semi-supervised annotation, which differs from unsupervised tissue domain segmentation tools such as iStar or MISO. Moreover, we removed the uncertain T cell proliferation pathway, standardized color schemes for multi-method comparisons, and expanded the discussion of Thor's limitations.
4. **Code and documentation:** We improved code availability and documentation, detailing parameter settings and installation instructions for reproducibility.

We trust that these revisions address the critical concerns raised and enhance the overall clarity and robustness of our study.

REVIEWER COMMENTS

Reviewer #1 (Remarks to the Author):

The authors have addressed most of the minor comments but only a subset of the major concerns. While the manuscript has been improved, several key issues remain unresolved and should be addressed before publication.

Response: Thank you for reviewing our manuscript. In response to the key issues you raised, we carefully addressed key points by (1) adding a comprehensive discussion of Thor's place alongside newly published tools (MISO, Spotiphy), (2) providing concrete comparisons with baseline methods and Bin2Cell (3) refining our analysis of suboptimal histology images, (4) clarifying Thor's limitations, especially regarding multi-sample integration and gene expression gradients, and (5) refining user scope of Thor's semi-supervised annotations, setting apart from tissue domain segmentation functionalities in tools such as iStar. We believe these clarifications and additional results adequately resolve your concerns while highlighting Thor's unique capabilities and contributions.

Major Concerns:

Q1. Novelty and Comparison with Recent Methods

Several new methods have been published in the past few months, including MISO (Coleman et al., PMID: 39815104, Nature Methods, March 2025) and Spotiphy (Yang et al., PMID: 40074951, Nature Methods, March 2025), among others in high-impact journals. These methods offer robust and novel approaches, raising concerns regarding the novelty of Thor. A key question that remains unaddressed is: How does Thor improve upon these recently established methods? A more thorough comparison with these methods is necessary to justify Thor's contributions.

Response: We thank the reviewer for mentioning these recently published methods. Spotiphy uses external scRNA-seq data in conjunction with the cell locations segmented from the histology image to resolve cell-level gene expression from spot-level data; and MISO further integrates multiple diverse modalities, including gene expression, epigenetics, metabolomics, and tissue histology, into a unified framework, providing a comprehensive multi-modality perspective. Because these tools either focus on multiple data modalities or rely on additional datasets (e.g., external single-cell profiles), a direct head-to-head evaluation beyond our current scope. Therefore, we do not present a direct quantitative comparison with these latest tools. Instead, we highlight Thor's strengths and use cases that underscore its value as a complementary addition to the existing suite of spatial analysis methods.

- (i) Thor functions as a unified analysis platform, integrating diverse analytical tools into one system for streamlined analysis.
- (ii) Thor eliminates the need for complex training protocols and external scRNA-seq data, enhancing its accessibility and ease of use.

- (iii) Thor provides interactive visualization of gigapixel images of histology or other features on standard devices, facilitating detailed, intuitive exploration of large datasets.
- (iv) Preliminary results show Thor's applicability to emerging high-resolution spatial platforms, such as Visium HD.

These features of Thor complement existing methods by focusing on accessibility, scalability, and a unified analysis platform. On the other hand, Thor's limitation in degraded performance with suboptimal cell segmentation or multi-modal integration may be resolved by those new tools. We have expanded the discussion in the revised manuscript to highlight Thor's strengths and limitations (Lines 675 – 691).

Q2. Comparison with Existing Methods (Original Comment #1 & #2)

The authors claim that Thor maintains robust accuracy despite cell dropout (Response Figure 1, Figure S1b), yet no direct comparison with existing methods is provided. Benchmarking against other state-of-the-art tools would significantly strengthen the claims.

Response: We appreciate the reviewer's suggestion for additional benchmarking. Regarding how Thor performs with cells missed by cell segmentation from the histology image, we have added two baseline tools for comparison.

- (i) Method 1: Directly assigning the nearest spot-level gene expression data to detected cells, which represents a straightforward but commonly used approach.
- (ii) Method 2: Inferring gene expression at the "subspot" level (i.e., generating smaller artificial spots using BayesSpace) and subsequently assigning these inferred expressions to the nearest detected cells. This represents a more sophisticated attempt to resolve finer-grained spatial gene expression.

Our updated results show that Thor outperforms these baseline methods (**Response Figure 1**).

Response Figure 1 (Figure S1b): Thor inference has a reliable performance on simulation data. (a) Expression profile of a gene in simulated spot-resolution data (spot separation: 100

μm), the ground truth, and Thor-predicted single-cell data. (b) Thor shows robust accuracy with cell missouts or perturbations in the cell-cell network. NRMSE values provide a quantitative measure of the normalized deviation of Thor-inferred gene expression from the ground-truth gene expression.

Additionally, to provide more clarity on our simulation rationale, we have expanded the description of our experimental design in the manuscript as follows,

- (i) Our simulations mimic suboptimal histology conditions, such as poor image quality, insufficient staining, or overlapping cells, by randomly removing cells from the ground-truth cell set.
- (ii) We do not modify the underlying spot-level gene expression profiles, ensuring the only challenge introduced is the absence of certain cells, rather than gene expression dropouts.
- (iii) We assess the robustness of Thor's algorithm in predicting gene expression for the cells that remain, given missing cells resulting from segmentation issues. Consequently, those missing cells are excluded from further investigation.

By delineating these simulation conditions and benchmarking against baseline methods, our revised manuscript strengthens the scope of our analysis and our claim (Lines 172 – 189).

Similarly, the performance of Thor on Visium HD data imputation has not been compared with existing approaches such as Bin2Cell (Response Figure 2, Figure S26). A side-by-side comparison with these methods is essential to contextualize Thor's effectiveness.

Response: As recommended, we compared Thor with Bin2Cell for reconstructing single-cell gene expression from Visium HD (2 μm ; subcellular) bins in our in-house bladder cancer dataset. Bin2Cell assigns subcellular bin-level gene expression data to cells based on overlaps between these bins and nucleus masks (with expansion) identified through histology. Both Thor and Bin2Cell effectively reduce spurious expression signals in non-tissue regions, mitigating transcript diffusion artifacts (**Response Figure 2a**). For instance, both methods accurately localize an epithelial marker gene (*SPINK1*) to epithelial cells and remove signals from empty areas. We also noted that Bin2Cell left approximately half of the detected cells without assigned transcripts, whereas Thor predicted expression values to a substantially larger proportion of cells. Future investigations will help elucidate the underlying factors for those differences more thoroughly.

Thor's inference process utilizes H&E-based features to align gene expression more closely with histology, thereby facilitating cell-type determination (**Response Figure 2b**). In contrast, Bin2Cell relies on bin-level expression values and cell masks, which may lead to unassigned cells under complex overlapping conditions.

It is important to note that Thor's Visium HD data imputation remains preliminary. Further methodological refinements and more extensive benchmarking across diverse tissue types,

staining protocols, and segmentation pipelines are planned to further evaluate and optimize Thor's performance. We trust that this comparison clarifies Thor's effectiveness for Visium HD imputation. We have included the results and a brief discussion in our revised manuscript (Lines 555 – 568).

Response Figure 2: Thor imputes Visium HD data and reconstructs gene expression patterns that align with pathology annotations in a bladder cancer sample. (a) Pathology annotations (left) highlight immune cells (ROI 1), invasive carcinoma (ROI 2), and a tumor fragment (ROI 3). Gene expression patterns inferred by Thor (middle), from Visium HD (right), and inferred by Bin2Cell (right). The orange arrow points to a region with no cell. (b) Cell

clusters with a zoomed-in view of ROI 1 by Thor (left), bin clusters by Visium HD 8 μm bin data (middle) and cell clusters by Bin2Cell (right; with empty cells removed).

Q3. Discussion of Limitations

The Discussion section should be expanded to explicitly acknowledge Thor's limitations. For instance, Thor exhibits reduced performance in capturing gradient patterns for some genes, likely due to challenges in detecting flat cell nuclei. The authors should comment on this issue and propose potential solutions. Additionally, Thor's limitations in multi-sample integration should be discussed, as this remains a key challenge in the field.

Response: We thank the suggestion. As suggested, we have expanded the discussion regarding Thor's limitations and proposed solutions.

In the revised manuscript (Lines 740 – 745), “ ... Second, Thor exhibits reduced performance in capturing gradient patterns for some genes, likely due to challenges in detecting flat cell nuclei in peripheral areas. Incorporating higher precision imaging techniques (e.g. cell-membrane staining) or alternative analysis methods (e.g. iStar) that do not rely on cell segmentation are recommended. Third, Thor does not support multi-sample integration that handles batch effects in transcriptomics data and histological variability between tissue sections ...”

Q4. Color Scheme in Response Figure 4 (Figure S20)

The interpretation of Response Figure 4 is challenging because different color keys are used for results from the three methods. To improve readability, the same or similar colors should be used to represent the same cell type across methods.

Response: Thank you for the suggestion. We have updated Figure S20 by using the same color keys for the cell types across the three methods for readability.

Q5. T Cell Proliferation Inference (Original Comment #5)

The authors have not adequately addressed the concern regarding T cell proliferation inference. Additional supporting evidence would be necessary to strengthen this claim.

Response: We appreciate the reviewer's insight regarding T cell proliferation inference. In our study, we conducted an unbiased GO pathway enrichment analysis on heart failure samples to differentiate Thor-annotated fibrotic regions from other areas. However, due to the unavailability of the samples (the data was downloaded from the public domain) and the resulting lack of sufficient experimental evidence for T cell proliferation, we have removed the pathway from our analysis. These revisions are reflected in Figure 3 and the related section (Lines 371 – 384).

6. Validation of Single-Cell CNV Inference (Original Comment #6)

Comparing Thor's single-cell CNVs to spot-level CNVs is not sufficient, as both are inferred from spatial transcriptomics data. Neither method directly reflects genomic aneuploidy levels, making external validation crucial. To demonstrate accuracy, paired genomic data should be used to evaluate the consistency of Thor's CNV predictions.

Alternatively, a FISH assay could be performed to validate a subset of genes in regions where single-cell CNV gains were detected but missed by spot-level CNV analysis.

Response: We appreciate the reviewer's emphasis on the need for additional validation of single-cell CNV predictions. Because the dataset was downloaded from the 10x Genomics website, we did not have access to matched tissues or paired genomics data. To address this limitation, we devised the following approach to assess the accuracy of the CNV inferred by Thor (see the schematic of our approach in **Response Figure 3a**).

First step, we obtained a reference scRNA-seq dataset of DCIS, in which the CNV profiles were predicted by CopyKAT and validated by paired Whole Genome Sequencing (WGS) data (Gao, Bai et al. *Nat Biotechnol* 2021). This dataset provides a solid, independently validated reference for CNV profiles.

Second step, we applied CytoSPACE to map the reference scRNA-seq data onto the Visium dataset in our study. This allowed us to derive the spot-level aneuploid cell proportions by assigning the validated CNV profiles of the single cells to individual spots.

As anticipated, tumor regions show higher aneuploid proportions (**Response Figure 3b**). Overall, those proportions (as ground truth) aligned well with the distribution of aneuploid cells predicted by Thor. Specifically, we would like to highlight that in tumor region T7, Thor correctly predicted a mixture of aneuploid and diploid cells (**Response Figure 3c**). In contrast, the result obtained directly from CopyKAT on the Visium data classified that all the spots in the region as aneuploid, possibly due to aggregation of cell expressions in the spots.

While this approach does not replace direct genomic or FISH-based confirmation, it leverages independent single-cell data which was validated by WGS. We believe these analyses provide a reasonable assessment of Thor's CNV accuracy in the absence of directly paired genomic data. We have included these results and the methodology in the revised manuscript on Lines 503 – 504 and Lines 1006 – 1013.

Response Figure 3. Validation of CNV profiles. (a) Design of our approach for calculating WGS-validated Visium CNV profiles. CytoSPACE was utilized to transfer CNV profiles of single cells, which were validated by WGS DNA-seq data (Figures of single-cell CNV and WGS were adapted from Ref. (Gao, Bai et al. Nat Biotechnol 2021)), to Visium spots. (b) Spot-level aneuploid proportions mapped by CytoSPACE. (c) Aneuploid distribution in tumor region T7 predicted by Thor, raw Visium, and mapped by CytoSPACE from validated-reference data.

7. TLS Analysis (Original Comment #7)

The TLS analysis lacks novelty and does not introduce significant new findings. This section should be further condensed to maintain focus on Thor's core contributions.

Response: Thank you very much for this comment. Follow the reviewer's suggestion, we have condensed the section and adjusted the language to emphasize Thor's core contributions without overstating any novel biological findings (Lines 512 – 534).

8. Robustness with Imperfect Histology Images (Original Comment #8)

The claim that Thor is robust to imperfect histology images (Response Figure 6, Figure S1b) remains unsubstantiated. A comparison with existing methods that handle histological variations is necessary to support this claim.

Response: We thank the reviewer for raising this important concern regarding Thor's robustness with suboptimal histology images. We revisited our manuscript and recognized that our concluding remarks in the simulation section might overstate the scope of our analyses. Specifically, the statement "Collectively, these analyses highlight Thor's accuracy and robustness in the presence of suboptimal histology or transcriptomics data..." may be misleading because Figure S1b only simulates two relatively basic conditions of suboptimal images, rather than the wide range of imaging issues encountered in practice.

In the simulations, we focused on the evaluation of Thor's algorithm in predicting gene expression for the detected cells. When only a moderate number of cells are missed, Thor accurately predicts gene expression for those that are detected; however, its performance declines when a larger proportion of cells go undetected—particularly since Thor does not infer expression for undetected cells. Real-world issues such as out-of-focus imaging or poor staining can further degrade image quality, making it advisable in those cases to use alternative tools that do not rely on cell detection (e.g., iStar). Meanwhile, because Thor depends on external nucleus segmentation and image preprocessing tools, direct assessment of those external tools in handling histological variations, therefore, lies beyond the context of this analysis.

We have revised the language that might suggest broader robustness than tested in our simulation and extended our discussion on Thor's limitations (Lines 738 – 739).

9. Granularity of Semi-Supervised Annotation (Original Comment #9)

The quantitative assessment of Thor's semi-supervised annotation focuses primarily on large-scale tissue structures (e.g., vessel, node, adipose, fibrosis). However, it lacks fine-grained annotations, which are essential for a comprehensive evaluation.

For example, other methods, such as MISO (Coleman et al., PMID: 39815104) and iStar, reveal significantly more detailed tissue structures. A comparative analysis would help demonstrate the advantages and potential shortcomings of Thor's annotation approach.

While the manuscript has improved, the concerns above must be addressed to ensure the robustness and novelty of Thor's contributions. I encourage the authors to provide additional comparisons, validations, and refinements to strengthen their work.

Response: We appreciate the reviewer's comments on the granularity of Thor's semi-supervised annotation (SSA) tool. Our manuscript has demonstrated that Thor infers cell-level gene expression and fine-grained spatial patterns across various tissue types, including thin cell layers and neuron subtypes in the mouse olfactory bulb (**Response Figure 4**).

Response Figure 4 (Figure S10). Spatial expression profiles of marker genes in glomerular and mitral layers from the mouse olfactory bulb.

SSA, specifically designed for incorporating user-input regions and quickly searching for similar cells in the combined feature space of geometry, histology, and transcriptomics. Leveraging Thor-inferred high-resolution gene expression levels and high-resolution histological features extracted from the whole slide image, SSA can precisely retrieve refined cells. We anticipate that SSA may conduct a finer-grained search when the user provides appropriate reference cells. By selecting representative cell populations of interest (the “curated region”) at the sub-regional or sub-cell-type level, SSA navigates to label cells at a higher resolution.

In our previous quantitative evaluation, we focused on four representative, common tissue structures in heart tissues. However, the lack of robust fine-grained pathological spot-level annotations poses challenges for a thorough evaluation at more specialized levels. Nevertheless, we have added more example areas that show SSA identified fine structures (<spot size; **Response Figure 5**).

Response Figure 5. Thor SSA marks fine, localized tissue structures. Positive/background spots (cells) are colored in orange/blue.

It is worth emphasizing that although SSA can be leveraged for fine-grained insights, it is designed to run exclusively within Mjolnir for targeted exploration in a semi-supervised fashion. Meanwhile, comprehensive global segmentation requires fully unsupervised techniques such as iStar. MISO and iStar excel at fine-resolution tissue domain segmentation, delineating global tissue architectures without requiring substantial prior knowledge. We view Thor's SSA as complementary to such approaches, enabling targeted exploration and searching/cell retrieval of user-guided cellular morphologies or localized patterns of interest. Combining strengths of these methods, researchers can achieve both an overarching view of tissue morphology and a user-engaged, cell-by-cell search and follow-up molecular analyses. Accordingly, we have refined the description of the tool in the revised manuscript to provide a clearer context for Thor's contributions while acknowledging its scope (Lines 647 – 656).

Reviewer #2 (Remarks to the Author):

Response: Thank you for reviewing our manuscript. Please find our point-by-point responses in the combined report.

Reviewer #3 (Remarks to the Author):

The author provided a good explanation and appropriate analysis for our comments, without further questions.

Response: We truly appreciate the reviewer's encouragement and compliment.

Reviewer #4 (Remarks to the Author):

Thank you for your thorough revisions. Overall, the previous concerns have been well addressed in this version of the manuscript.

With the emergence of methods that cover whole genes with higher resolution, which are widely applied in this field, like Visium HD, it is good to see that these advancements have been incorporated into this revision. However, it would be beneficial to provide detailed documentation on parameter optimization when making the related code available on GitHub. This will help ensure reproducibility and facilitate further research based on your work.

Response: We thank the reviewer for the constructive suggestions which significantly improved our manuscript. Following your suggestions, we now provide more detailed documentation on parameter optimization — especially for the main class for inference – for reproducibility and usability. Please find the documentation in the following supplementary materials: the docstrings (`finest.set_params` in `source_codes/Thor/src/thor/finest.py`) and the webpage (`website_htmls/_autosummary/thor.fineST.html`).

Reviewer #4 (Remarks on code availability):

The code is well described, and example notebooks are good for understanding how to run this code.

As a future work, after publishing this code to the public, detailed parameter settings and installation methods for various users could be beneficial.

Response: Thank you for the helpful suggestion. In response, we have added more detailed descriptions of key parameter settings across all case studies to guide users in adapting Thor to their own datasets (under `source_codes/Thor/parameters/*.json`). Those parameter files will be part of the main GitHub repository.

Additionally, we have refined the Python environment specification and clarified package dependencies to ensure robust installation (under `source_codes/Thor/README.rst` and webpage `website_htmls/installation.html`). Installation of Thor has now been successfully tested on Linux, macOS, and Windows systems. These updates aim to make Thor more accessible and user-friendly across diverse computing environments.

References

Gao, R., S. Bai, Y. C. Henderson, Y. Lin, A. Schalck, Y. Yan, T. Kumar, M. Hu, E. Sei, A. Davis, F. Wang, S. F. Shaitelman, J. R. Wang, K. Chen, S. Moulder, S. Y. Lai and N. E. Navin (2021). "Delineating copy number and clonal substructure in human tumors from single-cell transcriptomes." Nat Biotechnol **39**(5): 599-608.

Reviewer #1

Comment: The authors have made efforts to address several of the comments from the previous review round, including expanding the limitations section and rephrasing certain text. However, not all prior concerns have been fully resolved. The authors should ensure that all remaining points are thoroughly addressed, either in the discussion or limitations sections—particularly with regard to the method's novelty and robustness in the context of recently published tools that were not included in the benchmarking of this study. Additionally, Figure 1 could be enhanced to more effectively illustrate the workflow and highlight the novelty of the proposed method. A clearer visual representation would aid in conveying the unique aspects and contributions of this work.

Response: We thank the reviewer for this constructive feedback and have improved the manuscript accordingly.

First, we expanded and re-phrased the discussion of methodological novelty and robustness (Lines 675 to 695). We now explain more clearly how Thor differs from, and complements, recent tools. Spotiphy generates single cell-level gene expression from spot-level by leveraging scRNA-seq references and cell masks segmented from histology image. Although both inferring single cell-level gene expression, Thor leverages histological features of segmented cells in addition to the spatial coordinates to construct a cell-cell graph for Markov diffusion. Thor does not require scRNA-seq data. MISO integrates multiple orthogonal modalities based on spot-level features and excels in spatial domain segregation. MISO extracts histology features of H&E image patches using a deep-learning model (Vision Transformer trained on H&E images). In contrast to MISO, Thor operates at single cell level yet only focusing on histology images (including IF images), and transcriptomics. We also emphasize Thor's demonstrated robustness across various tissue types, two spatial resolutions (standard Visium and Visium HD), and extensive simulations of cell-segmentation miss-outs, false cell-cell connections, and gene dropouts, where it achieves low normalized-error scores and agrees closely with external ground-truth such as WGS-validated CNV profiles and pathology annotations. These results position Thor as a lightweight, resource-efficient alternative for working with histology-plus-transcriptome data at single-cell level .

Second, we redesigned Figure 1 to present a clearer, step-by-step workflow, and explicitly highlighted innovative components of Thor: namely, construction of cell-cell graph based on features in the joint space of locations, histology, and transcriptomics, as well as single-cell level gene expression inference that is independent of external scRNA-seq data.

We hope these revisions satisfactorily address the reviewer's remaining concerns regarding novelty, robustness, and visual clarity.